# Non-Smooth Weakly-Convex Finite-sum Coupled Compositional Optimization

**Quanqi Hu**
Department of Computer Science
Texas A&M University
College Station, TX 77843
quanqi-hu@tamu.edu

**Dixian Zhu**
Department of Genetics
Stanford University
Stanford, CA 94305
dixian-zhu@stanford.edu

**Tianbao Yang**
Department of Computer Science
Texas A&M University
College Station, TX 77843
tianbao-yang@tamu.edu

## Abstract

This paper investigates new families of compositional optimization problems, called **n**on-**s**mooth **w**eakly-**c**onvex **f**inite-sum **c**oupled **c**ompositional **op**timization (NSWC FCCO). There has been a growing interest in FCCO due to its wide-ranging applications in machine learning and AI, as well as its ability to address the shortcomings of stochastic algorithms based on empirical risk minimization. However, current research on FCCO presumes that both the inner and outer functions are smooth, limiting their potential to tackle a more diverse set of problems. Our research expands on this area by examining non-smooth weakly-convex FCCO, where the outer function is weakly convex and non-decreasing, and the inner function is weakly-convex. We analyze a single-loop algorithm and establish its complexity for finding an $\epsilon$-stationary point of the Moreau envelop of the objective function. Additionally, we also extend the algorithm to solving novel non-smooth weakly-convex tri-level finite-sum coupled compositional optimization problems, which feature a nested arrangement of three functions. Lastly, we explore the applications of our algorithms in deep learning for two-way partial AUC maximization and multi-instance two-way partial AUC maximization, using empirical studies to showcase the effectiveness of the proposed algorithms.

## 1 Introduction

In this paper, we consider two classes of non-convex compositional optimization problems. The first class is formulated as following:

$$\min_{\mathbf{w}\in\mathbb{R}^d} F(\mathbf{w}) := \frac{1}{n}\sum_{i\in\mathcal{S}} f_i(\mathbb{E}_{\xi\sim\mathcal{D}_i}[g_i(\mathbf{w};\xi)]), \tag{1}$$

where $\mathcal{S}$ denotes a finite set of $n$ items and $\mathcal{D}_i$ denotes a distribution that could depend on $i$. The second class is given by:

$$\min_{\mathbf{w}\in\mathbb{R}^d} F(\mathbf{w}) := \frac{1}{n_1}\sum_{i\in\mathcal{S}_1} f_i\left(\frac{1}{n_2}\sum_{j\in\mathcal{S}_2} g_i(\mathbb{E}_{\xi\sim\mathcal{D}_{i,j}}[h_{i,j}(\mathbf{w};\xi)])\right), \tag{2}$$

where $\mathcal{S}_1$ denotes a finite set of $n_1$ items and $\mathcal{S}_2$ denotes a finite set of $n_2$ items and $\mathcal{D}_{ij}$ denotes a distribution that could depend on $(i,j)$. For simplicity of discussion, we denote by $g_i(\mathbf{w}) =$

$\mathbb{E}_{\xi \sim \mathcal{D}_i}[g_i(\mathbf{w}; \xi)] : \mathbb{R}^d \to \mathbb{R}^{d_1}$ and by $h_{i,j}(\mathbf{w}) = \mathbb{E}_{\xi \sim \mathcal{D}_{i,j}}[h_{i,j}(\mathbf{w}; \xi)] : \mathbb{R}^d \to \mathbb{R}^{d_2}$. For both classes of problems, we focus our attention on **non-convex** $F$ **with non-smooth non-convex functions** $f_i$ **and** $g_i$, which, to the best of our knowledge, has not been studied in any prior works.

The first problem (1) with smooth functions $f_i$ and $g_i$ has been explored in previous works [26, 15, 21, 33], which is known as finite-sum coupled compositional optimization (FCCO). It is subtly different from standard stochastic compositional optimization (SCO) [27] and conditional stochastic optimization (CSO) [13]. FCCO has been successfully applied to optimizing a wide range of X-risks [33] with convergence guarantee, including smooth surrogate losses of areas under the curves [20] and ranking measures [21], listwise losses [21], and contrastive losses [36]. The second problem (2) is a novel class and is referred to as tri-level finite-sum coupled compositional optimization (TCCO). Both problems differ from traditional two-level or multi-level compositional optimization due to the coupling of variables $i, \xi$ in (1) or the coupling of variables $i, j, \xi$ in (2) at the inner most level.

One limitation of prior works about non-convex FCCO is that their convergence analysis heavily rely on the smoothness conditions of $f_i$ and $g_i$ [26, 15]. This raises a concern about whether existing techniques can be leveraged for solving *non-smooth non-convex FCCO problems* with non-asymptotic convergence guarantee. Non-smooth non-convex FCCO and TCCO problems have important applications in ML and AI, e.g., group distributionally robust optimization [4] and two-way partial AUC maximization for deep learning [44]. We defer discussions and formulations of these problems to Section 5. The difficulty for solving smooth FCCO lies at high costs of computing a stochastic gradient $\nabla g_i(\mathbf{w}) \nabla f_i(g_i(\mathbf{w}))$ for a randomly sampled $i$ and the overall gradient $\nabla F(\mathbf{w})$. To approximate the stochastic gradient, a variance-reduced estimator of $g_i(\mathbf{w}_t)$ denoted by $u_{i,t}$ is usually maintained and updated for sampled data in the mini-batch $i \in \mathcal{B}_t$. As a result, the stochastic gradient can be approximated by $\nabla g_i(\mathbf{w}_t; \xi_t) \nabla f_i(u_{i,t})$, where $\xi_t \sim \mathcal{D}_i$ is a random sample. The overall gradient can be estimated by averaging the stochastic gradient estimator over the mini-batch or using variance-reduction techniques. A key insight of the convergence analysis for smooth FCCO is to bound the following error using the $L$-smoothness of $f_i$, which reduces to bounding the error of $u_{i,t}$ for estimating $g_i(\mathbf{w}_t)$:

$$\|\nabla g_i(\mathbf{w}_t; \xi_t) \nabla f_i(u_{i,t}) - \nabla g_i(\mathbf{w}_t; \xi_t) \nabla f_i(g_i(\mathbf{w}_t))\|^2 \le \|\nabla g_i(\mathbf{w}_t; \xi_t)\|^2 L \|u_{i,t} - g_i(\mathbf{w}_t)\|^2.$$

A central question to be addressed in this paper is *"Can these gradient estimators be used in stochastic optimization for solving non-smooth non-convex FCCO with provable convergence guarantee"?* To address this question we focus our attention on a specific class of FCCO/TCCO called **non-smooth weakly-convex (NSWC) FCCO/TCCO**. This approach aligns with many established works on NSWC optimization [6–9]. Nevertheless, NSWC FCCO/TCCO is more complex than a standard weakly-convex optimization problem because an unbiased stochastic subgradient is not readily accessible. In addition, the convergence measure in terms of the gradient norm of smooth non-convex objectives is not applicable to weakly convex optimization, which will complicate the analysis involving the biased stochastic gradient estimator $\partial g_i(\mathbf{w}_t; \xi_t) \partial f_i(u_i^t)$ [1].

**Contributions.** A major contribution of this paper is to present *novel convergence analysis* of single-loop stochastic algorithms for solving NSWC FCCO/TCCO problems, respectively. In particular,

- For non-smooth FCCO, we analyze the following single-loop updates:

$$\mathbf{w}_{t+1} = \mathbf{w}_t - \eta \frac{1}{B} \sum_{i \in \mathcal{B}_t} \partial g_i(\mathbf{w}_t; \xi_t) \partial f_i(u_{i,t}), \tag{3}$$

where $\mathcal{B}_t$ is a random mini-batch of $B$ items, and $u_{i,t}$ is an appropriate variance-reduced estimator of $g_i(\mathbf{w}_t)$ that is updated only for $i \in \mathcal{B}_t$ at the $t$-th iteration. To overcome the non-smoothness, we adopt the tool of Moreau envelop of the objective as in previous works [6, 7]. The key difference of our convergence analysis from previous ones for smooth FCCO is that we bound the inner product $\langle \mathbb{E}_i \partial g_i(\mathbf{w}) \partial f_i(u_{i,t}), \widehat{\mathbf{w}}_t - \mathbf{w}_t \rangle$, where $\widehat{\mathbf{w}}_t$ is the solution of the proximal mapping of the objective at $\mathbf{w}_t$. To this end, specific conditions of $f_i, g_i$ are imposed, i.e., $f_i$ is weakly convex and non-decreasing and $g_i(\mathbf{w})$ is weakly convex, under which we establish an iteration complexity of $T = \mathcal{O}(\epsilon^{-6})$ for finding an $\epsilon$-stationary point of the Moreau envelope of $F(\cdot)$.

- For non-smooth TCCO, we analyze the following single-loop updates:

$$\mathbf{w}_{t+1} = \mathbf{w}_t - \eta \frac{1}{B_1} \sum_{i \in \mathcal{B}_1^t} \left[ \frac{1}{B_2} \sum_{j \in \mathcal{B}_2^t} \partial h_{i,j}(\mathbf{w}_t; \xi_t) \partial g_i(v_{i,j,t}) \right] \partial f_i(u_{i,t}), \tag{4}$$

---

[1] We use $\nabla$ to denote gradient of a differentiable function and $\partial$ to denote a subgradient of a non-smooth function.

Table 1: Comparison with prior works for solving (1) and (2). In the monotonicity column, notation ↑ means the given function is required to be non-decreasing. If not specified, the given function is only required to be monotone.

| Method | Objective | Smoothness | Weak Convexity | Monotonicity | Complexity |
|--------|-----------|-----------|----------------|--------------|-----------|
| SOX [26] | (1) | $f_i, g_i$ | none | none | $\mathcal{O}(\epsilon^{-4})$ |
| MSVR [15] | (1) | $f_i, g_i$ | none | none | $\mathcal{O}(\epsilon^{-3})$ |
| SONX (Ours) | (1) | none | $f_i, g_i$ | $f_i \uparrow$ | $\mathcal{O}(\epsilon^{-6})$ |
| SONT (Ours) | (2) | none | $f_i, g_i, h_{i,j}$ | $f_i \uparrow, g_i \uparrow$ | $\mathcal{O}(\epsilon^{-6})$ |
| SONT (Ours) | (2) | $h_{i,j}$ | $f_i, g_i$ | $f_i \uparrow, g_i$ | $\mathcal{O}(\epsilon^{-6})$ |

where $\mathcal{B}_t^1$ and $\mathcal{B}_t^2$ are random mini-batches of $B_1$ and $B_2$ items, respectively, and $u_{i,t}$ is an appropriate variance-reduced estimator of $\frac{1}{n_2} \sum_{j \in \mathcal{S}_2} g_i(h_{ij}(\mathbf{w}_t))$ that is updated only for $i \in \mathcal{B}_t^1$, and $v_{i,j,t}$ is an appropriate variance-reduced estimator of $h_{i,j}(\mathbf{w}_t)$ that is updated only for $i \in \mathcal{B}_t^1, j \in \mathcal{B}_t^2$. To prove the convergence, we impose conditions of $f_i, g_i, h_{i,j}$, i.e., $f_i$ is weakly convex and non-decreasing and $g_i(\cdot)$ is weakly convex and non-decreasing (or monotonic), $h_{ij}$ is weakly convex (or smooth), and establish an iteration complexity of $T = \mathcal{O}(\epsilon^{-6})$ for finding an $\epsilon$-stationary point of the Moreau envelope of $F(\cdot)$.

- We extend the above algorithms to solving (multi-instance) two-way partial AUC maximization for deep learning, and conduct extensive experiments to verify the effectiveness of the both algorithms.

## 2 Related work

**Smooth SCO.** There are many studies about two-level smooth SCO [27, 38, 10, 19, 3, 28] and multi-level smooth SCO [32, 32, 1, 39]. The complexities of finding an $\epsilon$-stationary point for two-level smooth SCO have been improved from $O(\epsilon^{-5})$ [27] to $O(\epsilon^{-3})$ [19], and that for multi-level smooth SCO have been improved from a level-dependent complexity of $O(\epsilon^{-(7+K)/2})$ [32] to a level-independent complexity of $O(\epsilon^{-3})$ [32], where $K$ is the number of levels. The improvements mostly come from using advanced variance reduction techniques for estimating each level function or its Jacobian and for estimating the overall gradient. Two stochastic algorithms have been developed in [13] for CSO but suffer a limitation of requiring large batch sizes.

**Smooth FCCO.** FCCO was first introduced in [20] for optimizing average precision. Its algorithm and convergence analysis was improved in [26] and [15]. The former work [26] proposed an algorithm named SOX by using moving average (MA) to estimate the inner function values and the overall gradient. In the smooth non-convex setting, SOX is proved to achieve an iteration complexity of $\mathcal{O}(\epsilon^{-4})$. The latter work [15] proposed a novel multi-block-single-probe variance reduced (MSVR) estimator for estimating the inner function values, which helps achieve a lower iteration complexity $\mathcal{O}(\epsilon^{-3})$. Recently, [11] proposed an extrapolation based estimator for the inner function, which yields a method with a complexity that matches MSVR when $n \leq \epsilon^{2/3}$. These techniques have been employed for optimizing various X-risks, including contrastive losses [36], ranking measures and listwise losses [21], and other objectives [26, 15]. However, all of these prior works assume the smoothness of $f_i$ and $g_i$. Hence, their analysis is not applicable to NSWC FCCO problems. Our novel analysis of a simple algorithm for NSWC FCCO problems yields an iteration complexity of $O(\epsilon^{-6})$ for using the MSVR estimators of the inner functions. The comparison with [26, 15] is shown in Table 1.

**Non-smooth Weakly Convex Optimization.** Analysis of weakly convex optimization with unbiased stochastic subgradients was pioneered by [6, 7]. Optimization of compositional functions that are weakly convex have been tackled in earlier works [8, 9], where the inner function is deterministic or does not involve coupling between two random variables. A closely related work to our NSWC FCCO is weakly-convex concave minimax optimization [22]. Assuming $f_i$ is convex, (1) can be written as: $\min_{\mathbf{w}} \max_{\pi \in \mathbb{R}^n} \frac{1}{n} \sum_{i \in \mathcal{S}} \langle \pi_i, g_i(\mathbf{w}) \rangle - f_i^*(\pi_i)$, where $f_i^*(\cdot)$ is the convex conjugate of $f_i$. It can be solved using existing methods [22, 31, 41, 43, 17] but with several limitations: (i) the algorithms in [22, 31, 41, 43] have a comparable complexity of $O(1/\epsilon^6)$ but have unnecessary double loops which require setting the number of iterations for the inner loop; (ii) the algorithm in [17] is single loop but has a worse complexity of $\mathcal{O}(1/\epsilon^8)$; (iii) these existing algorithms and analysis does not account for complexity of updating all coordinates of $\pi$, which could be prohibitive in

many applications; iv) these approaches are not applicable to NSWC FCCO/TCCO with weakly convex $f_i$. In fact, the double loop algorithm has been leveraged and extended to solving the two-way partial AUC maximization problem, a special case of NSWC FCCO [44], by sampling and updating a batch of coordinates of $\pi$ at each iteration. However, it is less practical thus not implemented and its analysis did not explicitly show the convergence rate dependency on $n_+, n_-$ and the block batch size.

A special case of NSWC SCO problem was considered in [46], which is given by
$$\min_{x \in \mathcal{X}} f(x, g(x)), \text{ with } f(x, u) = \mathbb{E}_\zeta[u + \varkappa \max(0, g(x; \zeta) - u)], \quad g(x) = \mathbb{E}_\xi[g(x; \xi)].$$
They proposed two methods, SCS for smooth $g(x)$ and SCS with SPIDER for non-smooth $g(x)$. For both proposed methods, they proved a sample complexity of $\mathcal{O}(1/\epsilon^6)$ for achieving an $\epsilon$-stationary point of the objective's Moreau envelope [2]. We would like to remark that the above problem with a non-smooth $g(x)$ is a special case of NSWC FCCO with only a convex outer function, one block and no coupled structure. Nevertheless, their algorithm for non-smooth $g(\cdot)$ suffers a limitation of requiring a large batch size in the order of $O(1/\epsilon^2)$ for achieving the same convergence.

Finally, we would like to mention that non-smooth convex or strongly convex SCO problems have been considered in [27, 42, 26], which, however, are out of scope of the present work.

## 3 Preliminaries

Let $\| \cdot \|$ be the Euclidean norm of a vector and spectral norm of a matrix. We use $\Pi_C[\cdot]$ to denote the Euclidean projection onto $\{v \in \mathbb{R}^m : \|v\| \le C\}$. For vectors, inequality notations including $\le, \ge, >, <$ are used to denote element-wise inequality. For an expectation function $f(\cdot) = \mathbb{E}_\xi[f(\cdot; \xi)]$, let $f(\cdot; \mathcal{B}) = \frac{1}{|\mathcal{B}|} \sum_{\xi \in \mathcal{B}} f(\cdot; \xi)$ be its stochastic unbiased estimator evaluated on a sample batch $\mathcal{B}$. A stochastic unbiased estimator is said to have bounded variance $\sigma^2$ if $\mathbb{E}_\xi[\|f(\cdot) - f(\cdot; \xi)\|^2] \le \sigma^2$. The Jacobian matrix of function $f : \mathbb{R}^{m_1} \to \mathbb{R}^{m_2}$ is in dimension $\mathbb{R}^{m_1 \times m_2}$. We recall the definition of general subgradient and subdifferential following [6, 24].

**Definition 3.1** (subgradient and subdifferential). Consider a function $f : \mathbb{R}^n \to \mathbb{R} \cup \{\infty\}$ and a point with $f(x)$ finite. A vector $v \in \mathbb{R}^n$ is a general subgradient of $f$ at $x$, if
$$f(y) \ge f(x) + \langle v, y - x \rangle + o(\|y - x\|), \quad \text{as } y \to x.$$
The subdifferential $\partial f(x)$ is the set of subgradients of $f$ at point $x$.

For simplicity, we abuse the notation and also use $\partial f(x)$ to denote one subgradient from the corresponding subgradient set when no confusion could be caused. We use $\partial f(x; \mathcal{B})$ to represent a stochastic unbiased estimator of the subgradient $\partial f(x)$ that is evaluated on a sample batch $\mathcal{B}$. A function is called $C^1$-smooth if it is continuously differentiable. A function $f = (f_1, \ldots, f_{m_2}) : \mathbb{R}^{m_1} \to \mathbb{R}^{m_2}$ is called monotone if $\forall i \in \{1, \ldots, m_2\}$, $f_i : \mathbb{R}^{m_1} \to \mathbb{R}$ is monotone with respect to each element of the input. Note that if a Lipschitz continuous function $f : O \to \mathbb{R}^{m_2}$ is assumed to be non-increasing (resp. non-decreasing), where the domain $O \subset \mathbb{R}^{m_1}$ is open, then all subgradients of $f$ are element-wise non-positive (resp. non-negative). We refer the details to Appendix D.1.

A function $f$ is $C$-*Lipschitz continuous* if $\|f(x) - f(y)\| \le C\|x - y\|$. A differentiable function $f$ is $L$-*smooth* if $\|\nabla f(x) - \nabla f(y)\| \le L\|x - y\|$. A function $f : \mathbb{R}^d \to \mathbb{R} \cup \{\infty\}$ is $\rho$-*weakly-convex* if the function $f(\cdot) + \frac{\rho}{2}\|\cdot\|^2$ is convex. A vector-valued function $f : \mathbb{R}^d \to \{\mathbb{R} \cup \{\infty\}\}^m$ is called $\rho$-weakly-convex if it is $\rho$-weakly-convex for each output. It is difficult sometimes impossible to find an $\epsilon$-stationary point of a non-smooth weakly-convex function $F$, i.e., $\text{dist}(0, \partial F(\mathbf{w})) \le \epsilon$. For example, an $\epsilon$-stationary point of function $f(x) = |x|$ does not exist for $0 \le \epsilon < 1$ unless it is the optimal solution. To tackle this issue, [6] proposed to use the stationarity of the problem's Moreau envelope as the convergence metric, which has become a standard metric for solving weakly-convex problems [7, 22, 31, 41, 43, 17]. Given a weakly-convex function $\varphi : \mathbb{R}^m \to \mathbb{R}$, its Moreau envelope and proximal map with $\lambda > 0$ are constructed as
$$\varphi_\lambda(x) := \min_y \{\varphi(y) + \frac{1}{2\lambda}\|y - x\|^2\}, \quad \text{prox}_{\lambda\varphi}(x) := \arg\min_y \{\varphi(y) + \frac{1}{2\lambda}\|y - x\|^2\}.$$

The Moreau envelope is an implicit smoothing of the original problem. Thus it attains a continuous differentiation. As a formal statement, the following lemma follows from standard results [6, 18].

**Lemma 3.2.** *Given a $\rho$-weakly-convex function $\varphi$ and $\lambda < \rho^{-1}$, the envelope $\varphi_\lambda$ is $C^1$-smooth with gradient given by $\nabla \varphi_\lambda(x) = \lambda^{-1}(x - prox_{\lambda\varphi}(x))$.*

---

[2]It is notable that we use a slightly different definition of $\epsilon$-stationary point with $\|\nabla F_\rho(\mathbf{w})\|^2 \le \epsilon^2$.

---

**Algorithm 1** Stochastic Optimization algorithm for Non-smooth FCCO (SONX)

---

1: Initialization: $\mathbf{w}_0, \{u_{i,0} : i \in \mathcal{S}\}$.
2: **for** $t = 0, \ldots, T - 1$ **do**
3:     Draw sample batches $\mathcal{B}_1^t \sim \mathcal{S}$, and $\mathcal{B}_{2,i}^t \sim \mathcal{D}_i$ for each $i \in \mathcal{B}_1^t$.
4:     $u_{i,t+1} = \begin{cases} (1 - \tau)u_{i,t} + \tau g_i(\mathbf{w}_t; \mathcal{B}_{2,i}^t) + \gamma(g_i(\mathbf{w}_t; \mathcal{B}_{2,i}^t) - g_i(\mathbf{w}_{t-1}; \mathcal{B}_{2,i}^t)), & i \in \mathcal{B}_1^t \\ u_{i,t}, \quad i \notin \mathcal{B}_1^t \end{cases}$
5:     Compute $G_t = \frac{1}{B_1} \sum_{i \in \mathcal{B}_1^t} \partial g_i(\mathbf{w}_t; \mathcal{B}_{2,i}^t) \partial f_i(u_{i,t})$
6:     Update $\mathbf{w}_{t+1} = \mathbf{w}_t - \eta G_t$
7: **end for**

---

Moreover, for any point $x \in \mathbb{R}^m$, the proximal point $\hat{x} := \mathrm{prox}_{\lambda \varphi}(x)$ satisfies [6]

$$\|\hat{x} - x\| = \lambda \|\nabla \varphi_\lambda(x)\|, \quad \varphi(\hat{x}) \leq \varphi(x), \quad \mathrm{dist}(0, \partial \varphi(\hat{x})) \leq \|\nabla \varphi_\lambda(x)\|.$$

Thus if $\|\nabla \varphi_\lambda(x)\| \leq \epsilon$, we can say $x$ is close to a point $\hat{x}$ that is $\epsilon$-stationary, which is called nearly $\epsilon$-stationary solution of $\varphi(x)$.

## 4 Algorithms and Convergence

### 4.1 Non-Smooth Weakly-Convex FCCO

In this section, we assume the following conditions hold for the FCCO problem (1).

**Assumption 4.1.** For all $i \in \mathcal{S}$, we assume that

- $f_i$ is $\rho_f$-weakly-convex, $C_f$-Lipschitz continuous and non-decreasing;

- $g_i(\cdot)$ is $\rho_g$-weakly-convex and $g_i(\cdot; \xi)$ is $C_g$-Lipschitz continuous;

- Stochastic gradient estimators $g_i(\mathbf{w}; \xi)$ and $\partial g_i(\mathbf{w}; \xi)$ have bounded variance $\sigma^2$.

**Proposition 4.2.** *Under Assumption 4.1, $F(\mathbf{w})$ in (1) is $\rho_F$ weakly convex with $\rho_F = \sqrt{d_1} \rho_g C_f + \rho_f C_g^2$.*

One challenge in solving FCCO is the lack of access to unbiased estimation of the subgradients $\frac{1}{n} \sum_{i \in \mathcal{S}} \partial g_i(\mathbf{w}) \partial f_i(g_i(\mathbf{w}))$ due to the expectation form of $g_i(\mathbf{w})$ inside a non-linear function $f_i$. A common solution in existing works for solving smooth FCCO is to maintain function value estimators $\{u_i : i \in \mathcal{S}\}$ for $\{g_i(\mathbf{w}) : i \in \mathcal{S}\}$, and approximate the true gradient by a stochastic version $\frac{1}{B_1} \sum_{i \in \mathcal{B}_1} \partial g_i(\mathbf{w}; \mathcal{B}_2) \partial f_i(u_i)$ [26, 15], where $\mathcal{B}_1, \mathcal{B}_2$ are sampled mini-batches. Simply using a mini-batch estimator of $g_i$ inside $f_i$ does not ensure convergence if mini-batch size is small.

Inspired by existing algorithms of smooth FCCO, a simple method for solving non-smooth FCCO is presented in Algorithm 1 referred to as SONX. A key step is the step 4, which uses the multi-block-single-probe variance reduced (MSVR) estimator proposed in [15] to update $\{u_i : i \in \mathcal{S}\}$ in a block-wise manner. It is an advanced variance reduced update strategy for multi-block variable inspired by STORM [5]. In the update of MSVR estimator, for each sampled $i \in \mathcal{B}_1^t$, $u_{i,t}$ is updated following a STORM-like rule with a specialized parameter $\gamma = \frac{n - B_1}{B_1(1 - \tau)} + (1 - \tau)$ for the error correction term. For the unsampled $i \notin \mathcal{B}_1^t$, no update for $u_{i,t}$ is needed. When $\gamma = 0$, the estimator becomes the moving average estimator analyzed in [26] for smooth FCCO, which is also analyzed in the Appendix. With the function values of $\{g_i(\mathbf{w}_t) : i \in \mathcal{S}\}$ well-estimated, the gradient can be approximated by $G_t$ in step 5. Next, we directly update $\mathbf{w}_t$ by subgradient descent using the stochastic gradient estimator $G_t$. Note that unlike existing works on smooth FCCO that often maintain a moving average estimator [26] or a STORM estimator [15] for the overall gradient to attain better rates, this is not possible in the non-smooth case as those variance reduction techniques for the overall gradient critically rely on the Lipschitz continuity of $\nabla F$, i.e., the smoothness of $F$.

### 4.2 Non-Smooth Weakly-Convex TCCO

In this section, we consider non-smooth TCCO problem and aim to extend Algorithm 1 to solve it. First of all, for convergence analysis and to ensure the weak convexity of $F(\mathbf{w})$ in (2), we make the following assumptions.

**Assumption 4.3.** For all $(i, j) \in \mathcal{S}_1 \times \mathcal{S}_2$, we assume that

---

**Algorithm 2** Stochastic Optimization algorithm for Non-smooth TCCO (SONT)

---

1: Initialization: $\mathbf{w}_0$, $\{u_{i,0} : i \in \mathcal{S}_1\}$, $v_{i,j,0} = h_{i,j}(\mathbf{w}_0; \mathcal{B}_{3,i,j}^0)$ for all $(i,j) \in \mathcal{S}_1 \times \mathcal{S}_2$.
2: **for** $t = 0, \ldots, T-1$ **do**
3:     Sample batches $\mathcal{B}_1^t \subset \mathcal{S}_1$, $\mathcal{B}_2^t \subset \mathcal{S}_2$, and $\mathcal{B}_{3,i,j}^t \subset \mathcal{D}_{i,j}$ for $i \in \mathcal{B}_1^t$ and $j \in \mathcal{B}_2^t$.
4:     $v_{i,j,t+1} = \begin{cases} \Pi_{\tilde{C}_h}[(1-\tau_1)v_{i,j,t} + \tau_1 h_{i,j}(\mathbf{w}_t; \mathcal{B}_{3,i,j}^t) + \gamma_1(h_{i,j}(\mathbf{w}_t; \mathcal{B}_{3,i,j}^t) - h_{i,j}(\mathbf{w}_{t-1}; \mathcal{B}_{3,i,j}^t))], \\ \hspace{7cm} (i,j) \in \mathcal{B}_1^t \times \mathcal{B}_2^t \\[4pt] v_{i,j,t}, \quad (i,j) \notin \mathcal{B}_1^t \times \mathcal{B}_2^t \end{cases}$

5:     $u_{i,t+1} = \begin{cases} (1-\tau_2)u_{i,t} + \frac{1}{B_2}\sum_{j \in \mathcal{B}_2^t}[\tau_2 g_i(v_{i,j,t}) + \gamma_2(g_i(v_{i,j,t}) - g_i(v_{i,j,t-1})], & i \in \mathcal{B}_1^t \\ u_{i,t}, \quad i \notin \mathcal{B}_1^t \end{cases}$

6:     $G_t = \frac{1}{B_1}\sum_{i \in \mathcal{B}_1^t}\left[\left(\frac{1}{B_2}\sum_{i \in \mathcal{B}_2^t}\nabla h_{i,j}(\mathbf{w}_t; \mathcal{B}_{3,i,j}^t)\partial g_i(v_{i,j,t})\right)\partial f_i(u_{i,t})\right]$
7:     Update $\mathbf{w}_{t+1} = \mathbf{w}_t - \eta G_t$
8: **end for**

---

- $f_i$ is $C_f$-Lipschitz continuous, $\rho_f$-weakly-convex and non-decreasing;

- $g_i$ is $\rho_g$-weakly-convex and $C_g$-Lipschitz continuous. $h_{i,j}(\cdot; \xi)$ is $C_h$-Lipschitz continuous.

- Either $g_i$ is non-decreasing, $h_{i,j}$ is $L_h$-weakly-convex or $g_i$ is monotone, $h_{i,j}$ is $L_h$-smooth.

- Stochastic estimators $h_{i,j}(\mathbf{w}, \xi)$ and $\partial h_{i,j}(\mathbf{w}, \xi)$ have bounded variance $\sigma^2$, and $\|h_{i,j}(\mathbf{w})\| \leq \tilde{C}_h$. $\mathbb{E}_i\|g_i(v) - \frac{1}{n_2}\sum_{j \in \mathcal{S}_2} g_i(v)\|^2 \leq \sigma^2$ for any $v$.

The weak convexity of $F(\mathbf{w})$ in (2) is guaranteed by the following Proposition.

**Proposition 4.4.** *Under Assumption 4.3, $F(\mathbf{w})$ in (2) is $\rho_F$-weakly-convex with $\rho_F = \sqrt{d_1}(\sqrt{d_2}L_h C_g + \rho_g C_h^2)C_f + \rho_f C_g^2 C_h^2$.*

We extend SONX to Algorithm 2 for (2), which is referred to as SONT. For dealing with the extra layer of compositional problem, we maintain another multi-block variable to track the extra layer of function value estimation. To understand this, we first write down the true subgradient:

$$\partial F(\mathbf{w}) = \frac{1}{n_1}\sum_{i \in \mathcal{S}_1}\left[\left(\frac{1}{n_2}\sum_{j \in \mathcal{S}_2}\nabla h_{i,j}(\mathbf{w})\partial g_i(h_{i,j}(\mathbf{w}))\right)\partial f_i\left(\frac{1}{n_2}\sum_{j \in \mathcal{S}_2}g_i(h_{i,j}(\mathbf{w}))\right)\right].$$

To approximate this subgradient, we need the estimations of $\frac{1}{n_2}\sum_{j \in \mathcal{S}_2} g_i(h_{i,j}(\mathbf{w}))$ and $h_{i,j}(\mathbf{w})$, which can be tracked by using MSVR estimators denoted by $\{u_{i,t} : i \in \mathcal{S}_1\}$ and $\{v_{i,j,t} : (i,j) \in \mathcal{S}_1 \times \mathcal{S}_2\}$, respectively. As a result, a stochastic estimation of $\partial F(\mathbf{w}_t)$ is computed in step 6 of Algorithm 2, and the model parameter is updated similarly as before.

### 4.3 Convergence Analysis

In this section, we present the proof sketch of the convergence guarantee for Algorithm 1. The analysis for Algorithm 2 follows in a similar manner. The detailed proofs can be found in Appendix A (please refer to the supplement). Before starting the proof, we define a constant $M^2 \geq C_f^2 C_g^2$ so that under Assumption 4.1 we have $\mathbb{E}_t[\|G_t\|^2] \leq M^2$. Then we start by giving the error bound of the MSVR estimator in Algorithm 1. The following norm bound of the estimation error follows from the squared-norm error bound in Lemma 1 from [15], whose proof is given in Appendix D.3.

**Lemma 4.5.** *Consider the update for $\{u_{i,t} : i \in \mathcal{S}\}$ in Algorithm 1. Assume $g_i$ is $C_g$-Lipschitz for all $i \in \mathcal{S}$. With $\gamma = \frac{n-B_1}{B_1(1-\tau)} + (1-\tau)$, $\tau \leq \frac{1}{2}$, we have*

$$\mathbb{E}\left[\frac{1}{n}\sum_{i \in \mathcal{S}}\|u_{i,t+1} - g_i(\mathbf{w}_{t+1})\|\right] \leq (1 - \frac{B_1\tau}{2n})^{t+1}\frac{1}{n}\sum_{i \in \mathcal{S}}\|u_{i,0} - g_i(\mathbf{w}_0)\| + \frac{2\tau^{1/2}\sigma}{B_2^{1/2}} + \frac{4nC_g M\eta}{B_1\tau^{1/2}}.$$

For simplicity, denote by $\hat{\mathbf{w}}_t := \text{prox}_{F/\bar{\rho}}(\mathbf{w}_t)$. Then using the definition of Moreau envelope and the update rule of $\mathbf{w}_t$, we can obtain a bound for the change in the Moreau envelope,

$$\mathbb{E}_t[F_{1/\bar{\rho}}(\mathbf{w}_{t+1})] \leq F_{1/\bar{\rho}}(\mathbf{w}_t) + \bar{\rho}\eta\langle \hat{\mathbf{w}}_t - \mathbf{w}_t, \mathbb{E}_t[G_t]\rangle + \frac{\eta^2 \bar{\rho} M^2}{2}. \tag{5}$$

where $\mathbb{E}_t[G_t] = \frac{1}{n}\sum_{i \in S_1}\partial g_i(\mathbf{w}_t)\partial f_i(u_{i,t})$ is the subgradient approximation based on the MSVR estimator $u_{i,t}$ of the inner function value. This is a standard result in weakly-convex optimization [6].

To bound the inner product $\langle \hat{\mathbf{w}}_t - \mathbf{w}_t, \mathbb{E}_t[G_t] \rangle$ on the right-hand-side of (5), we apply the assumptions that $f_i$ is weakly-convex, Lipschitz continuous and non-decreasing, and $g_i$ is weakly-convex. Its upper bound is given as follows.

$$(\hat{\mathbf{w}}_t - \mathbf{w}_t)^\top \mathbb{E}_t[G_t] \le F(\hat{\mathbf{w}}_t) - F(\mathbf{w}_t) + \frac{1}{n}\sum_{i \in \mathcal{S}}[f_i(g_i(\mathbf{w}_t)) - f(u_{i,t}) - \partial f(u_{i,t})^\top(g_i(\mathbf{w}_t) - u_{i,t})$$

$$+ \rho_f \|g_i(\mathbf{w}_t) - u_{i,t}\|^2 + (\frac{\rho_g C_f}{2} + \rho_f C_g^2)\|\hat{\mathbf{w}}_t - \mathbf{w}_t\|^2]. \tag{6}$$

Due to the $\rho_F$-weak convexity of $F(\mathbf{w})$, we have $(\bar{\rho} - \rho_F)$-strong convexity of $\mathbf{w} \mapsto F(\mathbf{w}) + \frac{\bar{\rho}}{2}\|\mathbf{w}_t - \mathbf{w}\|^2$. Then it follows $F(\hat{\mathbf{w}}_t) - F(\mathbf{w}_t) \le (\frac{\rho_F}{2} - \bar{\rho})\|\mathbf{w}_t - \hat{\mathbf{w}}_t\|^2$. Combining this with inequalities (5), (6), and setting $\bar{\rho}$ sufficiently large we have

$$\mathbb{E}_t[F_{1/\bar{\rho}}(\mathbf{w}_{t+1})] \le F_{1/\bar{\rho}}(\mathbf{w}_t) + \frac{\eta^2 \bar{\rho} M^2}{2} + \frac{\bar{\rho}\eta}{n}\sum_{i \in \mathcal{S}}[-\frac{\bar{\rho}}{2}\|\mathbf{w}_t - \hat{\mathbf{w}}_t\|^2$$

$$+ f_i(g_i(\mathbf{w}_t)) - f(u_{i,t}) - \partial f_i(u_{i,t})^\top(g_i(\mathbf{w}_t) - u_{i,t}) + \rho_f\|g_i(\mathbf{w}_t) - u_{i,t}\|^2]. \tag{7}$$

Recall Lemma 3.2, we have $\|\mathbf{w}_t - \hat{\mathbf{w}}_t\|^2 = \frac{1}{\bar{\rho}^2}\|\nabla F_{1/\bar{\rho}}(\mathbf{w}_t)\|^2$. Moreover, the last three terms on the R.H.S of inequality (7) can be bounded using the Lipschitz continuity of $f_i$ and the error bound given in Lemma 4.5. Then we can conclude the complexity of SONX with the following theorem.

**Theorem 4.6.** *Under Assumption 4.1 with* $\gamma = \frac{n - B_1}{B_1(1-\tau)} + (1 - \tau)$, $\tau = \mathcal{O}(B_2\epsilon^4) \le \frac{1}{2}$, $\eta = \mathcal{O}(\frac{B_1 B_2^{1/2}\epsilon^4}{n})$, *and* $\bar{\rho} = \rho_F + \rho_g C_f + 2\rho_f C_g^2$, *Algorithm 1 converges to an $\epsilon$-stationary point of the Moreau envelope* $F_{1/\bar{\rho}}$ *in* $T = \mathcal{O}(\frac{n}{B_1 B_2^{1/2}}\epsilon^{-6})$ *iterations.*

**Remark.** Similar to the complexity for smooth FCCO problems [26, 15], Theorem 4.6 guarantees that SONX for NSWC FCCO has a parallel speed-up in terms of the batch size $B_1$ and linear dependency on $n$. The dependency of the complexity on the batch size $B_2$ is due to the use of MSVR estimator, which matches the results in [15]. If the MSVR estimator in SONX is replaced by moving average estimator, the complexity becomes $\mathcal{O}(\frac{n}{B_1 B_2}\epsilon^{-8})$ (cf. Appendix B).

Following a similar proof strategy, the convergence guarantee of Algorithm 2 is given below.

**Theorem 4.7.** *(Informal) Under Assumption 4.3, with appropriate values of* $\gamma_1, \gamma_2, \tau_1, \tau_2, \eta$ *and a proper constant* $\bar{\rho}$, *Algorithm 2 converges to an $\epsilon$-stationary point of the Moreau envelope* $F_{1/\bar{\rho}}$ *in*

$$T = \mathcal{O}\left(\max\left\{\frac{1}{B_3^{1/2}}, \frac{n_1^{1/4}}{B_1^{1/4}n_2^{1/4}}, \frac{n_1^{1/2}}{B_1^{1/2}n_2^{1/2}}\right\}\frac{n_1 n_2}{B_1 B_2}\epsilon^{-6}\right) \text{ iterations.}$$

**Remark.** In the worst case, the complexity has a worse dependency on $n_1/B_1$, i.e., $\mathcal{O}(n_1^{3/2}/B_1^{3/2})$. This is caused by the two layers of block-sampling update for $\{u_{i,t}, i \in \mathcal{S}_1\}$ and $\{v_{i,j,t} : (i,j) \in \mathcal{S}_1 \times \mathcal{S}_2\}$. When $n_1 = B_1 = 1$ and $B_3 \le \sqrt{n_2}$, the complexity of SONT becomes similar as SONX, which is understandable as the inner two levels in TCCO is the same as FCCO.

## 5 Applications

NSWC FCCO finds important applications in group distributionally robust optimization (group DRO) and two-way partial AUC (TPAUC) maximization.

Consider $N$ groups with different distributions. Each group $k$ has an averaged loss $L_k(w) = \frac{1}{n_k}\sum_{i=1}^{n_k}\ell(f_w(x_i^k), y_i^k)$, where $w$ is the the model parameter and $(x_i^k, y_i^k)$ is a data point. It has been shown in previous study [23] that the group DRO problem can be formulated into

$$\min_w \min_s F(w, s) = \frac{1}{K}\sum_{k=1}^{N}[L_k(w) - s]_+ + s.$$

This formulation can be mapped into non-smooth weakly-convex FCCO under certain assumptions. Due to space limitation, we defer the comprehensive discussion of group DRO to Appendix E. The rest of this section focuses on TPAUC maximization.

Let $X$ denote an input example and $h_{\mathbf{w}}(X)$ denote a prediction of a parameterized deep net on data $X$. Denote by $\mathcal{S}_+$ the set of $n_+$ positive examples and by $\mathcal{S}_-$ the set of $n_-$ negative examples. TPAUC measures the area under ROC curve where the true positive rate (TPR) is higher than $\alpha$ and the false positive rate (FPR) is lower than an upper bound $\beta$. A surrogate loss for optimizing TPAUC

with FPR$\leq \beta$, TPR$\geq \alpha$ is given by [34]:

$$\min_{\mathbf{w}} \frac{1}{n_+} \frac{1}{n_-} \sum_{X_i \in \mathcal{S}_+^\uparrow[1,k_1]} \sum_{X_j \in \mathcal{S}_-^\downarrow[1,k_2]} \ell(h_{\mathbf{w}}(X_j) - h_{\mathbf{w}}(X_i)), \tag{8}$$

where $\ell(\cdot)$ is a convex, monotonically non-decreasing surrogate loss of the indicator function $\mathbb{I}(h_{\mathbf{w}}(X_j) \geq h_{\mathbf{w}}(X_i))$, $\mathcal{S}_+^\uparrow[1,k_1]$ is the set of positive examples with $k_1 = \lfloor n_+\alpha \rfloor$ smallest scores, and $\mathcal{S}_-^\downarrow[1,k_2]$ is the set of negative examples with $k_2 = \lfloor n_-\beta \rfloor$ largest scores. To tackle the challenge of selecting examples from $\mathcal{S}_+^\uparrow[1,k_1]$ and $\mathcal{S}_-^\downarrow[1,k_2]$, the above problem is cast into the following [44]:

$$\min_{\mathbf{w},s',\mathbf{s}} \frac{1}{n_+} \sum_{X_i \in \mathcal{S}_+} f_i(\psi_i(\mathbf{w},s_i), s'), \tag{9}$$

where $f_i(g,s') = s' + \frac{(g-s')_+}{\alpha}$, $\psi_i(\mathbf{w},s_i) = \frac{1}{n_-}\sum_{X_j \in \mathcal{S}_-} s_i + \frac{(\ell(h_{\mathbf{w}}(X_j) - h_{\mathbf{w}}(X_i)) - s_i)_+}{\beta}$,

where $\mathbf{s} = (s_1,\ldots,s_{n_+})$. We will consider two scenarios, namely regular learning scenario where $X_i \in \mathbb{R}^{d_0}$ is an instance, and multi-instance learning (MIL) scenario where $X_i = \{\mathbf{x}_i^1,\ldots,\mathbf{x}_i^{m_i} \in \mathbb{R}^{d_0}\}$ contains multiple instances (e.g., one patient has hundreds of high-resolution CT images). A challenge in MIL is that the number of instances $m_i$ for each data might be large such that it is difficult to load all instances into the memory for mini-batch training. It becomes more nuanced especially because MIL involves a pooling operation that aggregates the predicted information of individual instances into a single prediction, which can be usually written as a compositional function with the inner function being an average over instances from $X$. For simplicity of exposition, below we consider the mean pooling $h_{\mathbf{w}}(X) = \frac{1}{|X|} \sum_{\mathbf{x} \in X} e(\mathbf{w}_e; \mathbf{x})^\top \mathbf{w}_c$, where $e(\mathbf{w}_e, \mathbf{x})$ is the encoded feature representation of instance $\mathbf{x}$ with a parameter $\mathbf{w}_e$, and $\mathbf{w}_c$ is the parameter of the classifier. We will map the regular learning problem as NSWC FCCO and the MIL problem as NSWC TCCO.

The problem (9) is slightly more complicated than (1) or (2) due to the presence of $s', \mathbf{s}$. In order to understand the applicability of our analysis and results to (9), we ignore $s', \mathbf{s}$ for a moment. In the regular learning setting when $h_{\mathbf{w}}(X) = e(\mathbf{w}_e, X)^\top \mathbf{w}_c$ can be directly computed, we can map the problem into NSWC FCCO, where $f_i(g,s')$ is non-smooth, convex, and non-decreasing in terms of $g$, and $g_i(\mathbf{w},s_i) = \psi_i(\mathbf{w},s_i)$ is non-smooth, and is proved to be weakly when $\ell(\cdot)$ is convex and $h_{\mathbf{w}}(X)$ is smooth in terms of $\mathbf{w}$. In the MIL setting with mean pooling, we can map the problem into NSWC TCCO by defining $h_i(\mathbf{w}) = \frac{1}{|X_i|}\sum_{\mathbf{x} \in X_i} e(\mathbf{w}_e; \mathbf{x})^\top \mathbf{w}_c$, $h_{ij}(\mathbf{w}) = h_j(\mathbf{w}) - h_i(\mathbf{w})$ and $g_i(h_{i,j}(\mathbf{w}), s_i) = s_i + \frac{(\ell(h_{i,j}(\mathbf{w})) - s_i)_+}{\beta}$, and $f_i(g_i, s') = s' + \frac{(g_i - s')_+}{\alpha}$, where $f_i$ is non-smooth, convex, and non-decreasing in terms of $g_i$, and $g_i(h_{ij}(\mathbf{w}), s_i)$ is non-smooth, convex, monotonic in terms of $h_{ij}(\mathbf{w})$ when $\ell(\cdot)$ is convex and monotonically non-decreasing, and $g_i(h_{ij}(\mathbf{w}), s_i)$ is weakly convex in terms of $\mathbf{w}$ when $h_{ij}(\mathbf{w})$ is smooth and Lipchitz continuous in terms of $\mathbf{w}$. Hence, the problem (9) satisfies the conditions in Assumption 4.1 for the regular learning setting and that in Assumption 4.3 for the MIL with mean pooling under mild regularity conditions of the neural network. We present full details in Appendix C.1 for interested readers.

To compute the gradient estimator w.r.t $\mathbf{w}$, $u_{i,t}$ will be maintained for tracking $g_i(\mathbf{w},s_i)$ in the regular setting or $\frac{1}{n_-}\sum_{X_j \in \mathcal{S}_-} g_i(h_{i,j}(\mathbf{w}), s_i)$ in the MIL setting, $v_{i,t}$ will be maintained for tracking $h_i(\mathbf{w})$ in the MIL setting, which are updated similar to that in SONX and SONT. One difference from SONT is that $v_{i,j,t}$ is decoupled into $v_{i,t}$ and $v_{j,t}$ due to that $h_{i,j}$ can be decoupled. In terms of the extra variable $s', \mathbf{s}$, the objective function is convex w.r.t both $s'$ and $\mathbf{s}$, which allows us to simply update $s'$ by SGD using the stochastic gradient estimator $\frac{1}{B_1}\sum_{i \in \mathcal{B}_1^t} \partial_{s'} f_i(u_{i,t}, s_t')$ and we update $s_i$ by SGD using the stochastic gradient estimator $\left[\frac{1}{B_2}\sum_{j \in \mathcal{B}_2^t} \partial_{s_i} g_i(v_{j,t} - v_{i,t}, s_{i,t})\right] \partial_u f_i(u_{i,t}, s_t')$. Detailed updates are presented in Algorithm 5 and Algorithm 6 in Appendix C.2. We can extend the convergence analysis of SONX and SONT to the two learning settings of TPAUC maximization, which is included in Appendix C.4. Finally, it is worth mentioning that we can also extend the results to other pooling operations, including smoothed max pooling and attention-based pooling [45]. Due to limit of space, we include discussions in Appendix C.3 as well.

## 6 Experimental Results

We justify the effectiveness of the proposed SONX and SONT algorithms for TPAUC Maximization in the regular learning setting and MIL setting [14, 45].

Table 2: Testing TPAUC on molecule datasets (top) and on MIL datasets (bottom). The two numbers in parentheses of the second line refers to the lower bound of TPR and the upper bound of FPR for evaluating TPAUC. The two numbers of each method refers to the mean TPAUC and its std.

| Method | moltox21 (t0) | | molmuv (t1) | | molpcba (t0) | |
|---|---|---|---|---|---|---|
| | (0.6, 0.4) | (0.5, 0.5) | (0.6, 0.4) | (0.5, 0.5) | (0.6, 0.4) | (0.5, 0.5) |
| CE | 0.067 (0.001) | 0.208 (0.001) | 0.161 (0.034) | 0.469 (0.018) | 0.095 (0.001) | 0.264 (0.001) |
| AUC-SH | 0.064 (0.008) | 0.217 (0.014) | 0.260 (0.130) | 0.444 (0.128) | 0.140 (0.003) | 0.312 (0.003) |
| AUC-M | 0.066 (0.009) | 0.209 (0.01) | 0.114 (0.079) | 0.433 (0.053) | 0.142 (0.009) | 0.313 (0.003) |
| MB | 0.067 (0.015) | 0.215 (0.023) | 0.173 (0.153) | 0.426 (0.118) | 0.095 (0.002) | 0.262 (0.003) |
| AW-poly | 0.064 (0.01) | 0.206 (0.025) | 0.172 (0.144) | 0.393 (0.123) | 0.110 (0.001) | 0.281 (0.002) |
| SOTA-s | 0.068 (0.018) | 0.23 (0.021) | 0.327 (0.164) | 0.526 (0.122) | 0.143 (0.001) | 0.314 (0.002) |
| SONX | **0.07 (0.035)** | **0.252 (0.025)** | **0.347 (0.175)** | **0.575 (0.122)** | **0.158 (0.006)** | **0.335 (0.006)** |

| Method | MUSK2 | | | Fox | | |
|---|---|---|---|---|---|---|
| | (0.5, 0.5) | (0.3, 0.7) | (0.1, 0.9) | (0.5, 0.5) | (0.3, 0.7) | (0.1, 0.9) |
| AUC-M (att) | 0.675 (0.1) | 0.783 (0.067) | **0.867 (0.036)** | 0.032 (0.03) | 0.253 (0.098) | 0.444 (0.118) |
| MIDAM (smx) | 0.525 (0.2) | 0.667 (0.149) | 0.8 (0.097) | 0.048 (0.059) | 0.265 (0.119) | 0.449 (0.113) |
| MIDAM (att) | 0.6 (0.215) | 0.717 (0.135) | 0.819 (0.092) | 0.016 (0.032) | 0.249 (0.125) | 0.509 (0.065) |
| SOTAs (att) | 0.6 (0.267) | 0.683 (0.178) | 0.819 (0.097) | 0.024 (0.032) | 0.278 (0.059) | 0.477 (0.046) |
| SONT (att) | **0.7 (0.1)** | **0.8 (0.067)** | **0.867 (0.036)** | **0.12 (0.131)** | **0.343 (0.176)** | **0.578 (0.119)** |

| Method | Colon | | | Lung | | |
|---|---|---|---|---|---|---|
| | (0.5, 0.5) | (0.3, 0.7) | (0.1, 0.9) | (0.5, 0.5) | (0.3, 0.7) | (0.1, 0.9) |
| AUC-M (att) | 0.576 (0.1) | 0.739 (0.061) | 0.803 (0.038) | 0.32 (0.181) | 0.609 (0.113) | 0.744 (0.082) |
| MIDAM (smx) | 0.646 (0.083) | 0.787 (0.04) | 0.863 (0.026) | 0.43 (0.195) | 0.68 (0.128) | 0.824 (0.055) |
| MIDAM (att) | 0.548 (0.253) | 0.738 (0.149) | 0.826 (0.102) | 0.544 (0.261) | 0.716 (0.189) | 0.815 (0.129) |
| SOTAs (att) | 0.772 (0.124) | 0.862 (0.073) | 0.911 (0.045) | 0.539 (0.153) | 0.745 (0.077) | 0.841 (0.049) |
| SONT (att) | **0.8 (0.166)** | **0.875 (0.099)** | **0.916 (0.065)** | **0.639 (0.137)** | **0.779 (0.041)** | **0.865 (0.028)** |

**Baselines.** For *regular TPAUC maximization*, we compare SONX with the following competitive methods: 1) Cross Entropy (CE) loss minimization; 2) AUC maximization with squared hinge loss (AUC-SH); 3) AUC maximization with min-max margin loss (AUC-M) [37]; 4) Mini-Batch based heuristic loss (MB) [16]; 5) Adhoc-Weighting based method with polynomial function (AW-poly) [35]; 5) a single-loop algorithm (SOTAs) for optimizing a smooth surrogate for TPAUC [44]. For *MIL TPAUC maximization*, we consider the following baselines: 1) AUC-M with attention-based pooling (AUC-M [att]); 2) SOTAs with attention-based pooling, which is a natural combination between advanced TPAUC optimization and MIL pooling technique; 3) the recently proposed provable multi-instance deep AUC maximization methods with stochastic smoothed-max pooling and attention-based pooling (MIDAM [smx] and MIDAM [att]) [45]. The first two baselines use naive mini-batch pooling for computing the loss function in AUC-M and SOTAs. We implement SONT for MIL TPAUC maximization with attention-based pooling, which is referred to as SONT (att).

**Datasets.** For regular TPAUC maximization, we use three molecule datasets as in [44], namely moltox21 (the No.0 target), molmuv (the No.1 target) and molpcba (the No.0 target) [29]. For MIL TPAUC maximization, we use four MIL datasets, including two tabular datasets MUSK2 and Fox, and two medical image datasets Colon and Lung. MUSK2 and Fox are two tabular datasets that have been widely adopted for MIL benchmark study [14]. Colon and Lung are two histopathology (medical image) datasets that have large image size ($512 \times 512$) but local interests for classification [2]. For Colon dataset, the adenocarcinoma is regarded as positive label and benign is negative; for Lung dataset, we treat adenocarcinoma as positive and squamous cell carcinoma as negative [3]. For both of the histopathology datasets, we uniformly randomly sample 100 positive and 1000 negative data for experiments. For all MIL datasets, we uniformly randomly split 10% as the testing and the remaining as the training and validation. The statistics for all used datasets are summarized in Table 3 and Table 4 in Appendix F.

**Experiment Settings.** For regular TPAUC maximization, we use the same setting as in [44]. The adopted backbone Graph Nueral Network (GNN) model is Graph Isomorphism Network (GIN), which has 5 mean-pooling layers with 64 number of hidden units and dropout rate 0.5 [30]. We utilize the sigmoid function for the final output layer to generate the prediction score, and set the surrogate loss $\ell(\cdot)$ as squared hinge loss with a margin parameter. We follow the setups for model training and tuning exactly the same as the prior work [44]. Essentially, the model is trained by 60 epochs and the learning rate is decreased by 10-fold after every 20 epochs. The model is initialized as a pretrained model from CE loss on the training datasets. We fix the learning rate of SONX as 1e-2 and moving average parameter $\tau$ as 0.9; tune the parameter $\gamma$ in {0, 1e-1,1e-2,1e-3}, the parameter $\alpha, \beta$ in {0.1,0.3,0.5} and fix the margin parameter of the surrogate loss $\ell$ as 1.0, which cost the same

---

[3]Data available: `https://www.kaggle.com/datasets/biplobdey/lung-and-colon-cancer`

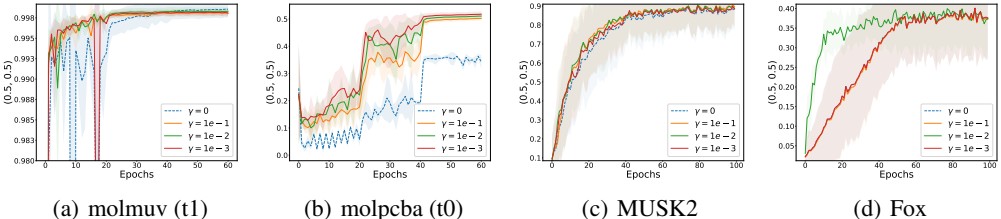

|  (a) molmuv (t1) | (b) molpcba (t0) | (c) MUSK2 | (d) Fox |

Figure 1: Training Curves of SONX (left two) and SONT (right two) for TPAUC maximization with different $\gamma$. The y-axis is the TPAUC (0.5, 0.5).

tuning effort as the other baselines. The weight decay is set as the same value (2e-4) with the other baselines. For baselines, we directly use the results reported in [44] since we use the same setting.

For MIL TPAUC maximization, we train a simple Feed Forward Neural Network (FFNN) with one hidden layer (the number of neurons equals to data dimension) for the two tabular datasets and ResNet20 for the two medical image datasets. Sigmoid transformation is adopted for the output layer to generate prediction score. The training epoch number is fixed as 100 epochs for all methods; the bag batch size is fixed as 16 (resp. 8) and the number of sampled instances per bag is fixed as 4 (resp. 128) for tabular (resp. medical image) datasets; the learning rate is tuned in {1e-2, 1e-3, 1e-4} and decreased by 10 folds at the end of 50-th and 75-th epoch for all baselines. For SONT (att), we set moving average parameter $\tau_1 = \tau_2$ as 0.9; tune the parameter $\gamma_1 = \gamma_2 = \gamma$ in {0, 1e-1,1e-2,1e-3} and fix the margin parameter of the surrogate loss $\ell$ as 0.5, and the parameter $\alpha, \beta$ in {0.1,0.5,0.9}. Similar parameters in baselines are set the same or tuned similarly. For all experiments, we utilize 5-fold-cross-validation to evaluate the testing performance based on the best validation performance with possible early stopping choice.

**Results.** The testing results for the regular and MIL TPUAC maximization with different TPAUC measures are summarized in the Table 2. From Table 2, we observe that our method SONX achieves the best performance for regular TPAUC maximization. It is better than the state-of-the-art method SOTAs for TPAUC maximization. We attribute the better performance of SONX to the fact that the objective of SONX is an exact estimator of TPAUC while the smoothed objective of SOTAs is an inexact estimator of TPAUC. We also observe that SONT (att) achieves the best performance in all cases, which is not surprising since it is the only one that directly optimizes the TPAUC surrogate. In contrast, other baselines either optimizes a different objective (MIDAM) or does not ensure convergence due to the use of mini-batch pooling (AUC-M, SOTAs).

**Ablation Study.** We conduct ablation studies to demonstrate the effect of the error correction term on the training convergence by varying the $\gamma$ value for SONX and SONT, where $\gamma_1 = \gamma_2 = \gamma$ is set as the same value in SONT. The training convergence results are presented in Figure 1. We can see that an appropriate value of $\gamma > 0$ can yield a faster convergence than $\gamma = 0$, which verifies the faster convergence of using MSVR estimators than using moving average estimators. However, we do observe a gap between theory and practice, as setting a large value of $\gamma > 1$ as in the theory might not yield convergence. This phenomenon is also observed in [12]. We conjecture that the gap could be fixed by considering convex objectives [40], which is left as future work.

## 7    Conclusions

In this paper, we have considered non-smooth weakly-convex two-level and tri-level finite-sum coupled compositional optimization problems. We presented novel convergence analysis of two stochastic algorithms and established their complexity. Applications in deep learning for two-way partial AUC maximization was considered and great performance of proposed algorithms were demonstrated through experiments on multiple datasets. A future work is to prove the convergence of both algorithms for convex objectives.

## Acknowledgements

We thank anonymous reviewers for constructive comments. Q. Hu, D. Zhu and T. Yang were partially supported by NSF Career Award 2246753, NSF Grant 2246757, 2246756 and 2306572.

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

# A  Proofs of Theorem 4.6 and Theorem 4.7

In this section, we provide the detailed proofs for Theorem 4.6 and Theorem 4.7. We first give a basic property for weakly-convex functions.

**Proposition A.1** (Proposition 2.1 in [7]). *Suppose function $g : \mathbb{R}^d \to \mathbb{R} \cup \{\infty\}$ is lower-semicontinuous. Then $g$ is $\rho$-weakly-convex if and only if*

$$g(y) \geq g(x) + \langle v, y - x \rangle - \frac{\rho}{2}\|y - x\|^2 \tag{10}$$

*holds for all vectors $v \in \partial g(x)$ and $x, y \in \mathbb{R}^d$.*

## A.1  Proof of Theorem 4.6

Note that the proof of Lemma 4.5 also implies the following squared-norm error bound,

$$\mathbb{E}\left[\frac{1}{n}\sum_{i \in \mathcal{S}}\|u_{i,t+1} - g_i(\mathbf{w}_{t+1})\|^2\right] \leq (1 - \frac{B_1\tau}{2n})^{t+1}\frac{1}{n}\sum_{i \in \mathcal{S}}\|u_{i,0} - g_i(\mathbf{w}_0)\|^2 + \frac{4\tau\sigma^2}{B_2} + \frac{16n^2C_g^2M^2\eta^2}{B_1^2\tau}.$$

*Proof of Theorem 4.6.* Define $\hat{\mathbf{w}}_t := \text{prox}_{F/\bar{\rho}}(\mathbf{w}_t)$. For a given $i \in \mathcal{S}$, we have

$f_i(g_i(\hat{\mathbf{w}}_t)) - f_i(u_{i,t})$

$\overset{(a)}{\geq} \partial f_i(u_{i,t})^\top(g_i(\hat{\mathbf{w}}_t) - u_{i,t}) - \frac{\rho_f}{2}\|g_i(\hat{\mathbf{w}}_t) - u_{i,t}\|^2$

$\geq \partial f_i(u_{i,t})^\top(g_i(\hat{\mathbf{w}}_t) - u_{i,t}) - \rho_f\|g_i(\hat{\mathbf{w}}_t) - g_i(\mathbf{w}_t)\|^2 - \rho_f\|g_i(\mathbf{w}_t) - u_{i,t}\|^2$

$\geq \partial f_i(u_{i,t})^\top(g_i(\hat{\mathbf{w}}_t) - u_{i,t}) - \rho_f C_g^2\|\hat{\mathbf{w}}_t - \mathbf{w}_t\|^2 - \rho_f\|g_i(\mathbf{w}_t) - u_{i,t}\|^2$

$\overset{(b)}{\geq} \partial f_i(u_{i,t})^\top\left[g_i(\mathbf{w}_t) - u_{i,t} + \partial g_i(\mathbf{w}_t)^\top(\hat{\mathbf{w}}_t - \mathbf{w}_t) - \frac{\rho_g}{2}\|\hat{\mathbf{w}}_t - \mathbf{w}_t\|^2\right]$

$\quad - \rho_f C_g^2\|\hat{\mathbf{w}}_t - \mathbf{w}_t\|^2 - \rho_f\|g_i(\mathbf{w}_t) - u_{i,t}\|^2$

$\overset{(c)}{\geq} \partial f_i(u_{i,t})^\top(g_i(\mathbf{w}_t) - u_{i,t}) + \partial f_i(u_{i,t})^\top\partial g_i(\mathbf{w}_t)^\top(\hat{\mathbf{w}}_t - \mathbf{w}_t) - (\frac{\rho_g C_f}{2} + \rho_f C_g^2)\|\hat{\mathbf{w}}_t - \mathbf{w}_t\|^2$

$\quad - \rho_f\|g_i(\mathbf{w}_t) - u_{i,t}\|^2$

where (a) follows from the $\rho_f$-weak-convexity of $f_i$, (b) follows from that $f_i(\cdot)$ is non-decreasing and the weak convexity of $g_i$, (c) is due to $0 \leq \partial f_i(u_{i,t}) \leq C_f$. Then it follows

$\frac{1}{n}\sum_{i \in \mathcal{S}}\partial f_i(u_{i,t})^\top\partial g_i(\mathbf{w}_t)^\top(\hat{\mathbf{w}}_t - \mathbf{w}_t)$

$\leq \frac{1}{n}\sum_{i \in \mathcal{S}}\left[f_i(g_i(\hat{\mathbf{w}}_t)) - f_i(u_{i,t}) - \partial f_i(u_{i,t})^\top(g_i(\mathbf{w}_t) - u_{i,t}) + (\frac{\rho_g C_f}{2} + \rho_f C_g^2)\|\hat{\mathbf{w}}_t - \mathbf{w}_t\|^2\right.$

$\quad \left. + \rho_f\|g_i(\mathbf{w}_t) - u_{i,t}\|^2\right]$

$$\tag{11}$$

Now we consider the change in the Moreau envelope:

$$\mathbb{E}_t[F_{1/\bar{\rho}}(\mathbf{w}_{t+1})] = \mathbb{E}_t\left[\min_{\tilde{\mathbf{w}}} F(\tilde{\mathbf{w}}) + \frac{\bar{\rho}}{2}\|\tilde{\mathbf{w}} - \mathbf{w}_{t+1}\|^2\right]$$

$$\leq \mathbb{E}_t\left[F(\hat{\mathbf{w}}_t) + \frac{\bar{\rho}}{2}\|\hat{\mathbf{w}}_t - \mathbf{w}_{t+1}\|^2\right]$$

$$= F(\hat{\mathbf{w}}_t) + \mathbb{E}_t\left[\frac{\bar{\rho}}{2}\|\hat{\mathbf{w}}_t - (\mathbf{w}_t - \eta G_t)\|^2\right] \tag{12}$$

$$\leq F(\hat{\mathbf{w}}_t) + \frac{\bar{\rho}}{2}\|\hat{\mathbf{w}}_t - \mathbf{w}_t\|^2 + \bar{\rho}\mathbb{E}_t[\eta\langle\hat{\mathbf{w}}_t - \mathbf{w}_t, G_t\rangle] + \frac{\eta^2\bar{\rho}M^2}{2}$$

$$= F_{1/\bar{\rho}}(\mathbf{w}_t) + \bar{\rho}\eta\langle\hat{\mathbf{w}}_t - \mathbf{w}_t, \mathbb{E}_t[G_t]\rangle + \frac{\eta^2\bar{\rho}M^2}{2}$$

where

$$\mathbb{E}_t[G_t] = \frac{1}{n}\sum_{i \in \mathcal{S}} \partial g_i(\mathbf{w}_t)\partial f_i(u_{i,t}),$$

and the second inequality uses the bound of $\mathbb{E}[\|G_t\|^2]$, which follows from the Lipschitz continuity and bounded variance assumptions and is denoted by $M$.

Combining inequality 29 and 30 yields

$$
\begin{aligned}
&\mathbb{E}_t[F_{1/\bar{\rho}}(\mathbf{w}_{t+1})] \\
&\leq F_{1/\bar{\rho}}(\mathbf{w}_t) + \frac{\eta^2\bar{\rho}M^2}{2} + \frac{\bar{\rho}\eta}{n}\sum_{i \in \mathcal{S}}\bigg[ f_i(g_i(\hat{\mathbf{w}}_t)) - f_i(u_{i,t}) \\
&\quad - \partial f_i(u_{i,t})^\top(g_i(\mathbf{w}_t) - u_{i,t}) + (\frac{\rho_g C_f}{2} + \rho_f C_g^2)\|\hat{\mathbf{w}}_t - \mathbf{w}_t\|^2 + \rho_f\|g_i(\mathbf{w}_t) - u_{i,t}\|^2 \bigg] \\
&= F_{1/\bar{\rho}}(\mathbf{w}_t) + \frac{\eta^2\bar{\rho}M^2}{2} + \frac{\bar{\rho}\eta}{n}\sum_{i \in \mathcal{S}}\bigg[ F_i(\hat{\mathbf{w}}_t) - F_i(\mathbf{w}_t) + f_i(g_i(\mathbf{w}_t)) - f_i(u_{i,t}) \\
&\quad - \partial f_i(u_{i,t})^\top(g_i(\mathbf{w}_t) - u_{i,t}) + (\frac{\rho_g C_f}{2} + \rho_f C_g^2)\|\hat{\mathbf{w}}_t - \mathbf{w}_t\|^2 + \rho_f\|g_i(\mathbf{w}_t) - u_{i,t}\|^2 \bigg]
\end{aligned}
\tag{13}
$$

Due to the $\rho_F$-weak convexity of $F_i(\mathbf{w})$, we have $(\bar{\rho} - \rho_F)$-strong convexity of $\mathbf{w} \mapsto F_i(\mathbf{w}) + \frac{\bar{\rho}}{2}\|\mathbf{w}_t - \mathbf{w}\|^2$. Then it follows

$$
\begin{aligned}
F_i(\hat{\mathbf{w}}_t) - F_i(\mathbf{w}_t) &= \bigg[F_i(\hat{\mathbf{w}}_t) + \frac{\bar{\rho}}{2}\|\mathbf{w}_t - \hat{\mathbf{w}}_t\|^2\bigg] - \bigg[F_i(\mathbf{w}_t) + \frac{\bar{\rho}}{2}\|\mathbf{w}_t - \mathbf{w}_t\|^2\bigg] - \frac{\bar{\rho}}{2}\|\mathbf{w}_t - \hat{\mathbf{w}}_t\|^2 \\
&\leq (\frac{\rho_F}{2} - \bar{\rho})\|\mathbf{w}_t - \hat{\mathbf{w}}_t\|^2
\end{aligned}
\tag{14}
$$

Plugging inequality 32 into inequality 31 yields

$$
\begin{aligned}
\mathbb{E}_t[F_{1/\bar{\rho}}(\mathbf{w}_{t+1})] \leq \ &\mathbb{E}[F_{1/\bar{\rho}}(\mathbf{w}_t)] + \frac{\eta^2\bar{\rho}M^2}{2} + \frac{\bar{\rho}\eta}{n}\sum_{i \in \mathcal{S}}\bigg[ (\frac{\rho_F}{2} - \bar{\rho})\|\mathbf{w}_t - \hat{\mathbf{w}}_t\|^2 \\
&+ f_i(g_i(\mathbf{w}_t)) - f_i(u_{i,t}) - \partial f_i(u_{i,t})^\top(g_i(\mathbf{w}_t) - u_{i,t}) \\
&+ (\frac{\rho_g C_f}{2} + \rho_f C_g^2)\|\hat{\mathbf{w}}_t - \mathbf{w}_t\|^2 + \rho_f\|g_i(\mathbf{w}_t) - u_{i,t}\|^2 \bigg]
\end{aligned}
\tag{15}
$$

Set $\bar{\rho} = \rho_F + \rho_g C_f + 2\rho_f C_g^2$. We have

$$
\begin{aligned}
\mathbb{E}_t[F_{1/\bar{\rho}}(\mathbf{w}_{t+1})] \leq\ & F_{1/\bar{\rho}}(\mathbf{w}_t) + \frac{\eta^2\bar{\rho}M^2}{2} + \frac{\bar{\rho}\eta}{n_+}\sum_{i \in \mathcal{S}}\bigg[ -\frac{\bar{\rho}}{2}\|\mathbf{w}_t - \hat{\mathbf{w}}_t\|^2 + f_i(g_i(\mathbf{w}_t)) - f_i(u_{i,t}) \\
&- \partial f_i(u_{i,t})^\top(g_i(\mathbf{w}_t) - u_{i,t}) + \rho_f\|g_i(\mathbf{w}_t) - u_{i,t}\|^2 \bigg] \\
\overset{(a)}{\leq}\ & F_{1/\bar{\rho}}(\mathbf{w}_t) + \frac{\eta^2\bar{\rho}M^2}{2} - \frac{\eta}{2}\|\nabla F_{1/\bar{\rho}}(\mathbf{w}_t)\|^2 + \frac{\bar{\rho}\eta}{n}\sum_{i \in \mathcal{S}}\bigg[ f_i(g_i(\mathbf{w}_t)) - f_i(u_{i,t}) \\
&- \partial f_i(u_{i,t})^\top(g_i(\mathbf{w}_t) - u_{i,t}) + \rho_f\|g_i(\mathbf{w}_t) - u_{i,t}\|^2 \bigg]
\end{aligned}
$$

where inequality (a) follows from Lemma 3.2.

Using the Lipschitz continuity of $f_i$, we have

$$
\begin{aligned}
\mathbb{E}_t[F_{1/\bar{\rho}}(\mathbf{w}_{t+1})] \leq\ & F_{1/\bar{\rho}}(\mathbf{w}_t) + \frac{\eta^2\bar{\rho}M^2}{2} - \frac{\eta}{2}\|\nabla F_{1/\bar{\rho}}(\mathbf{w}_t)\|^2 + \frac{\bar{\rho}\eta}{n}\sum_{i \in \mathcal{S}} 2C_f\|g_i(\mathbf{w}_t) - u_{i,t}\| \\
&+ \frac{\bar{\rho}\eta}{n}\sum_{i \in \mathcal{S}} \rho_f\|g_i(\mathbf{w}_t) - u_{i,t}\|^2
\end{aligned}
$$

By Lemma 4.5, the error bound of the MSVR update gives

$$\mathbb{E}\left[\frac{1}{n}\sum_{i\in\mathcal{S}}\|u_{i,t}-g_i(\mathbf{w}_t)\|\right] \le (1-\mu)^t\frac{1}{n}\sum_{i\in\mathcal{S}}\|u_{i,0}-g_i(\mathbf{w}_0)\| + R_1,$$

$$\mathbb{E}\left[\frac{1}{n}\sum_{i\in\mathcal{S}}\|u_{i,t}-g_i(\mathbf{w}_t)\|^2\right] \le (1-\mu)^t\frac{1}{n}\sum_{i\in\mathcal{S}}\|u_{i,0}-g_i(\mathbf{w}_0)\|^2 + R_2,$$

where

$$\mu = \frac{B_1\tau}{2n}, \quad R_1 = \frac{2\tau^{1/2}\sigma}{B_2^{1/2}} + \frac{4nC_gM\eta}{B_1\tau^{1/2}}, \quad R_2 = \frac{4\tau\sigma^2}{B_2} + \frac{16n^2C_g^2M^2\eta^2}{B_1^2\tau}$$

Then

$$\mathbb{E}[F_{1/\bar{\rho}}(\mathbf{w}_{t+1})] \le F_{1/\bar{\rho}}(\mathbf{w}_t) + \frac{\eta^2\bar{\rho}M^2}{2} - \frac{\eta}{2}\mathbb{E}[\|\nabla F_{1/\bar{\rho}}(\mathbf{w}_t)\|^2]$$

$$+ 2C_f\bar{\rho}\eta\left((1-\mu)^t\frac{1}{n}\sum_{i\in\mathcal{S}}\|g_i(\mathbf{w}_0)-u_{i,0}\| + R_1\right) \tag{16}$$

$$+ C\rho_f\bar{\rho}\eta\left((1-\mu)^t\frac{1}{n}\sum_{i\in\mathcal{S}}\|g_i(\mathbf{w}_0)-u_{i,0}\|^2 + R_2\right)$$

Taking summation from $t=0$ to $T-1$ yields

$$\mathbb{E}[F_{1/\bar{\rho}}(\mathbf{w}_T)]$$

$$\le F_{1/\bar{\rho}}(\mathbf{w}_0) + \frac{\eta^2\bar{\rho}M^2T}{2} - \frac{\eta}{2}\sum_{t=0}^{T-1}\mathbb{E}[\|\nabla F_{1/\bar{\rho}}(\mathbf{w}_t)\|^2]$$

$$+ 2C_f\bar{\rho}\eta\left(\sum_{t=0}^{T-1}(1-\mu)^t\frac{1}{n}\sum_{i\in\mathcal{S}}\|g_i(\mathbf{w}_0)-u_{i,0}\| + R_1T\right)$$

$$+ C\rho_f\bar{\rho}\eta\left((1-\mu)^t\frac{1}{n}\sum_{i\in\mathcal{S}}\|g_i(\mathbf{w}_0)-u_{i,0}\|^2 + R_2T\right)$$

$$\overset{(a)}{\le} F_{1/\bar{\rho}}(\mathbf{w}_0) + \frac{\eta^2\bar{\rho}M^2T}{2} - \frac{\eta}{2}\sum_{t=0}^{T-1}\mathbb{E}[\|\nabla F_{1/\bar{\rho}}(\mathbf{w}_t)\|^2]$$

$$+ \frac{2C_f\bar{\rho}\eta}{n\mu}\sum_{i\in\mathcal{S}}\|g_i(\mathbf{w}_0)-u_{i,0}\| + 2C_f\bar{\rho}\eta R_1T + \frac{\rho_f\bar{\rho}\eta}{n\mu}\sum_{i\in\mathcal{S}}\|g_i(\mathbf{w}_0)-u_{i,0}\|^2 + 2\rho_f\bar{\rho}\eta R_2T,$$

$$\tag{17}$$

where (a) uses $\sum_{t=0}^{T-1}(1-\mu)^t \le \frac{1}{\mu}$.

Lower bounding the left-hand-side by $\min_{\mathbf{w}} F(\mathbf{w})$, we obtain

$$\frac{1}{T}\sum_{t=0}^{T-1}\mathbb{E}[\|\nabla F_{1/\bar{\rho}}(\mathbf{w}_t)\|^2]$$

$$\le \frac{2}{\eta T}\left[F_{1/\bar{\rho}}(\mathbf{w}_0) - \min_{\mathbf{w}} F(\mathbf{w}) + \frac{\eta^2\bar{\rho}M^2T}{2} + \frac{2C_f\bar{\rho}\eta}{n}\sum_{i\in\mathcal{S}}\|g_i(\mathbf{w}_0)-u_{i,0}\| + 2C_f\bar{\rho}\eta R_1T\right.$$

$$\left. + \frac{\rho_f\bar{\rho}\eta}{n}\sum_{i\in\mathcal{S}}\|g_i(\mathbf{w}_0)-u_{i,0}\|^2 + \rho_f\bar{\rho}\eta R_2T\right]$$

$$\le \frac{2\Delta}{\eta T} + \eta\bar{\rho}M^2 + \frac{4C_f\bar{\rho}}{\mu Tn}\sum_{i\in\mathcal{S}}\|g_i(\mathbf{w}_0)-u_{i,0}\| + 4C_f\bar{\rho}R_1 + \frac{2\rho_f\bar{\rho}}{\mu Tn}\sum_{i\in\mathcal{S}}\|g_i(\mathbf{w}_0)-u_{i,0}\|^2 + 2\rho_f\bar{\rho}R_2$$

$$\le \frac{C}{T}(\frac{1}{\eta} + \frac{1}{\mu}) + C(\eta + R_1 + R_2)$$

where we assume $F_{1/\bar{\rho}}(\mathbf{w}_0, \mathbf{s}_0, s_0') - \min_{\mathbf{w},\mathbf{s},s'} F(\mathbf{w},\mathbf{s},s') \le \Delta$ and

$$C = \max\{8\Delta, 12\bar{\rho}M^2, \frac{16C_f\bar{\rho}}{n}\sum_{i\in\mathcal{S}}\|g_i(\mathbf{w}_0)-u_{i,0}\|, \frac{8\rho_f\bar{\rho}}{n}\sum_{i\in\mathcal{S}}\|g_i(\mathbf{w}_0)-u_{i,0}\|^2, 16C_f\bar{\rho}, 8\rho_f\bar{\rho}\}.$$

Thus

$$\frac{1}{T}\sum_{t=0}^{T-1}\mathbb{E}[\|\nabla F_{1/\bar{\rho}}(\mathbf{w}_t)\|^2]$$

$$\leq \frac{C}{T}(\frac{1}{\eta}+\frac{2n}{B_1\tau}) + C(\eta + \frac{2\tau^{1/2}\sigma}{B_2^{1/2}} + \frac{4nC_gM\eta}{B_1\tau^{1/2}} + \frac{4\tau\sigma^2}{B_2} + \frac{16n^2C_g^2M^2\eta^2}{B_1^2\tau})$$

$$= \mathcal{O}\left(\frac{1}{T}(\frac{1}{\eta}+\frac{n}{B_1\tau}) + (\eta + \frac{\tau^{1/2}\sigma}{B_2^{1/2}} + \frac{n\eta}{B_1\tau^{1/2}} + \frac{\tau\sigma^2}{B_2} + \frac{n^2\eta^2}{B_1^2\tau})\right)$$

Setting

$$\tau = \mathcal{O}(B_2\epsilon^4), \quad \eta = \mathcal{O}\left(\frac{B_1B_2^{1/2}\epsilon^4}{n}\right)$$

To reach an $\epsilon$-stationary point, we need

$$T = \mathcal{O}\left(\frac{n}{B_1B_2^{1/2}\epsilon^6}\right)$$

$\square$

## A.2   Proof of Theorem 4.7

A formal statement in given below.

**Theorem A.2.** *Under Assumption 4.3, with* $\gamma_1 = \frac{n_1n_2-B_1B_2}{B_1B_2(1-\tau_1)} + (1-\tau_1)$, $\gamma_2 = \frac{n_1-B_1}{B_1(1-\tau_2)} +$
$(1 - \tau_2)$, $\tau_1 = \mathcal{O}\left(\min\{B_3, \frac{B_1^{1/2}n_2^{1/2}}{n_1^{1/2}}\}\epsilon^4\right) \leq \frac{1}{2}$, $\tau_2 = \mathcal{O}(B_2\epsilon^4) \leq \frac{1}{2}$, $\eta =$
$\mathcal{O}\left(\min\left\{B_3^{1/2}, \frac{B_1^{1/4}n_2^{1/4}}{n_1^{1/4}}, \frac{B_1^{1/2}n_2^{1/2}}{n_1^{1/2}}\right\} \frac{B_1B_2}{n_1n_2}\epsilon^4\right)$, *and* $\bar{\rho} = \rho_F + 4\rho_fC_g^2 + 2\rho_gC_fC_h^2 + C_fC_gL_h$,
*Algorithm 2 converges to an* $\epsilon$-*stationary point of the Moreau envelope* $F_{1/\bar{\rho}}$ *in* $T =$
$\mathcal{O}\left(\max\left\{\frac{1}{B_3^{1/2}}, \frac{n_1^{1/4}}{B_1^{1/4}n_2^{1/4}}, \frac{n_1^{1/2}}{B_1^{1/2}n_2^{1/2}}\right\} \frac{n_1n_2}{B_1B_2}\epsilon^{-6}\right)$ *iterations.*

We first define constant $M^2 \geq \max\{\frac{3C_f^2C_g^2\sigma^2}{B_3} + \frac{3C_f^2C_g^2C_h^2}{B_2} + \frac{3C_f^2C_g^2C_h^2}{B_1}, \tilde{C}_h^2 + \sigma^2\}$ so that $\mathbb{E}_t[\|G_t\|^2] \leq$
$M^2$ and $\|v_{i,j,t}\|^2 \leq M^2$ for all $i \in \mathcal{S}_1, j \in \mathcal{S}_2$ and $t$. Then to prove Theorem A.2, we need the
following Lemmas.

**Lemma A.3.** *Consider MSVR update for* $v$. *Assume* $h_{i,j}(\mathbf{w};\xi)$ *is* $C_h$-*Lipshitz for all* $(i,j) \in S_1 \times S_2$
, *and* $\mathbb{E}[\|G_t\|^2] \leq M^2$. *With* $\gamma_1 = \frac{n_1n_2-B_1B_2}{B_1B_2(1-\tau_1)} + (1-\tau_1)$, *and* $\tau_1 \leq \frac{1}{2}$, *we have*

$$\mathbb{E}\left[\frac{1}{n_1}\sum_{i\in S_1}\frac{1}{n_2}\sum_{j\in S_2}\|v_{i,j,t+1} - h_{i,j}(\mathbf{w}_{t+1})\|\right]$$

$$\leq (1 - \frac{B_1B_2\tau_1}{2n_1n_2})^{t+1}\frac{1}{n_1}\sum_{i\in S_1}\frac{1}{n_2}\sum_{j\in S_2}\|v_{i,j,0} - h_{i,j}(\mathbf{w}_0)\| + \frac{2\tau_1^{1/2}\sigma}{B_3^{1/2}} + \frac{4n_1n_2C_hM\eta}{B_1B_2\tau_1^{1/2}}$$

$$\mathbb{E}\left[\frac{1}{n_1}\sum_{i\in S_1}\frac{1}{n_2}\sum_{j\in S_2}\|v_{i,j,t+1} - h_{i,j}(\mathbf{w}_{t+1})\|^2\right]$$

$$\leq (1 - \frac{B_1B_2\tau_1}{2n_1n_2})^{2(t+1)}\frac{1}{n_1}\sum_{i\in S_1}\frac{1}{n_2}\sum_{j\in S_2}\|v_{i,j,0} - h_{i,j}(\mathbf{w}_0)\|^2 + \frac{4\tau_1\sigma^2}{B_3} + \frac{16n_1^2n_2^2C_h^2M^2\eta^2}{B_1^2B_2^2\tau_1}$$

**Lemma A.4.** *Consider MSVR update for $u$. Assume $g_i(\cdot)$ is $C_g$-Lipshitz for all $i \in S_1$. With $\gamma_2 = \frac{n_+ - B_1}{B_1(1-\tau_2)} + (1-\tau_2)$ and $\tau_2 \leq \frac{1}{2}$, we have*

$$\mathbb{E}\left[\frac{1}{n_1}\sum_{i \in S_1}\left\|u_{i,t+1} - \frac{1}{n_2}\sum_{j \in S_2}g_i(v_{i,j,t+1})\right\|\right]$$

$$\leq (1 - \frac{B_1\tau_2}{2n_1})^{t+1}\frac{1}{n_1}\sum_{i \in S_1}\left\|u_{i,0} - \frac{1}{n_2}\sum_{j \in S_2}g_i(v_{i,j,0})\right\| + \frac{2\tau_2^{1/2}\sigma}{B_2^{1/2}} + \frac{C_2 n_1^{1/2}B_2^{1/2}\tau_1}{B_1^{1/2}n_2^{1/2}\tau_2^{1/2}} + \frac{C_2 n_1^{3/2}n_2^{1/2}\eta}{B_1^{3/2}B_2^{1/2}\tau_2^{1/2}}$$

*where $C_2$ is a constant defined in the proof.*

*Proof of Theorem A.2.* Consider the change in the Moreau envelope:

$$\begin{aligned}
\mathbb{E}_t[F_{1/\bar{\rho}}(\mathbf{w}_{t+1})] &= \mathbb{E}_t\left[\min_{\tilde{\mathbf{w}}}F(\tilde{\mathbf{w}}) + \frac{\bar{\rho}}{2}\|\tilde{\mathbf{w}} - \mathbf{w}_{t+1}\|^2\right] \\
&\leq \mathbb{E}_t\left[F(\hat{\mathbf{w}}_t) + \frac{\bar{\rho}}{2}\|\hat{\mathbf{w}}_t - \mathbf{w}_{t+1}\|^2\right] \\
&= F(\hat{\mathbf{w}}_t) + \mathbb{E}_t\left[\frac{\bar{\rho}}{2}\|\hat{\mathbf{w}}_t - (\mathbf{w}_t - \eta G_t)\|^2\right] \qquad (18) \\
&\leq F(\hat{\mathbf{w}}_t) + \frac{\bar{\rho}}{2}\left(\|\hat{\mathbf{w}}_t - \mathbf{w}_t\|^2\right) + \bar{\rho}\mathbb{E}_t[\eta\langle\hat{\mathbf{w}}_t - \mathbf{w}_t, G_t\rangle] + \frac{\eta^2\bar{\rho}M^2}{2} \\
&= F_{1/\bar{\rho}}(\mathbf{w}_t) + \bar{\rho}\mathbb{E}_t[\eta\langle\hat{\mathbf{w}}_t - \mathbf{w}_t, G_t\rangle] + \frac{\eta^2\bar{\rho}M^2}{2}
\end{aligned}$$

Note that

$$\mathbb{E}_t[G_t] = \frac{1}{n_1}\sum_{i=1}^{n_1}\left[\frac{1}{n_2}\sum_{j=1}^{n_2}\nabla h_{i,j}(\mathbf{w}_t)\partial g_i(v_{i,j,t})\right]\partial f_i(u_{i,t}),$$

and the second inequality uses the bound of $\mathbb{E}[\|G_t\|^2]$, which follows from the Lipschitz continuity and bounded variance assumptions and is denoted by $M$.

Define $\hat{\mathbf{w}}_t := \text{prox}_{F/\bar{\rho}}(\mathbf{w}_t)$. For a given $i \in \{1, \dots, m\}$, we have

$$\frac{1}{n_1} \sum_{i \in S_1} f_i\left(\frac{1}{n_2} \sum_{j \in S_2} g_i(h_{i,j}(\hat{\mathbf{w}}_t))\right) - \frac{1}{n_1} \sum_{i \in S_1} f_i(u_{i,t})$$

$$\overset{(a)}{\geq} \frac{1}{n_1} \sum_{i \in S_1} \partial f_i(u_{i,t})^\top \left(\frac{1}{n_2} \sum_{j \in S_2} g_i(h_{i,j}(\hat{\mathbf{w}}_t)) - u_{i,t}\right) - \frac{1}{n_1} \sum_{i \in S_1} \frac{\rho_f}{2} \left\|\frac{1}{n_2} \sum_{j \in S_2} g_i(h_{i,j}(\hat{\mathbf{w}}_t)) - u_{i,t}\right\|^2$$

$$\geq \frac{1}{n_1} \sum_{i \in S_1} \partial f_i(u_{i,t})^\top \left(\frac{1}{n_2} \sum_{j \in S_2} g_i(h_{i,j}(\hat{\mathbf{w}}_t)) - u_{i,t}\right)$$

$$- \frac{1}{n_1} \sum_{i \in S_1} \rho_f \left\|\frac{1}{n_2} \sum_{j \in S_2} g_i(h_{i,j}(\hat{\mathbf{w}}_t)) - \frac{1}{n_2} \sum_{j \in S_2} g_i(v_{i,j,t})\right\|^2 - \frac{1}{n_1} \sum_{i \in S_1} \rho_f \left\|\frac{1}{n_2} \sum_{j \in S_2} g_i(v_{i,j,t}) - u_{i,t}\right\|^2$$

$$\geq \frac{1}{n_1} \sum_{i \in S_1} \partial f_i(u_{i,t})^\top \left(\frac{1}{n_2} \sum_{j \in S_2} g_i(h_{i,j}(\hat{\mathbf{w}}_t)) - u_{i,t}\right) - \frac{1}{n_1} \sum_{i \in S_1} \frac{1}{n_2} \sum_{j \in S_2} \rho_f C_g^2 \|h_{i,j}(\hat{\mathbf{w}}_t) - v_{i,j,t}\|^2$$

$$- \frac{1}{n_1} \sum_{i \in S_1} \rho_f \left\|\frac{1}{n_2} \sum_{j \in S_2} g_i(v_{i,j,t}) - u_{i,t}\right\|^2$$

$$\overset{(b)}{\geq} \frac{1}{n_1} \sum_{i \in S_1} \partial f_i(u_{i,t})^\top \left[\frac{1}{n_2} \sum_{j \in S_2} g_i(v_{i,j,t}) - u_{i,t} + \frac{1}{n_2} \sum_{j \in S_2} \partial g_i(v_{i,j,t})^\top (h_{i,j}(\hat{\mathbf{w}}_t) - v_{i,j,t})\right.$$

$$\left. - \frac{1}{n_2} \sum_{j \in S_2} \frac{\rho_g}{2} \|h_{i,j}(\hat{\mathbf{w}}_t) - v_{i,j,t}\|^2\right] - \frac{1}{n_1} \sum_{i \in S_1} \frac{1}{n_2} \sum_{j \in S_2} 2\rho_f C_g^2 \|h_{i,j}(\mathbf{w}_t) - v_{i,j,t}\|^2$$

$$- 2\rho_f C_g^2 \|\hat{\mathbf{w}}_t - \mathbf{w}_t\|^2 - \frac{1}{n_1} \sum_{i \in S_1} \rho_f \left\|\frac{1}{n_2} \sum_{j \in S_2} g_i(v_{i,j,t}) - u_{i,t}\right\|^2$$

$$\overset{(c)}{\geq} \frac{1}{n_1} \sum_{i \in S_1} \partial f_i(u_{i,t})^\top \left[\frac{1}{n_2} \sum_{j \in S_2} g_i(v_{i,j,t}) - u_{i,t}\right]$$

$$+ \frac{1}{n_1} \sum_{i \in S_1} \frac{1}{n_2} \sum_{j \in S_2} \underbrace{\langle \partial f_i(u_{i,t})^\top \partial g_i(v_{i,j,t})^\top (h_{i,j}(\hat{\mathbf{w}}_t) - v_{i,j,t})}_{A_1}$$

$$- \frac{1}{n_1} \sum_{i \in S_1} \frac{1}{n_2} \sum_{j \in S_2} \frac{\rho_g C_f}{2} \|h_{i,j}(\hat{\mathbf{w}}_t) - v_{i,j,t}\|^2 - \frac{1}{n_1} \sum_{i \in S_1} \frac{1}{n_2} \sum_{j \in S_2} 2\rho_f C_g^2 \|h_{i,j}(\mathbf{w}_t) - v_{i,j,t}\|^2$$

$$- 2\rho_f C_g^2 \|\hat{\mathbf{w}}_t - \mathbf{w}_t\|^2 - \frac{1}{n_1} \sum_{i \in S_1} \rho_f \left\|\frac{1}{n_2} \sum_{j \in S_2} g_i(v_{i,j,t}) - u_{i,t}\right\|^2$$

$$\geq \frac{1}{n_1} \sum_{i \in S_1} \partial f_i(u_{i,t})^\top \left[\frac{1}{n_2} \sum_{j \in S_2} g_i(v_{i,j,t}) - u_{i,t}\right]$$

$$+ \frac{1}{n_1} \sum_{i \in S_1} \frac{1}{n_2} \sum_{j \in S_2} \underbrace{\langle \partial f_i(u_{i,t})^\top \partial g_i(v_{i,j,t})^\top (h_{i,j}(\hat{\mathbf{w}}_t) - v_{i,j,t})}_{A_1}$$

$$- \frac{1}{n_1} \sum_{i \in S_1} \frac{1}{n_2} \sum_{j \in S_2} (2\rho_f C_g^2 + \rho_g C_f) \|h_{i,j}(\mathbf{w}_t) - v_{i,j,t}\|^2$$

$$- (2\rho_f C_g^2 + \rho_g C_f C_h^2) \|\hat{\mathbf{w}}_t - \mathbf{w}_t\|^2 - \frac{1}{n_1} \sum_{i \in S_1} \rho_f \left\|\frac{1}{n_2} \sum_{j \in S_2} g_i(v_{i,j,t}) - u_{i,t}\right\|^2$$

$$\tag{19}$$

where (a) follows from the convexity of $f_i$, (b) uses the assumption that $f_i(\cdot)$ is non-decreasing and $g_i$ is weak convex, (c) is due to $0 \leq \partial f_i(u_{i,t}) \leq C_f$.

The $L_h$-smoothness assumption of $h_{i,j}(\mathbf{w})$ (or weakly-convexity of $h_{i,j}(\mathbf{w})$, then only the second inequality holds) for all $i, \mathbf{w}$ implies

$$h_{i,j}(\hat{\mathbf{w}}_t) \leq h_{i,j}(\mathbf{w}_t) + \nabla h_{i,j}(\mathbf{w}_t)^\top (\hat{\mathbf{w}}_t - \mathbf{w}_t) + \frac{L_h}{2} \|\hat{\mathbf{w}}_t - \mathbf{w}_t\|^2,$$

$$h_{i,j}(\hat{\mathbf{w}}_t) \geq h_{i,j}(\mathbf{w}_t) + \nabla h_{i,j}(\mathbf{w}_t)^\top (\hat{\mathbf{w}}_t - \mathbf{w}_t) - \frac{L_h}{2} \|\hat{\mathbf{w}}_t - \mathbf{w}_t\|^2. \tag{20}$$

We first assume that $g_i(\cdot)$ is non-increasing. Since $\partial f_i(u_{i,t}) \geq 0$ and $\partial g_i(v_{i,j,t}) \leq 0$, we bound $A_1$ as following

$$A_1 = \partial f_i(u_{i,t}) \partial^\top g_i(v_{i,j,t})^\top (h_{i,j}(\hat{\mathbf{w}}_t) - v_{i,j,t})$$

$$\overset{(a)}{\geq} \langle \partial f_i(u_{i,t})^\top \partial g_i(v_{i,j,t})^\top (h_{i,j}(\mathbf{w}_t) - v_{i,j,t}) + \partial f_i(u_{i,t})^\top \partial g_i(v_{i,j,t})^\top \nabla h_{i,j}(\mathbf{w}_t)^\top (\hat{\mathbf{w}}_t - \mathbf{w}_t)$$

$$+ \partial f_i(u_{i,t})^\top \partial g_i(v_{i,j,t})^\top \frac{L_h}{2} \|\hat{\mathbf{w}}_t - \mathbf{w}_t\|^2 \rangle$$

$$\overset{(b)}{\geq} -C_f C_g \|h_{i,j}(\mathbf{w}_t) - v_{i,j,t}\| + \partial f_i(u_{i,t})^\top \partial g_i(v_{i,j,t})^\top \nabla h_{i,j}(\mathbf{w}_t)^\top (\hat{\mathbf{w}}_t - \mathbf{w}_t)$$

$$- \frac{C_f C_g L_h}{2} \|\hat{\mathbf{w}}_t - \mathbf{w}_t\|^2 \tag{21}$$

where inequality (a) follows from the first inequality in (20), (b) follows from the Lipschitz continuity and monotone assumptions on $f_i, g_i, h_{i,j}$. On the other hand, if we assume $g_i(\cdot)$ is non-decreasing, we may use the second inequality in (20) and obtain the same result as (21). Now plugging the new formulation of $A_1$ back to inequality 19 yields

$$\frac{1}{n_1} \sum_{i \in S_1} f_i\left(\frac{1}{n_2} \sum_{j \in S_2} g_i(h_{i,j}(\hat{\mathbf{w}}_t))\right) - \frac{1}{n_1} \sum_{i \in S_1} f_i(u_{i,t})$$

$$\geq \frac{1}{n_1} \sum_{i \in S_1} \partial f_i(u_{i,t})^\top \left[\frac{1}{n_2} \sum_{j \in S_2} g_i(v_{i,j,t}) - u_{i,t}\right] + \frac{1}{n_1} \sum_{i \in S_1} \frac{1}{n_2} \sum_{j \in S_2} -C_f C_g \|h_{i,j}(\mathbf{w}_t) - v_{i,j,t}\|$$

$$+ \frac{1}{n_1} \sum_{i \in S_1} \frac{1}{n_2} \sum_{j \in S_2} \partial f_i(u_{i,t})^\top \partial g_i(v_{i,j,t})^\top \nabla h_{i,j}(\mathbf{w}_t)^\top (\hat{\mathbf{w}}_t - \mathbf{w}_t) - \frac{C_f C_g L_h}{2} \|\hat{\mathbf{w}}_t - \mathbf{w}_t\|^2$$

$$- \frac{1}{n_1} \sum_{i \in S_1} \frac{1}{n_2} \sum_{j \in S_2} (2\rho_f C_g^2 + \rho_g C_f) \|h_{i,j}(\mathbf{w}_t) - v_{i,j,t}\|^2$$

$$- (2\rho_f C_g^2 + \rho_g C_f C_h^2) \|\hat{\mathbf{w}}_t - \mathbf{w}_t\|^2 - \frac{1}{n_1} \sum_{i \in S_1} \rho_f \left\|\frac{1}{n_2} \sum_{j \in S_2} g_i(v_{i,j,t}) - u_{i,t}\right\|^2$$

$$\geq \frac{1}{n_1} \sum_{i \in S_1} -C_f \left\|\frac{1}{n_2} \sum_{j \in S_2} g_i(v_{i,j,t}) - u_{i,t}\right\| + \frac{1}{n_1} \sum_{i \in S_1} \frac{1}{n_2} \sum_{j \in S_2} -C_f C_g \|h_{i,j}(\mathbf{w}_t) - v_{i,j,t}\|$$

$$+ \langle \mathbb{E}_t[G_t], \hat{\mathbf{w}}_t - \mathbf{w}_t \rangle - \frac{1}{n_1} \sum_{i \in S_1} \frac{1}{n_2} \sum_{j \in S_2} (2\rho_f C_g^2 + \rho_g C_f) \|h_{i,j}(\mathbf{w}_t) - v_{i,j,t}\|^2$$

$$- (2\rho_f C_g^2 + \rho_g C_f C_h^2 + \frac{C_f C_g L_h}{2}) \|\hat{\mathbf{w}}_t - \mathbf{w}_t\|^2 - \frac{1}{n_1} \sum_{i \in S_1} \rho_f \left\|\frac{1}{n_2} \sum_{j \in S_2} g_i(v_{i,j,t}) - u_{i,t}\right\|^2$$

It follows
$$\langle \mathbb{E}_t[G_t], \hat{\mathbf{w}}_t - \mathbf{w}_t \rangle$$

$$\leq \frac{1}{n_1} \sum_{i \in S_1} f_i\left(\frac{1}{n_2} \sum_{j \in S_2} g_i(h_{i,j}(\hat{\mathbf{w}}_t))\right) - \frac{1}{n_1} \sum_{i \in S_1} f_i(u_{i,t}) + \frac{1}{n_1} \sum_{i \in S_1} C_f \left\|\frac{1}{n_2} \sum_{j \in S_2} g_i(v_{i,j,t}) - u_{i,t}\right\|$$

$$+ \frac{1}{n_1} \sum_{i \in S_1} \frac{1}{n_2} \sum_{j \in S_2} C_f C_g \|h_{i,j}(\mathbf{w}_t) - v_{i,j,t}\| + \frac{1}{n_1} \sum_{i \in S_1} \frac{1}{n_2} \sum_{j \in S_2} (2\rho_f C_g^2 + \rho_g C_f) \|h_{i,j}(\mathbf{w}_t) - v_{i,j,t}\|^2$$

$$+ (2\rho_f C_g^2 + \rho_g C_f C_h^2 + \frac{C_f C_g L_h}{2}) \|\hat{\mathbf{w}}_t - \mathbf{w}_t\|^2 + \frac{1}{n_1} \sum_{i \in S_1} \rho_f \left\|\frac{1}{n_2} \sum_{j \in S_2} g_i(v_{i,j,t}) - u_{i,t}\right\|^2 \tag{22}$$

Combining inequality 22 and 18 yields

$$\mathbb{E}_t[F_{1/\bar{\rho}}(\mathbf{w}_{t+1})]$$

$$\leq F_{1/\bar{\rho}}(\mathbf{w}_t) + \frac{\eta^2\bar{\rho}M^2}{2} + \bar{\rho}\eta\bigg\{\frac{1}{n_1}\sum_{i\in S_1}\bigg[f_i(\frac{1}{n_2}\sum_{j\in S_2}g_i(h_{i,j}(\hat{\mathbf{w}}_t))) - f_i(u_{i,t})$$

$$+ C_f\bigg\|\frac{1}{n_2}\sum_{j\in S_2}g_i(v_{i,j,t}) - u_{i,t}\bigg\| + \frac{1}{n_2}\sum_{j\in S_2}C_fC_g\|h_{i,j}(\mathbf{w}_t) - v_{i,j,t}\|$$

$$+ \frac{1}{n_2}\sum_{j\in S_2}(2\rho_fC_g^2 + \rho_gC_f)\|h_{i,j}(\mathbf{w}_t) - v_{i,j,t}\|^2$$

$$+ (2\rho_fC_g^2 + \rho_gC_fC_h^2 + \frac{C_fC_gL_h}{2})\|\hat{\mathbf{w}}_t - \mathbf{w}_t\|^2 + \rho_f\bigg\|\frac{1}{n_2}\sum_{j\in S_2}g_i(v_{i,j,t}) - u_{i,t}\bigg\|^2\bigg]\bigg\}$$

$$\leq F_{1/\bar{\rho}}(\mathbf{w}_t) + \frac{\eta^2\bar{\rho}M^2}{2} + \bar{\rho}\eta\bigg\{\frac{1}{n_1}\sum_{i\in S_1}\bigg[F_i(\hat{\mathbf{w}}_t) - F_i(\mathbf{w}_t) + F_i(\mathbf{w}_t) - f_i(\frac{1}{n_2}\sum_{j\in S_2}g_i(v_{i,j,t}))$$

$$+ f_i(\frac{1}{n_2}\sum_{j\in S_2}g_i(v_{i,j,t})) - f_i(u_{i,t}) + C_f\bigg\|\frac{1}{n_2}\sum_{j\in S_2}g_i(v_{i,j,t}) - u_{i,t}\bigg\|$$

$$+ \frac{1}{n_2}\sum_{j\in S_2}C_fC_g\|h_{i,j}(\mathbf{w}_t) - v_{i,j,t}\| + \frac{1}{n_2}\sum_{j\in S_2}(2\rho_fC_g^2 + \rho_gC_f)\|h_{i,j}(\mathbf{w}_t) - v_{i,j,t}\|^2$$

$$+ (2\rho_fC_g^2 + \rho_gC_fC_h^2 + \frac{C_fC_gL_h}{2})\|\hat{\mathbf{w}}_t - \mathbf{w}_t\|^2 + \rho_f\bigg\|\frac{1}{n_2}\sum_{j\in S_2}g_i(v_{i,j,t}) - u_{i,t}\bigg\|^2\bigg]\bigg\}$$

$$\overset{(a)}{\leq} F_{1/\bar{\rho}}(\mathbf{w}_t) + \frac{\eta^2\bar{\rho}M^2}{2} + \bar{\rho}\eta\bigg\{\frac{1}{n_1}\sum_{i\in S_1}\bigg[F_i(\hat{\mathbf{w}}_t) - F_i(\mathbf{w}_t) + 2C_f\bigg\|\frac{1}{n_2}\sum_{j\in S_2}g_i(v_{i,j,t}) - u_{i,t}\bigg\|$$

$$+ \frac{1}{n_2}\sum_{j\in S_2}2C_fC_g\|h_{i,j}(\mathbf{w}_t) - v_{i,j,t}\| + \frac{1}{n_2}\sum_{j\in S_2}(2\rho_fC_g^2 + \rho_gC_f)\|h_{i,j}(\mathbf{w}_t) - v_{i,j,t}\|^2$$

$$+ (2\rho_fC_g^2 + \rho_gC_fC_h^2 + \frac{C_fC_gL_h}{2})\|\hat{\mathbf{w}}_t - \mathbf{w}_t\|^2 + \rho_f\bigg\|\frac{1}{n_2}\sum_{j\in S_2}g_i(v_{i,j,t}) - u_{i,t}\bigg\|^2\bigg]\bigg\}$$

where (a) follows from the Lipschitz continuity of $f_i, g_i, h_{i,j}$.

Due to the $\rho_F$-weak convexity of $F_i(\mathbf{w})$, we have $(\bar{\rho} - \rho_F)$-strong convexity of $\mathbf{w} \mapsto F_i(\mathbf{w}) + \frac{\bar{\rho}}{2}\|\mathbf{w}_t - \mathbf{w}\|^2$. Then it follows

$$F_i(\hat{\mathbf{w}}_t) - F_i(\mathbf{w}_t) = \bigg[F_i(\hat{\mathbf{w}}_t) + \frac{\bar{\rho}}{2}\|\mathbf{w}_t - \hat{\mathbf{w}}_t\|^2\bigg] - \bigg[F_i(\mathbf{w}_t) + \frac{\bar{\rho}}{2}\|\mathbf{w}_t - \mathbf{w}_t\|^2\bigg] - \frac{\bar{\rho}}{2}\|\mathbf{w}_t - \hat{\mathbf{w}}_t\|^2$$

$$\leq (\frac{\rho_F}{2} - \bar{\rho})\|\mathbf{w}_t - \hat{\mathbf{w}}_t\|^2$$

$$\tag{23}$$

Plugging inequality 23 back into A.2, we obtain

$$\mathbb{E}_t[F_{1/\bar{\rho}}(\mathbf{w}_{t+1})]$$

$$\leq F_{1/\bar{\rho}}(\mathbf{w}_t) + \frac{\eta^2 \bar{\rho} M^2}{2} + \bar{\rho}\eta \Bigg\{ \frac{1}{n_1} \sum_{i \in S_1} \Bigg[ (\frac{\rho_F}{2} - \bar{\rho})\|\mathbf{w}_t - \hat{\mathbf{w}}_t\|^2 + 2C_f \Bigg\| \frac{1}{n_2} \sum_{j \in S_2} g_i(v_{i,j,t}) - u_{i,t} \Bigg\|$$

$$+ \frac{1}{n_2} \sum_{j \in S_2} 2C_f C_g \|h_{i,j}(\mathbf{w}_t) - v_{i,j,t}\| + \frac{1}{n_2} \sum_{j \in S_2} (2\rho_f C_g^2 + \rho_g C_f)\|h_{i,j}(\mathbf{w}_t) - v_{i,j,t}\|^2$$

$$+ (2\rho_f C_g^2 + \rho_g C_f C_h^2 + \frac{C_f C_g L_h}{2})\|\hat{\mathbf{w}}_t - \mathbf{w}_t\|^2 + \rho_f \Bigg\| \frac{1}{n_2} \sum_{j \in S_2} g_i(v_{i,j,t}) - u_{i,t} \Bigg\|^2 \Bigg] \Bigg\}$$

$$\overset{(a)}{\leq} F_{1/\bar{\rho}}(\mathbf{w}_t) + \frac{\eta^2 \bar{\rho} M^2}{2} + \bar{\rho}\eta \Bigg\{ \frac{1}{n_1} \sum_{i \in S_1} \Bigg[ -\frac{\bar{\rho}}{2}\|\mathbf{w}_t - \hat{\mathbf{w}}_t\|^2 + C_1 \Bigg\| \frac{1}{n_2} \sum_{j \in S_2} g_i(v_{i,j,t}) - u_{i,t} \Bigg\|$$

$$+ \frac{1}{n_2} \sum_{j \in S_2} C_1 \|h_{i,j}(\mathbf{w}_t) - v_{i,j,t}\| + \frac{1}{n_2} \sum_{j \in S_2} C_1 \|h_{i,j}(\mathbf{w}_t) - v_{i,j,t}\|^2$$

$$+ C_1 \Bigg\| \frac{1}{n_2} \sum_{j \in S_2} g_i(v_{i,j,t}) - u_{i,t} \Bigg\|^2 \Bigg] \Bigg\}$$

$$\overset{(b)}{=} F_{1/\bar{\rho}}(\mathbf{w}_t) + \frac{\eta^2 \bar{\rho} M^2}{2} - \frac{\eta}{2}\|\nabla F_{1/\bar{\rho}}(\mathbf{w}_t)\|^2 + C_1 \bar{\rho}\eta \frac{1}{n_1} \sum_{i \in S_1} \Bigg\| \frac{1}{n_2} \sum_{j \in S_2} g_i(v_{i,j,t}) - u_{i,t} \Bigg\|$$

$$+ C_1 \bar{\rho}\eta \frac{1}{n_1} \sum_{i \in S_1} \frac{1}{n_2} \sum_{j \in S_2} \|h_{i,j}(\mathbf{w}_t) - v_{i,j,t}\| + C_1 \bar{\rho}\eta \frac{1}{n_1} \sum_{i \in S_1} \frac{1}{n_2} \sum_{j \in S_2} \|h_{i,j}(\mathbf{w}_t) - v_{i,j,t}\|^2$$

$$+ C_1 \bar{\rho}\eta \frac{1}{n_1} \sum_{i \in S_1} \Bigg\| \frac{1}{n_2} \sum_{j \in S_2} g_i(v_{i,j,t}) - u_{i,t} \Bigg\|^2$$

where in inequality (a) we use $\bar{\rho} = \rho_F + 4\rho_f C_g^2 + 2\rho_g C_f C_h^2 + C_f C_g L_h$ and $C_1 = \max\{2C_f C_g, 2C_f, (2\rho_f C_g^2 + \rho_g C_f), \rho_f\}$, and equality (b) uses Lemma 3.2.

With general error bounds

$$\frac{1}{n_1} \sum_{i \in S_1} \frac{1}{n_2} \sum_{j \in S_2} \mathbb{E}[\|h_{i,j}(\mathbf{w}_t) - v_{i,j,t}\|] \leq (1 - \mu_1)^t \frac{1}{n_1} \sum_{i \in S_1} \frac{1}{n_2} \sum_{j \in S_2} \|h_{i,j}(\mathbf{w}_0) - v_{i,j,0}\| + R_1,$$

$$\frac{1}{n_1} \sum_{i \in S_1} \frac{1}{n_2} \sum_{j \in S_2} \mathbb{E}[\|h_{i,j}(\mathbf{w}_t) - v_{i,j,t}\|^2] \leq (1 - \mu_1)^t \frac{1}{n_1} \sum_{i \in S_1} \frac{1}{n_2} \sum_{j \in S_2} \|h_{i,j}(\mathbf{w}_0) - v_{i,j,0}\|^2 + R_2,$$

$$\frac{1}{n_1} \sum_{i \in S_1} \mathbb{E}\Bigg[\Bigg\| \frac{1}{n_2} \sum_{j \in S_2} g_i(v_{i,j,t}) - u_{i,t} \Bigg\|\Bigg] \leq (1 - \mu_2)^t \frac{1}{n_+} \sum_{i \in S_+} \Bigg\| \frac{1}{n_2} \sum_{j \in S_2} g_i(v_{i,j,0}) - u_{i,0} \Bigg\| + R_3,$$

$$\frac{1}{n_1} \sum_{i \in S_1} \mathbb{E}\Bigg[\Bigg\| \frac{1}{n_2} \sum_{j \in S_2} g_i(v_{i,j,t}) - u_{i,t} \Bigg\|^2\Bigg] \leq (1 - \mu_2)^t \frac{1}{n_+} \sum_{i \in S_+} \Bigg\| \frac{1}{n_2} \sum_{j \in S_2} g_i(v_{i,j,0}) - u_{i,0} \Bigg\|^2 + R_4,$$

we have

$$\mathbb{E}[F_{1/\bar{\rho}}(\mathbf{w}_{t+1})]$$

$$\leq \mathbb{E}[F_{1/\bar{\rho}}(\mathbf{w}_t)] + \frac{\eta^2 \bar{\rho} M^2}{2} - \frac{\eta}{2}\mathbb{E}[\|\nabla F_{1/\bar{\rho}}(\mathbf{w}_t)\|^2] + C_1 \bar{\rho}\eta(1 - \mu_{min})^t \Bigg[ \frac{1}{n_1} \sum_{i \in S_1} \Bigg\| \frac{1}{n_2} \sum_{j \in S_2} g_i(v_{i,j,0}) - u_{i,0} \Bigg\|$$

$$+ \frac{1}{n_1} \sum_{i \in S_1} \Bigg\| \frac{1}{n_2} \sum_{j \in S_2} g_i(v_{i,j,0}) - u_{i,0} \Bigg\|^2 + \frac{1}{n_1} \sum_{i \in S_1} \frac{1}{n_2} \sum_{j \in S_2} \|h_{i,j}(\mathbf{w}_0) - v_{i,j,0}\|$$

$$+ \frac{1}{n_1} \sum_{i \in S_1} \frac{1}{n_2} \sum_{j \in S_2} \|h_{i,j}(\mathbf{w}_0) - v_{i,j,0}\|^2 \Bigg] + C_1 \bar{\rho}\eta(R_1 + R_2 + R_3 + R_4),$$

where $\mu_{min} = \min\{\mu_1, \mu_2\}$.

Taking summation from $t = 0$ to $T - 1$ yields

$\mathbb{E}[F_{1/\bar{\rho}}(\mathbf{w}_T)]$

$$\leq \mathbb{E}[F_{1/\bar{\rho}}(\mathbf{w}_0)] + \frac{\eta^2 \bar{\rho} M^2 T}{2} - \frac{\eta}{2} \sum_{t=0}^{T-1} \mathbb{E}[\|\nabla F_{1/\bar{\rho}}(\mathbf{w}_t)\|^2] + C_1 \bar{\rho} \eta \sum_{t=0}^{T-1} (1 - \mu_{min})^t \Delta_0$$

$$+ T C_1 \bar{\rho} \eta (R_1 + R_2 + R_3 + R_4)$$

$$\leq \mathbb{E}[F_{1/\bar{\rho}}(\mathbf{w}_0)] + \frac{\eta^2 \bar{\rho} M^2 T}{2} - \frac{\eta}{2} \sum_{t=0}^{T-1} \mathbb{E}[\|\nabla F_{1/\bar{\rho}}(\mathbf{w}_t)\|^2] + \frac{C_1 \bar{\rho} \eta \Delta_0}{\mu_{min}} + T C_1 \bar{\rho} \eta (R_1 + R_2 + R_3 + R_4)$$

where we use $\sum_{t=0}^{T-1} (1 - \mu_{min})^t \leq \frac{1}{\mu_{min}}$ and define constant $\Delta_0$ such that

$$\left[ \frac{1}{n_1} \sum_{i \in S_1} \left\| \frac{1}{n_2} \sum_{j \in S_2} g_i(v_{i,j,0}) - u_{i,0} \right\| + \frac{1}{n_1} \sum_{i \in S_1} \left\| \frac{1}{n_2} \sum_{j \in S_2} g_i(v_{i,j,0}) - u_{i,0} \right\|^2 \right.$$

$$\left. + \frac{1}{n_1} \sum_{i \in S_1} \frac{1}{n_2} \sum_{j \in S_2} \|h_{i,j}(\mathbf{w}_0) - v_{i,j,0}\| + \frac{1}{n_1} \sum_{i \in S_1} \frac{1}{n_2} \sum_{j \in S_2} \|h_{i,j}(\mathbf{w}_0) - v_{i,j,0}\|^2 \right] \leq \Delta_0.$$

Then it follows

$$\frac{1}{T} \sum_{t=0}^{T-1} \mathbb{E}[\|\nabla F_{1/\bar{\rho}}(\mathbf{w}_t)\|^2]$$

$$\leq \frac{2}{\eta T} \left[ F_{1/\bar{\rho}}(\mathbf{w}_0) - \mathbb{E}[F_{1/\bar{\rho}}(\mathbf{w}_T)] + \frac{\eta^2 \bar{\rho} M^2 T}{2} + \frac{C_1 \bar{\rho} \eta \Delta_0}{\mu_{min}} + T C_1 \bar{\rho} \eta (R_1 + R_2 + R_3 + R_4) \right]$$

$$\leq \frac{2\Delta}{\eta T} + \eta \bar{\rho} M^2 + \frac{2 C_1 \bar{\rho} \Delta_0}{\mu_{min} T} + 2 C_1 \bar{\rho} (R_1 + R_2 + R_3)$$

$$= \mathcal{O}\left( \frac{1}{T} \left( \frac{1}{\eta} + \frac{1}{\mu_{min}} \right) + \eta + R_1 + R_2 + R_3 + R_4 \right)$$

where we define constant $\Delta$ such that $F_{1/\bar{\rho}}(\mathbf{w}_0, \mathbf{s}_0, s_0') - \mathbb{E}[F_{1/\bar{\rho}}(\mathbf{w}_T, \mathbf{s}_T, s_T')] \leq \Delta$.

With MSVR updates for $v_{i,j,t}$ and $u_{i,t}$, following from Lemma A.3 and Lemma A.4, we have

$$\mu_1 = \frac{B_1 B_2 \tau_1}{2 n_1 n_2}, \quad \mu_2 = \frac{B_1 \tau_2}{2 n_1}, \quad R_1 = \frac{2 \tau_1^{1/2} \sigma}{B_3^{1/2}} + \frac{4 n_1 n_2 \sqrt{C_h} M \eta}{B_1 B_2 \tau_1^{1/2}}$$

$$R_2 = \frac{4 \tau_1 \sigma^2}{B_3} + \frac{16 n_1^2 n_2^2 C_h M^2 \eta^2}{B_1^2 B_2^2 \tau_1}, \quad R_3 = \frac{2 \tau_2^{1/2} \sigma}{B_2^{1/2}} + \frac{C_2 n_1^{1/2} B_2^{1/2} \tau_1}{B_1^{1/2} n_2^{1/2} \tau_2^{1/2}} + \frac{C_2 n_1^{3/2} n_2^{1/2} \eta}{B_1^{3/2} B_2^{1/2} \tau_2^{1/2}},$$

$$R_4 = \frac{4 \tau_2 \sigma^2}{B_2} + \frac{C_2^2 n_1 B_2 \tau_1^2}{B_1 n_2 \tau_2} + \frac{C_2^2 n_1^3 n_2 \eta^2}{B_1^3 B_2 \tau_2}.$$

Then

$$\frac{1}{T} \sum_{t=0}^{T-1} \mathbb{E}[\|\nabla F_{1/\bar{\rho}}(\mathbf{w}_t)\|^2]$$

$$\leq \mathcal{O}\left( \frac{1}{T} \left( \frac{1}{\eta} + \frac{1}{\mu_{min}} \right) + \eta + \frac{\tau_1^{1/2}}{B_3^{1/2}} + \frac{\tau_1}{B_3} + \frac{\tau_2^{1/2}}{B_2^{1/2}} + \frac{\tau_2}{B_2} \right.$$

$$\left. + \frac{n_1 n_2 \eta}{B_1 B_2 \tau_1^{1/2}} + \frac{n_1^2 n_2^2 \eta^2}{B_1^2 B_2^2 \tau_1} + \frac{n_1^{1/2} B_2^{1/2} \tau_1}{B_1^{1/2} n_2^{1/2} \tau_2^{1/2}} + \frac{n_1^{3/2} n_2^{1/2} \eta}{B_1^{3/2} B_2^{1/2} \tau_2^{1/2}} + \frac{n_1 B_2 \tau_1^2}{B_1 n_2 \tau_2} + \frac{n_1^3 n_2 \eta^2}{B_1^3 B_2 \tau_2} \right)$$

$$\leq \mathcal{O}\left( \frac{1}{T} \left( \frac{1}{\eta} + \frac{1}{\mu_{min}} \right) + \frac{\tau_1^{1/2}}{B_3^{1/2}} + \frac{\tau_2^{1/2}}{B_2^{1/2}} + \frac{n_1 n_2 \eta}{B_1 B_2 \tau_1^{1/2}} + \frac{n_1^{1/2} B_2^{1/2} \tau_1}{B_1^{1/2} n_2^{1/2} \tau_2^{1/2}} + \frac{n_1^{3/2} n_2^{1/2} \eta}{B_1^{3/2} B_2^{1/2} \tau_2^{1/2}} \right).$$

**Algorithm 3** Stochastic Optimization algorithm for Non-smooth FCCO with coordinate moving average

1: Initialization: $\mathbf{w}_0, \{u_{i,0} : i \in \mathcal{S}\}$.
2: **for** $t = 0, \ldots, T-1$ **do**
3:    Draw sample batches $\mathcal{B}_1^t \sim \mathcal{S}$, and $\mathcal{B}_{2,i}^t \sim \mathcal{D}_i$ for each $i \in \mathcal{B}_1^t$.
4:    $u_{i,t+1} = \begin{cases} (1-\tau)u_{i,t} + \tau g_i(\mathbf{w}_t; \mathcal{B}_{2,i}^t), & i \in \mathcal{B}_1^t \\ u_{i,t}, & i \notin \mathcal{B}_1^t \end{cases}$
5:    Compute $G_t = \frac{1}{B_1} \sum_{i \in \mathcal{B}_1^t} \partial g_i(\mathbf{w}_t; \mathcal{B}_{2,i}^t) \partial f_i(u_{i,t})$
6:    Update $\mathbf{w}_{t+1} = \mathbf{w}_t - \eta G_t$
7: **end for**
8: **return** $\mathbf{w}_{\bar{t}}$ with uniformly sampled $\bar{t} \in \{0, T-1\}$.

Setting

$$\tau_1 = \mathcal{O}\left(\min\{B_3, \frac{B_1^{1/2}n_2^{1/2}}{n_1^{1/2}}\}\epsilon^4\right), \quad \tau_2 = \mathcal{O}(B_2\epsilon^4),$$

$$\eta = \mathcal{O}\left(\min\left\{\frac{B_1 B_2}{n_1 n_2}\tau_1^{1/2}\epsilon^2, \frac{B_1^{3/2}B_2^{1/2}}{n_1^{3/2}n_2^{1/2}}\tau_2^{1/2}\right\}\right)$$

$$= \mathcal{O}\left(\min\left\{B_3^{1/2}, \frac{B_1^{1/4}n_2^{1/4}}{n_1^{1/4}}, \frac{B_1^{1/2}n_2^{1/2}}{n_1^{1/2}}\right\}\frac{B_1 B_2}{n_1 n_2}\epsilon^4\right),$$

then with

$$T = \mathcal{O}\left(\max\left\{\frac{1}{B_3^{1/2}}, \frac{n_1^{1/4}}{B_1^{1/4}n_2^{1/4}}, \frac{n_1^{1/2}}{B_1^{1/2}n_2^{1/2}}\right\}\frac{n_1 n_2}{B_1 B_2}\epsilon^{-6}\right),$$

we have

$$\frac{1}{T}\sum_{t=0}^{T-1}\mathbb{E}[\|\nabla F_{1/\bar{\rho}}(\mathbf{w}_t)\|^2] \leq \epsilon^2$$

$\square$

# B   Solving Non-smooth FCCO and TCCO with Coordinate Moving Average

In this section we consider solving non-smooth weakly-convex FCCO and TCCO without variance reduction method. To be specific, we use coordinate moving average updates for function values estimations instead of MSVR. This allows us to weaken the assumption on the Lipschitz continuity, i.e. the Lipschitz continuity of the stochastic function value estimation is not required, and can be replaced by the Lipschitz continuity of the function value. Moreover, compared with MSVR, coordinate moving average update does not need the stochastic evaluation from the previous iteration, and thus has a simpler implementation. However, as a result of not using variance reduction technique, the algorithms suffer from worse convergence rates in terms of $\epsilon$.

## B.1   Solving Non-smooth FCCO with Coordinate Moving Average

We first assume the followings assumptions hold.

**Assumption B.1.** For all $i \in \mathcal{S}$, we assume that

• $f_i(\cdot)$ is $\rho_f$-weakly-convex, $C_f$-Lipschitz continuous and non-decreasing;

• $g_i(\cdot)$ is $\rho_g$-weakly-convex and $C_g$-Lipschitz continuous;

• Stochastic gradient estimators $g_i(\mathbf{w}; \xi)$ and $\partial g_i(\mathbf{w}; \xi)$ have bounded variance $\sigma^2$.

With coordinate moving average update, we present the following lemma of error bound.

**Lemma B.2.** *Consider the coordinate moving average update for $\{u_{i,t} : i \in \mathcal{S}_1\}$ in Algorithm 3, assume $g_i(\mathbf{w})$ is $C_g$-Lipschitz continuous for all $i \in \mathcal{S}_1$ and $\tau \le 1$, then we have*

$$\mathbb{E}[\|u_{i,t+1} - g_i(\mathbf{w}_{t+1})\|] \le (1 - \frac{B_1\tau}{4n_1})^{t+1}\|u_{i,0} - g_i(\mathbf{w}_0)\| + \frac{2\sqrt{2}\tau^{1/2}\sigma}{B_2^{1/2}} + \frac{4\sqrt{2}n_1 C_g M\eta}{B_1\tau},$$

$$\mathbb{E}[\|u_{i,t+1} - g_i(\mathbf{w}_{t+1})\|^2] \le (1 - \frac{B_1\tau}{4n_1})^{2(t+1)}\|u_{i,0} - g_i(\mathbf{w}_0)\|^2 + \frac{8\tau\sigma^2}{B_2} + \frac{32n_1^2 C_g^2 M^2\eta^2}{B_1^2\tau^2}.$$

Then we have a convergence analysis similar to Theorem 4.6.

**Theorem B.3.** *Consider non-smooth weakly-convex FCCO problem, under Assumption B.1, setting $\tau = \mathcal{O}(B_2\epsilon^4) \le 1$, $\eta = \mathcal{O}(\frac{B_1 B_2}{n_1}\epsilon^6)$, Algorithm 3 converges to an $\epsilon$-stationary point of the Moreau envelope $F_{1/\bar{\rho}}$ in $T = \mathcal{O}(\frac{n_1}{B_1 B_2}\epsilon^{-8})$ iterations.*

*Proof of Theorem B.3.* Since the only difference between SONX and Algorithm 3 is the update for $\{u_{i,t} : i \in \mathcal{S}_1\}$, the proof of Theorem 4.6 still holds with the error bound replaced by Lemma B.2, i.e.,

$$\mathbb{E}\left[\frac{1}{n}\sum_{i\in\mathcal{S}}\|u_{i,t+1} - g_i(\mathbf{w}_{t+1})\|\right] \le (1 - \mu)^{t+1}\frac{1}{n}\sum_{i\in\mathcal{S}}\|u_{i,0} - g_i(\mathbf{w}_0)\| + R_1,$$

$$\mathbb{E}\left[\frac{1}{n}\sum_{i\in\mathcal{S}}\|u_{i,t+1} - g_i(\mathbf{w}_{t+1})\|^2\right] \le (1 - \mu)^{t+1}\frac{1}{n}\sum_{i\in\mathcal{S}}\|u_{i,0} - g_i(\mathbf{w}_0)\|^2 + R_2,$$

$$\mu = \frac{B_1\tau}{4n_1}, \quad R_1 = \frac{2\sqrt{2}\tau^{1/2}\sigma}{B_2^{1/2}} + \frac{4\sqrt{2}n_1 C_g M\eta}{B_1\tau}, \quad R_2 = \frac{8\tau\sigma^2}{B_2} + \frac{32n_1^2 C_g^2 M^2\eta^2}{B_1^2\tau^2}.$$

Then proof proceeds to

$$\frac{1}{T}\sum_{t=0}^{T-1}\mathbb{E}[\|\nabla F_{1/\bar{\rho}}(\mathbf{w}_t)\|^2] \le \mathcal{O}\left(\frac{1}{T}(\frac{1}{\eta} + \frac{1}{\mu}) + \eta + R_1 + R_2\right)$$

$$= \mathcal{O}\left(\frac{1}{T}(\frac{1}{\eta} + \frac{n_1}{B_1\tau}) + \eta + \frac{\tau^{1/2}\sigma}{B_2^{1/2}} + \frac{n_1\eta}{B_1\tau} + \frac{\tau\sigma^2}{B_2} + \frac{n_1^2\eta^2}{B_1^2\tau^2}\right).$$

Setting

$$\tau = \mathcal{O}(B_2\epsilon^4), \quad \eta = \mathcal{O}(\frac{B_1 B_2}{n_1}\epsilon^6),$$

then to reach a nearly $\epsilon$-stationary point, Algorithm 3 needs

$$T = \mathcal{O}(\frac{n_1}{B_1 B_2}\epsilon^{-8})$$

iterations. $\qquad\square$

## B.2 Solving Non-smooth TCCO with Coordinate Moving Average

We first assume the following assumptions hold.

**Assumption B.4.** For all $(i, j) \in \mathcal{S}_1 \times \mathcal{S}_2$, we assume that

- $f_i(\cdot)$ is $\rho_f$-weakly-convex, $C_f$-Lipschitz continuous and non-decreasing;

- $g_i(\cdot)$ is $\rho_g$-weakly-convex and $C_g$-Lipschitz continuous. $h_{i,j}(\cdot)$ is differentiable and $C_h$-Lipschitz continuous.

- Either $g_i$ is monotone and $h_{i,j}(\cdot)$ is $L_h$-smooth, or $g_i$ is non-decreasing and $h_{i,j}(\cdot)$ is $L_h$-weakly-convex.

- Stochastic estimators $h_{i,j}(\mathbf{w}, \xi)$, $\partial h_{i,j}(\mathbf{w}, \xi)$ and $g_i(v_{i,j})$ have bounded variance $\sigma^2$, and $\|h_{i,j}(\mathbf{w})\| \le \tilde{C}_h$.

With coordinate moving average update, we present the following lemmas of error bounds.

**Algorithm 4** Stochastic Optimization algorithm for Non-smooth TCCO with coordinate moving average

1: Initialization: $\mathbf{w}_0$, $\{u_{i,0} : i \in \mathcal{S}_1\}$, $v_{i,j,0} = h_{i,j}(\mathbf{w}_0; \mathcal{B}^0_{3,i,j})$ for all $(i,j) \in \mathcal{S}_1 \times \mathcal{S}_2$.
2: **for** $t = 0, \ldots, T-1$ **do**
3:    Sample batches $\mathcal{B}^t_1 \subset \mathcal{S}_1$, $\mathcal{B}^t_2 \subset \mathcal{S}_2$, and $\mathcal{B}^t_{3,i,j} \subset \mathcal{D}_{i,j}$ for $i \in \mathcal{B}^t_1$ and $j \in \mathcal{B}^t_2$.
4:    $v_{i,j,t+1} = \begin{cases} (1-\tau_1)v_{i,j,t} + \tau_1 h_{i,j}(\mathbf{w}_t; \mathcal{B}^t_{3,i,j}), & (i,j) \in \mathcal{B}^t_1 \times \mathcal{B}^t_2 \\ v_{i,j,t}, & (i,j) \notin \mathcal{B}^t_1 \times \mathcal{B}^t_2 \end{cases}$
5:    $u_{i,t+1} = \begin{cases} (1-\tau_2)u_{i,t} + \frac{1}{B_2}\sum_{j \in \mathcal{B}^t_2} \tau_2 g_i(v_{i,j,t}), & i \in \mathcal{B}^t_1 \\ u_{i,t}, & i \notin \mathcal{B}^t_1 \end{cases}$
6:    $G_t = \frac{1}{B_1}\sum_{i \in \mathcal{B}^t_1}\left[\left(\frac{1}{B_2}\sum_{i \in \mathcal{B}^t_2} \nabla h_{i,j}(\mathbf{w}_t; \mathcal{B}^t_{3,i,j})\partial g_i(v_{i,j,t})\right)\partial f_i(u_{i,t})\right]$
7:    Update $\mathbf{w}_{t+1} = \mathbf{w}_t - \eta G_t$
8: **end for**
9: **return** $\mathbf{w}_{\bar{t}}$ with uniformly sampled $\bar{t} \in \{0, T-1\}$.

**Lemma B.5.** *Consider the coordinate moving average update for $\{v_{i,j,t} : (i,j) \in \mathcal{S}_1 \times \mathcal{S}_2\}$ in Algorithm 4, assume $h_{i,j}(\mathbf{w})$ is $C_h$-Lipschitz continuous for all $(i,j) \in \mathcal{S}_1 \times \mathcal{S}_2$ and $\tau_1 \leq 1$, then we have*

$$\mathbb{E}\left[\frac{1}{n_1 n_2}\sum_{i \in \mathcal{S}_1}\sum_{j \in \mathcal{S}_2}\|v_{i,j,t+1} - h_{i,j}(\mathbf{w}_{t+1})\|\right]$$

$$\leq (1 - \frac{B_1 B_2 \tau_1}{4n_1 n_2})^{t+1}\frac{1}{n_1 n_2}\sum_{i \in \mathcal{S}_1}\sum_{j \in \mathcal{S}_2}\|v_{i,j,0} - h_{i,j}(\mathbf{w}_0)\| + \frac{2\sqrt{2}\tau_1^{1/2}\sigma}{B_3^{1/2}} + \frac{4\sqrt{2}n_1 n_2 C_h M\eta}{B_1 B_2 \tau_1},$$

$$\mathbb{E}\left[\frac{1}{n_1 n_2}\sum_{i \in \mathcal{S}_1}\sum_{j \in \mathcal{S}_2}\|v_{i,j,t+1} - h_{i,j}(\mathbf{w}_{t+1})\|^2\right]$$

$$\leq (1 - \frac{B_1 B_2 \tau_1}{4n_1 n_2})^{2(t+1)}\frac{1}{n_1 n_2}\sum_{i \in \mathcal{S}_1}\sum_{j \in \mathcal{S}_2}\|v_{i,j,0} - h_{i,j}(\mathbf{w}_0)\|^2 + \frac{8\tau_1\sigma^2}{B_3} + \frac{32n_1^2 n_2^2 C_h^2 M^2 \eta^2}{B_1^2 B_2^2 \tau_1^2}.$$

**Lemma B.6.** *Consider the coordinate moving average update for $\{u_{i,t} : i \in \mathcal{S}_1\}$ in Algorithm 4, assume $g_i(\cdot)$ is $C_g$-Lipschitz continuous for all $i \in \mathcal{S}_1$ and $\tau_2 \leq 1$, then we have*

$$\mathbb{E}\left[\frac{1}{n_1}\sum_{i \in \mathcal{S}_1}\|u_{i,t+1} - \frac{1}{n_2}\sum_{j \in \mathcal{S}_2} g_i(v_{i,j,t+1})\|\right]$$

$$\leq (1 - \frac{B_1 \tau_2}{4n_1})^{t+1}\frac{1}{n_1}\sum_{i \in \mathcal{S}_1}\|u_{i,0} - \frac{1}{n_2}\sum_{j \in \mathcal{S}_2} g_i(v_{i,j,0})\| + \frac{2\sqrt{2}\tau_2^{1/2}\sigma}{B_2^{1/2}} + \frac{4\sqrt{2}C_g M n_1^{1/2} B_2^{1/2}\tau_1}{B_1^{1/2}n_2^{1/2}\tau_2},$$

$$\mathbb{E}\left[\frac{1}{n_1}\sum_{i \in \mathcal{S}_1}\|u_{i,t+1} - \frac{1}{n_2}\sum_{j \in \mathcal{S}_2} g_i(v_{i,j,t+1})\|^2\right]$$

$$\leq (1 - \frac{B_1 \tau_2}{4n_1})^{2(t+1)}\frac{1}{n_1}\sum_{i \in \mathcal{S}_1}\|u_{i,0} - \frac{1}{n_2}\sum_{j \in \mathcal{S}_2} g_i(v_{i,j,0})\|^2 + \frac{8\tau_2\sigma^2}{B_2} + \frac{32C_g^2 M^2 n_1 B_2 \tau_1^2}{B_1 n_2 \tau_2^2}.$$

Then we have a convergence analysis similar to Theorem A.2.

**Theorem B.7.** *Consider non-smooth weakly-convex TCCO problem, under Assumption B.4, setting* $\tau_1 = \mathcal{O}\left(\min\left\{B_3\epsilon^4, \frac{B_1^{1/2}n_2^{1/2}}{n_1^{1/2}B_2^{1/2}}B_2\epsilon^6\right\}\right) \leq 1$, $\tau_2 = \mathcal{O}(B_2\epsilon^4) \leq 1$, $\eta = \mathcal{O}\left(\min\left\{B_3\epsilon^4, \frac{B_1^{1/2}n_2^{1/2}}{n_1^{1/2}B_2^{1/2}}B_2\epsilon^6\right\}\frac{B_1 B_2}{n_1 n_2}\epsilon^2\right)$, *Algorithm 4 converges to an $\epsilon$-stationary point of the Moreau envelope $F_{1/\bar{\rho}}$ in* $T = \mathcal{O}\left(\max\left\{\frac{1}{B_3}, \frac{n_1^{1/2}}{B_1^{1/2}B_2^{1/2}n_2^{1/2}}\epsilon^{-2}\right\}\frac{n_1 n_2}{B_1 B_2}\epsilon^{-8}\right)$ *iterations.*

*Proof of Theorem B.7.* Since the only difference between SONT and Algorithm 4 is the update for $\{u_{i,t} : i \in \mathcal{S}_1\}$ and $\{v_{i,j,t} : (i,j) \in \mathcal{S}_1 \times \mathcal{S}_2\}$, the proof of Theorem A.2 still holds with the error bound replaced by Lemma B.5 and Lemma B.6, i.e.,

$$\frac{1}{n_1 n_2} \sum_{i \in S_1} \sum_{j \in S_2} \mathbb{E}[\|h_{i,j}(\mathbf{w}_t) - v_{i,j,t}\|] \leq (1 - \mu_1)^t \frac{1}{n_1 n_2} \sum_{i \in S_1} \sum_{j \in S_2} \|h_{i,j}(\mathbf{w}_0) - v_{i,j,0}\| + R_1,$$

$$\frac{1}{n_1 n_2} \sum_{i \in S_1} \sum_{j \in S_2} \mathbb{E}[\|h_{i,j}(\mathbf{w}_t) - v_{i,j,t}\|^2] \leq (1 - \mu_1)^t \frac{1}{n_1 n_2} \sum_{i \in S_1} \sum_{j \in S_2} \|h_{i,j}(\mathbf{w}_0) - v_{i,j,0}\|^2 + R_2,$$

$$\frac{1}{n_1} \sum_{i \in S_1} \mathbb{E}\left[\left\|\frac{1}{n_2} \sum_{j \in S_2} g_i(v_{i,j,t}) - u_{i,t}\right\|\right] \leq (1 - \mu_2)^t \frac{1}{n_+} \sum_{i \in S_+} \left\|\frac{1}{n_2} \sum_{j \in S_2} g_i(v_{i,j,0}) - u_{i,0}\right\| + R_3,$$

$$\frac{1}{n_1} \sum_{i \in S_1} \mathbb{E}\left[\left\|\frac{1}{n_2} \sum_{j \in S_2} g_i(v_{i,j,t}) - u_{i,t}\right\|^2\right] \leq (1 - \mu_2)^t \frac{1}{n_+} \sum_{i \in S_+} \left\|\frac{1}{n_2} \sum_{j \in S_2} g_i(v_{i,j,0}) - u_{i,0}\right\|^2 + R_4,$$

with

$$\mu_1 = \frac{B_1 B_2 \tau_1}{4 n_1 n_2}, \quad \mu_2 = \frac{B_1 \tau_2}{4 n_1}, \quad R_1 = \frac{2\sqrt{2}\tau_1^{1/2}\sigma}{B_3^{1/2}} + \frac{4\sqrt{2}n_1 n_2 C_h M \eta}{B_1 B_2 \tau_1}$$

$$R_2 = \frac{8\tau_1 \sigma^2}{B_3} + \frac{32 n_1^2 n_2^2 C_h^2 M^2 \eta^2}{B_1^2 B_2^2 \tau_1^2}, \quad R_3 = \frac{2\sqrt{2}\tau_2^{1/2}\sigma}{B_2^{1/2}} + \frac{4\sqrt{2}C_g M n_1^{1/2} B_2^{1/2} \tau_1}{B_1^{1/2} n_2^{1/2} \tau_2},$$

$$R_4 = \frac{8\tau_2 \sigma^2}{B_2} + \frac{32 C_g^2 M^2 n_1 B_2 \tau_1^2}{B_1 n_2 \tau_2^2}$$

Then the proof proceeds to

$$\frac{1}{T} \sum_{t=0}^{T-1} \mathbb{E}[\|\nabla F_{1/\bar{\rho}}(\mathbf{w}_t)\|^2]$$

$$\leq \mathcal{O}\left(\frac{1}{T}(\frac{1}{\eta} + \frac{1}{\mu_{min}}) + \eta + R_1 + R_2 + R_3 + R_4\right)$$

$$\leq \mathcal{O}\left(\frac{1}{T}(\frac{1}{\eta} + \frac{1}{\mu_{min}}) + \eta + \frac{\tau_1^{1/2}\sigma}{B_3^{1/2}} + \frac{\tau_1 \sigma^2}{B_3} + \frac{\tau_2^{1/2}\sigma}{B_2^{1/2}} + \frac{\tau_2 \sigma^2}{B_2} + \frac{n_1 n_2 \eta}{B_1 B_2 \tau_1} + \frac{n_1^2 n_2^2 \eta^2}{B_1^2 B_2^2 \tau_1^2}\right.$$

$$\left. + \frac{n_1^{1/2} B_2^{1/2} \tau_1}{B_1^{1/2} n_2^{1/2} \tau_2} + \frac{n_1 B_2 \tau_1^2}{B_1 n_2 \tau_2^2}\right)$$

$$\leq \mathcal{O}\left(\frac{1}{T}(\frac{1}{\eta} + \frac{1}{\mu_{min}}) + \frac{\tau_1^{1/2}\sigma}{B_3^{1/2}} + \frac{\tau_2^{1/2}\sigma}{B_2^{1/2}} + \frac{n_1 n_2 \eta}{B_1 B_2 \tau_1} + \frac{n_1^{1/2} B_2^{1/2} \tau_1}{B_1^{1/2} n_2^{1/2} \tau_2}\right).$$

Setting

$$\tau_1 = \mathcal{O}\left(\min\left\{B_3 \epsilon^4, \frac{B_1^{1/2} n_2^{1/2}}{n_1^{1/2} B_2^{1/2}} B_2 \epsilon^6\right\}\right), \quad \tau_2 = \mathcal{O}(B_2 \epsilon^4),$$

$$\eta = \mathcal{O}\left(\min\left\{B_3 \epsilon^4, \frac{B_1^{1/2} n_2^{1/2}}{n_1^{1/2} B_2^{1/2}} B_2 \epsilon^6\right\} \frac{B_1 B_2}{n_1 n_2} \epsilon^2\right),$$

then to reach a nearly $\epsilon$-stationary point, Algorithm 4 need

$$T = \mathcal{O}\left(\max\left\{\frac{1}{B_3}, \frac{n_1^{1/2}}{B_1^{1/2} B_2^{1/2} n_2^{1/2}} \epsilon^{-2}\right\} \frac{n_1 n_2}{B_1 B_2} \epsilon^{-10}\right)$$

iterations. $\qquad\square$

# C Details for TPAUC Maximization

## C.1 Assumption Verification

We first present two lemmas about the weak convexity of the objective in the regular learning setting and in the multi-instance learning setting with mean pooling.

**Lemma C.1.** *Consider the formulation in problem (9) in the regular learning setting and assume that function $\ell(\cdot)$ is non-decreasing, $C_\ell$ Lipschitz continuous and $\rho_\ell$-weakly-convex, and function $h_\mathbf{w}(X_i)$ is $C_h$ Lipschitz continuous and $\rho_h$-weakly-convex. then the following statements are true:*

- $f_i(g, s')$ *is convex and $C_f$-Lipschitz continuous w.r.t. $(g, s')$, and non-decreasing w.r.t. $g$.*

- $\psi_i(\mathbf{w}, s_i)$ *is $\rho_\psi$-weakly-convex w.r.t. $(\mathbf{w}, s_i)$, and the stochastic estimator of the finite sum function value $\psi_i(\mathbf{w}, s_i)$ is $C_\psi$-Lipschitz continuous w.r.t. $(\mathbf{w}, s_i)$.*

- $\frac{1}{n_+} \sum_{i \in \mathcal{S}_+} f_i(\psi_i(\mathbf{w}, s_i), s')$ *is $\rho_F$-weakly-convex w.r.t. $(\mathbf{w}, \mathbf{s}, s')$.*

**Lemma C.2.** *Consider the formulation in problem (9) in the multi-instance learning setting with mean pooling, and assume that function $h_i(\mathbf{w}) = \frac{1}{|X_i|} \sum_{\mathbf{x} \in X_i} e(\mathbf{w}_e; \mathbf{x})^\top \mathbf{w}_c$ is $\tilde{L}_h$-smooth and is bounded by $\tilde{C}_h$, and $h_i(\mathbf{w}; \xi) = e(\mathbf{w}_e; \xi)^\top \mathbf{w}_c$ is $C_h$-Lipschitz continuous and has bounded variance $\sigma^2$, $\ell$ is non-decreasing and $L_\ell$-weakly-convex, then the followings are true:*

- $f_i(g, s')$ *is convex and $C_f$-Lipschitz-continuous w.r.t. $(g, s')$, and non-decreasing w.r.t. $g$;*

- $g_i(v, s_i) = s_i + \frac{(\ell(v) - s_i)_+}{\beta}$ *is $\rho_g$-weakly convex and non-decreasing w.r.t. $v$, convex w.r.t. $s_i$, and $C_g$-Lipschitz continuous w.r.t. $(v, s_i)$;*

- $h_{i,j}(\mathbf{w}) = h_j(\mathbf{w}) - h_i(\mathbf{w})$ *is $L_h$-weakly-convex, and $h_{i,j}(\mathbf{w}; \xi, \zeta)$ is $C_h$-Lipschitz continuous;*

- $\frac{1}{n_+} \sum_{X_i \in \mathcal{S}_+} f_i(g_i(h_{i,j}(\mathbf{w}), s_i), s')$ *is $\rho_F$-weakly-convex w.r.t. $(\mathbf{w}, \mathbf{s}, s')$.*

### C.1.1 Proof of Lemma C.1

*Proof of Lemma C.1.* The convexity of $f_i(g, s')$ with respect to $(g, s')$ follows from the convexity definition. With subgradients $\partial_{s'} f_i(g, s') \in [1 - \frac{1}{\alpha}, 1]$, $\partial_g f_i(g, s') \in [0, \frac{1}{\alpha}]$, we can see that $f_i(g, s')$ is $\frac{1}{\alpha}$-Lipschitz continuous w.r.t. $(g, s')$, and non-decreasing w.r.t. $u$.

We first show that $\ell(h_\mathbf{w}(X_j) - h_\mathbf{w}(X_i))$ is weakly-convex w.r.t. $\mathbf{w}$.

$\ell(h_{\tilde{\mathbf{w}}}(X_j) - h_{\tilde{\mathbf{w}}}(X_i))$

$\geq \ell(h_\mathbf{w}(X_j) - h_\mathbf{w}(X_i)) + \langle \partial \ell(h_\mathbf{w}(X_j) - h_\mathbf{w}(X_i)), (h_{\tilde{\mathbf{w}}}(X_j) - h_{\tilde{\mathbf{w}}}(X_i)) - (h_\mathbf{w}(X_j) - h_\mathbf{w}(X_i)) \rangle$

$\quad + \frac{\rho_\ell}{2} \| (h_{\tilde{\mathbf{w}}}(X_j) - h_{\tilde{\mathbf{w}}}(X_i)) - (h_\mathbf{w}(X_j) - h_\mathbf{w}(X_i)) \|^2$

$\overset{(a)}{\geq} \ell(h_\mathbf{w}(X_j) - h_\mathbf{w}(X_i)) + \langle \partial \ell(h_\mathbf{w}(X_j) - h_\mathbf{w}(X_i)), \langle \nabla h_\mathbf{w}(X_j) - \nabla h_\mathbf{w}(X_i), \tilde{\mathbf{w}} - \mathbf{w} \rangle \rangle$

$\quad + 2\rho_\ell C_h^2 \| \tilde{\mathbf{w}} - \mathbf{w} \|^2$

where (a) uses the weak-convexity of $h_\mathbf{w}(X_i)$ and $h_\mathbf{w}(X_j)$,

$$h_{\tilde{\mathbf{w}}}(X_j) - h_\mathbf{w}(X_j) \geq \langle \nabla h_\mathbf{w}(X_j), \tilde{\mathbf{w}} - \mathbf{w} \rangle - \frac{\rho_h}{2} \| \tilde{\mathbf{w}} - \mathbf{w} \|^2,$$

$$- h_{\tilde{\mathbf{w}}}(X_i) + h_\mathbf{w}(X_i) \geq - \langle \nabla h_\mathbf{w}(X_i), \tilde{\mathbf{w}} - \mathbf{w} \rangle + \frac{\rho_h}{2} \| \tilde{\mathbf{w}} - \mathbf{w} \|^2.$$

Thus $\ell(h_\mathbf{w}(X_j) - h_\mathbf{w}(X_i))$ is $4\rho_\ell C_h^2$-weakly-convex w.r.t. $\mathbf{w}$.

By convexity of $(\ell, s_i) \mapsto s_i + \frac{(\ell - s_i)_+}{\beta}$, we have

$$\psi_i(\tilde{\mathbf{w}}, \tilde{s}_i)$$

$$\geq \psi_i(\mathbf{w}, s_i) + \langle \partial_\ell \psi_i(\mathbf{w}, s_i), \ell(h_{\tilde{\mathbf{w}}}(X_j) - h_{\tilde{\mathbf{w}}}(X_i)) - \ell(h_{\mathbf{w}}(X_j) - h_{\mathbf{w}}(X_i)) \rangle + \langle \partial_{s_i} \psi_i(\mathbf{w}, s_i), \tilde{s}_i - s_i \rangle$$

$$\overset{(a)}{\geq} \psi_i(\mathbf{w}, s_i) + \partial_\ell \psi_i(\mathbf{w}, s_i) \left[ \partial \ell(h_{\mathbf{w}}(X_j) - h_{\mathbf{w}}(X_i)) \langle \nabla h_{\mathbf{w}}(X_j) - \nabla h_{\mathbf{w}}(X_i), \tilde{\mathbf{w}} - \mathbf{w} \rangle - 2\rho_\ell C_h^2 \|\tilde{\mathbf{w}} - \mathbf{w}\|^2 \right]$$

$$+ \langle \partial_{s_i} \psi_i(\mathbf{w}, s_i), \tilde{s}_i - s_i \rangle$$

$$\overset{(b)}{\geq} \psi_i(\mathbf{w}, s_i) + \partial_\ell \psi_i(\mathbf{w}, s_i) \partial \ell(h_{\mathbf{w}}(X_j) - h_{\mathbf{w}}(X_i)) \langle \nabla h_{\mathbf{w}}(X_j) - \nabla h_{\mathbf{w}}(X_i), \tilde{\mathbf{w}} - \mathbf{w} \rangle$$

$$+ \langle \partial_{s_i} \psi_i(\mathbf{w}, s_i), \tilde{s}_i - s_i \rangle - \frac{2\rho_\ell C_h^2}{\beta} \|\tilde{\mathbf{w}} - \mathbf{w}\|^2$$

where (a) follows from the monotonicity of $\psi_i$ w.r.t. $\ell$ and weak-convexity of $\ell(h_{\mathbf{w}}(X_j) - h_{\mathbf{w}}(X_i))$, and (b) is due to the Lipschitz continuity of $(\ell, s_i) \mapsto s_i + \frac{(\ell - s_i)_+}{\beta}$ w.r.t. $\ell$. Thus $\psi_i$ is $\frac{4\rho_\ell C_h^2}{\beta}$-weakly-convex w.r.t. $(\mathbf{w}, s_i)$.

With a similar argument using the convexity and Lipschitz continuity of $f_i(g, s')$ w.r.t. $(g, s')$ and the weak-convexity of $\psi_i(\mathbf{w}, s_i)$, we can show that $f_i(\psi_i(\mathbf{w}, s_i), s')$ is $\frac{4\rho_\ell C_h^2}{\beta}$-weakly-convex w.r.t. $(\mathbf{w}, s_i, s')$. Thus, $F(\mathbf{w}, s_i, s')$ is $\rho_F = \frac{4\rho_\ell C_h^2}{\beta}$-weakly-convex w.r.t. $(\mathbf{w}, \mathbf{s}, s')$.

Now we show the Lipschitz continuity of $\psi_i(\mathbf{w}, s_i; X_j)$, i.e. an unbiased stochastic estimator of $\psi_i(\mathbf{w}, s_i)$. We have

$$\|\psi_i(\mathbf{w}, s_i; X_j) - \psi_i(\tilde{\mathbf{w}}, \tilde{s}_i; X_j)\|^2$$

$$= \left\| (s_i + \frac{(\ell(h_{\mathbf{w}}(X_j) - h_{\mathbf{w}}(X_i)) - s_i)_+}{\beta}) - (\tilde{s}_i + \frac{(\ell(h_{\tilde{\mathbf{w}}}(X_j) - h_{\tilde{\mathbf{w}}}(X_i)) - \tilde{s}_i)_+}{\beta}) \right\|^2$$

$$\leq 2\|s_i - \tilde{s}_i\|^2 + 2 \left\| \frac{(\ell(h_{\mathbf{w}}(X_j) - h_{\mathbf{w}}(X_i)) - s_i)_+}{\beta} - \frac{(\ell(h_{\tilde{\mathbf{w}}}(X_j) - h_{\tilde{\mathbf{w}}}(X_i)) - \tilde{s}_i)_+}{\beta} \right\|^2$$

$$\leq 2\|s_i - \tilde{s}_i\|^2 + \frac{2}{\beta^2} (8C_\ell^2 C_h^2 \|\tilde{\mathbf{w}} - \mathbf{w}\|^2 + 2\|\tilde{s}_i - s_i\|^2)$$

$$\leq (2 + \frac{4 + 16C_\ell^2 C_h^2}{\beta^2})(\|\tilde{\mathbf{w}} - \mathbf{w}\|^2 + \|\tilde{s}_i - s_i\|^2).$$

Thus $\psi_i(\mathbf{w}, s_i; X_j)$ is $(2 + \frac{4 + 16C_\ell^2 C_h^2}{\beta^2})^{1/2}$-Lipschitz continuous w.r.t. $(\mathbf{w}, s_i)$. $\qquad\square$

### C.1.2 Proof of Lemma C.2

*Proof of Lemma C.2.* First of all, the convexity of $f_i(u, s')$ w.r.t. $(u, s')$ and the convexity of $g_i(v_{ij}, s_i)$ w.r.t. $(\ell, s_i)$ directly follows from the convexity definition. Moreover, one can see from the formulation that $\partial_{s'} f_i(g, s') \in [1 - \frac{1}{\alpha}, 1]$, $\partial_u f_i(g, s') \in [0, \frac{1}{\alpha}]$, $\partial_\ell g_i(v_{ij}, s_i) \in [1 - \frac{1}{\beta}, 1]$, $\partial_{s_i} g_i(v_{ij}, s_i) \in [0, \frac{1}{\beta}]$. Thus $f_i$ is $C_f = \frac{1}{\alpha}$-Lipschitz continuous w.r.t. $(u, s')$ and non-decreasing w.r.t. $u$, $g_i$ is $\frac{1}{\beta}$-Lipschitz continuous w.r.t. $(\ell, s_i)$ and non-decreasing w.r.t. $\ell$. Since $\ell(\cdot)$ is non-decreasing, $g_i(v_{ij}, s_i)$ is non-decreasing w.r.t. $v_{ij}$. As a result of Proposition 4.2, $g_i(v_{ij}, s_i)$ is $\rho_g = \frac{1}{\beta} L_\ell$-weakly-convex w.r.t. $v_{ij}$. Due to the composition structure and the Lipschitz continuity of $g_i$ and $\ell$, one can see that $g_i(v_{ij}, s_i)$ is $C_g = \frac{1}{\beta} C_\ell$-Lipschitz continuous w.r.t. $(v_{ij}, s_i)$.

The $L_h = 2\tilde{L}_h$-weakly-convexity of $h_{i,j}(\mathbf{w})$ and $C_h = 2\tilde{C}_h$-Lipschitz continuity of $h_{i,j}(\mathbf{w}; \xi, \zeta)$ directly follows from the $\tilde{L}_h$-smoothness of $h_i(\mathbf{w})$ and $\tilde{C}_h$-Lipschitz continuity of $h_i(\mathbf{w}; \xi)$. Finally,

we show the weakly-convexity of $f_i(g_i(h_{i,j}(\mathbf{w}), s_i), s')$:

$$f_i(g_i(h_{i,j}(\tilde{\mathbf{w}}), \tilde{s}_i)\tilde{s}')$$

$$\overset{(a)}{\geq} f_i(g_i(h_{i,j}(\mathbf{w}), s_i), s') + \langle \partial_{s'} f_i(g_i(h_{i,j}(\mathbf{w}), s_i), s'), \tilde{s}' - s' \rangle$$
$$+ \langle \partial_u f_i(g_i(h_{i,j}(\mathbf{w}), s_i), s'), g_i(h_{i,j}(\tilde{\mathbf{w}}), \tilde{s}_i) - g_i(h_{i,j}(\mathbf{w}), s_i) \rangle$$

$$\overset{(b)}{\geq} f_i(g_i(h_{i,j}(\mathbf{w}), s_i), s') + \langle \partial_{s'} f_i(g_i(h_{i,j}(\mathbf{w}), s_i), s'), \tilde{s}' - s' \rangle$$
$$+ \langle \partial_u f_i(g_i(h_{i,j}(\mathbf{w}), s_i), s'), \langle \partial_\ell g_i(h_{i,j}(\mathbf{w}), s_i), \ell(h_{i,j}(\tilde{\mathbf{w}})) - \ell(h_{i,j}(\mathbf{w})) \rangle \rangle$$
$$+ \langle \partial_u f_i(g_i(h_{i,j}(\mathbf{w}), s_i)s'), \langle \partial_{s_i} g_i(h_{i,j}(\mathbf{w}), s_i), \tilde{s}_i - s_i \rangle \rangle$$

$$\overset{(c)}{\geq} f_i(g_i(h_{i,j}(\mathbf{w}), s_i), s') + \langle \partial_{s'} f_i(g_i(h_{i,j}(\mathbf{w}), s_i), s'), \tilde{s}' - s' \rangle$$
$$+ \langle \partial_u f_i(g_i(h_{i,j}(\mathbf{w}), s_i), s') \partial_\ell g_i(h_{i,j}(\mathbf{w}), s_i) \partial \ell(h_{i,j}(\mathbf{w})), h_{i,j}(\tilde{\mathbf{w}}) - h_{i,j}(\mathbf{w}) \rangle$$
$$- \frac{C_f C_g L_\ell}{2} \| h_{i,j}(\tilde{\mathbf{w}}) - h_{i,j}(\mathbf{w}) \|^2 + \langle \partial_u f_i(s', g_i(h_{i,j}(\mathbf{w}), s_i)) \partial_{s_i} g_i(h_{i,j}(\mathbf{w}), s_i), \tilde{s}_i - s_i \rangle$$

$$\overset{(d)}{\geq} f_i(g_i(h_{i,j}(\mathbf{w}), s_i), s') + \langle \partial_{s'} f_i(g_i(h_{i,j}(\mathbf{w}), s_i), s'), \tilde{s}' - s' \rangle$$
$$+ \langle \partial_u f_i(g_i(h_{i,j}(\mathbf{w}), s_i), s') \partial_\ell g_i(h_{i,j}(\mathbf{w}), s_i) \partial \ell(h_{i,j}(\mathbf{w})) \nabla h_{i,j}(\mathbf{w}), \tilde{\mathbf{w}} - \mathbf{w} \rangle$$
$$+ \langle \partial_u f_i(g_i(h_{i,j}(\mathbf{w}), s_i), s') \partial_{s_i} g_i(h_{i,j}(\mathbf{w}), s_i), \tilde{s}_i - s_i \rangle - \left( \frac{C_f C_g C_h^2 L_\ell}{2} + \frac{C_f C_g L_h}{2} \right) \| \tilde{\mathbf{w}} - \mathbf{w} \|^2$$

where (a) uses the convexity of $f_i$, (b) uses the monotonicity of $f_i$ w.r.t. $u$ and convexity of $g_i(\ell, s_i)$ w.r.t. $(\ell, s_i)$, (c) uses monotonicity of $f_i$ w.r.t. $u$, monotonicity of $g_i$ w.r.t. $\ell$ and $L_\ell$-weak-convexity of $\ell$, (d) uses the smoothness of $h_{i,j}$. Thus $f_i(g_i(h_{i,j}(\mathbf{w}), s_i), s')$ is $\rho_F = (C_f C_g C_h^2 L_\ell + C_f C_g L_h)$-weakly-convex w.r.t. $(\mathbf{w}, s_i, s')$. Therefore, $\frac{1}{n_+} \sum_{i \in \mathcal{S}+} f_i(g_i(h_{i,j}(\mathbf{w}), s_i), s')$ is $\rho_F$-weakly-convex w.r.t. $(\mathbf{w}, \mathbf{s}, s')$. $\qquad\square$

## C.2 Algorithms for TPAUC and Multi-instance TPAUC Maximization

---
**Algorithm 5** SONX for TPAUC
---
1: Initialization: $\mathbf{w}_0, \{u_{i,0} : i \in \mathcal{S}_+\}, \{s_{i,0} : i \in \mathcal{S}_+\}, s'_0$
2: **for** $t = 0, \dots, T-1$ **do**
3:      Sample batches $\mathcal{B}_1^t \subset S_+$ and $\mathcal{B}_2^t \subset S_-$.
4:      $u_{i,t+1} = \begin{cases} (1-\tau)u_{i,t} + \tau \psi_i(\mathbf{w}_t, s_{i,t}; \mathcal{B}_2^t) + \gamma(\psi_i(\mathbf{w}_t, s_{i,t}; \mathcal{B}_2^t) - \psi_i(\mathbf{w}_{t-1}, s_{i,t-1}; \mathcal{B}_2^t)), \ i \in \mathcal{B}_1^t \\ u_{i,t}, \quad i \notin \mathcal{B}_1^t \end{cases}$
5:      $s_{i,t+1} = \begin{cases} s_{i,t} - \eta \frac{1}{B_1} \partial_s \psi_i(\mathbf{w}_t, s_{i,t}; \mathcal{B}_2^t) \partial_u f(u_{i,t}, s'_t), \quad i \in \mathcal{B}_1^t \\ s_{i,t}, \quad i \notin \mathcal{B}_1^t \end{cases}$
6:      $s'_{t+1} = s'_t - \eta \frac{1}{B_1} \sum_{i \in \mathcal{B}_1^t} \partial_{s'} f(u_{i,t}, s'_t)$
7:      Compute $G_t = \frac{1}{B_1} \sum_{i \in \mathcal{B}_1^t} \partial_w \psi_i(\mathbf{w}_t, s_{i,t}; \mathcal{B}_2^t) \partial_u f(u_{i,t}, s'_t)$
8:      Update $\mathbf{w}_{t+1} = \mathbf{w}_t - \eta G_t$
9: **end for**
10: **return** $\mathbf{w}_{\bar{t}}$ with $\bar{t}$ uniformly sampled from $\{0, \dots, T-1\}$.
---

**Algorithm 6** SONT for Multi-instance TPAUC

---
1: Initialization: $\mathbf{w}_0, \{u_{i,0} : i \in \mathcal{S}_+\}, \{s_{i,0} : i \in \mathcal{S}_+\}, s_0', \{v_{i,j,0} : (i,j) \in \mathcal{S}_+ \times \mathcal{S}_-\}$
2: **for** $t = 0, \ldots, T-1$ **do**
3:     Sample batches $\mathcal{B}_1^t \subset S_+$, $\mathcal{B}_2^t \subset S_-$, and $\mathcal{B}_{3,i}^t \subset X_i$ for $i \in \mathcal{B}_1^t \cup \mathcal{B}_2^t$.
4:     $v_{i,t+1} = \begin{cases} \Pi_{\tilde{C}_h}[(1-\tau_1)v_{i,t} + \tau_1 h_i(\mathbf{w}_t; \mathcal{B}_{3,i}^t) + \gamma_1(h_i(\mathbf{w}_t; \mathcal{B}_{3,i}^t) - h_i(\mathbf{w}_{t-1}; \mathcal{B}_{3,i}^t))], & i \in \mathcal{B}_1^t \\ \Pi_{\tilde{C}_h}[(1-\tau_1)v_{i,t} + \tau_1 h_i(\mathbf{w}_t; \mathcal{B}_{3,i}^t) + \gamma_2(h_i(\mathbf{w}_t; \mathcal{B}_{3,i}^t) - h_i(\mathbf{w}_{t-1}; \mathcal{B}_{3,i}^t))], & i \in \mathcal{B}_2^t \\ v_{i,t}, \quad i \notin \mathcal{B}_1^t \text{ and } i \notin \mathcal{B}_2^t \end{cases}$
5:     $u_{i,t+1} = \begin{cases} (1-\tau_2)u_{i,t} + \frac{1}{B_2}\sum_{j \in \mathcal{B}_2^t}[\tau_2 g(v_{j,t} - v_{i,t}, s_{i,t}) \\ \qquad + \gamma_3(g(v_{j,t} - v_{i,t}, s_{i,t}) - g(v_{j,t-1} - v_{i,t-1}, s_{i,t-1}))], & i \in \mathcal{B}_1^t \\ u_{i,t}, & i \notin \mathcal{B}_1^t \end{cases}$
6:     $s_{i,t+1} = \begin{cases} s_{i,t} - \eta_1 \frac{1}{B_1}\left[\frac{1}{B_2}\sum_{j \in \mathcal{B}_2^t} \partial_{s_i} g(v_{j,t} - v_{i,t}, s_{i,t})\right] \partial_u f(s_t', u_{i,t}), & i \in \mathcal{B}_1^t \\ s_{i,t}, & i \notin \mathcal{B}_1^t \end{cases}$
7:     $s_{t+1}' = s_t' - \eta_2 \frac{1}{B_1}\sum_{i \in \mathcal{B}_1^t} \partial_{s'} f(u_{i,t}, s_t')$
8:     $G_t = \frac{1}{B_1}\sum_{i \in \mathcal{B}_1^t} \partial_u f(u_{i,t}, s_t')$
9:     $\left[\frac{1}{B_2}\sum_{j \in \mathcal{B}_2^t}\left(\nabla h_j(\mathbf{w}_t; \mathcal{B}_{3,j}^t) - \nabla h_i(\mathbf{w}_t; \mathcal{B}_{3,i}^t)\right) \partial_v g(v_{j,t} - v_{i,t}, s_{i,t})\right]$
10:    Update $\mathbf{w}_{t+1} = \mathbf{w}_t - \eta G_t$
11: **end for**
12: **return** $\mathbf{w}_{\bar{t}}$ with $\bar{t}$ uniformly sampled from $\{0, \ldots, T-1\}$.

---

### C.3   TPAUC in MIL with smoothed-max pooling and attention-based pooling

We can extend our results to smoothed-max pooling and attention-based pooling.

**Smoothed-max Pooling.** The smoothed-max pooling can be written as [45]:

$$h_{\mathbf{w}}(X) = \tau \log\left(\frac{1}{|X|}\sum_{\mathbf{x} \in X} \exp(\phi(\mathbf{w}; \mathbf{x})/\tau)\right), \tag{24}$$

where $\tau > 0$ is a hyperparameter and $\phi(\mathbf{w}; \mathbf{x}) = e(\mathbf{w}_e, \mathbf{x})^\top \mathbf{w}_c$ is the prediction score for instance $\mathbf{x}$.

We can see that $h_{\mathbf{w}}(X)$ itself is a compositional function. To map the problem into TCCO, we define $h_i(\mathbf{w}) = \frac{1}{|X_i|}\sum_{\mathbf{x} \in X_i} \exp(\phi(\mathbf{w}; \mathbf{x})/\tau) + C$, where $C > 0$ is a constant. Then the objective function becomes

$$\min_{\mathbf{w}, s', \mathbf{s}} \frac{1}{n_+}\sum_{X_i \in \mathcal{S}_+} f_i(\psi_i(\mathbf{w}, s_i), s'),$$

$$\text{where } f_i(g, s') = s' + \frac{(g - s')_+}{\alpha}, \tag{25}$$

$$\psi_i(\mathbf{w}, s_i) = \frac{1}{n_-}\sum_{X_j \in \mathcal{S}_-} s_i + \frac{(\ell(\tau \log h_j(\mathbf{w}) - \tau \log h_i(\mathbf{w})) - s_i)_+}{\beta},$$

In this case we define $g_i(\ell(\mathbf{v}), s_i) = s_i + \frac{(\ell(\tau \log v_1 - \tau \log v_2) - s_i)_+}{\beta}$ and $h_{i,j}(\mathbf{w}) = [h_i(\mathbf{w}), h_j(\mathbf{w})]$. We can still prove that $g_i(\ell(\mathbf{v}), s_i)$ is monotone w.r.t to each component of $\mathbf{v}$. It is not difficult to prove that $\ell(\tau \log v_1 - \tau \log v_2)$ is weakly convex w.r.t $\mathbf{v}$ because $\tau \log v_1 - \tau \log v_2$ is a smooth mapping of $\mathbf{v}$ due to $\mathbf{v} \geq C$ and $\ell$ is a convex function [8]. As a result, since $g_i(\ell, s_i)$ is non-decreasing and convex w.r.t to $\ell$, it is easy to prove that $g_i(\ell(\mathbf{v}), s_i)$ is weakly convex w.r.t $\mathbf{v}$ and is monotone (either non-decreasing or non-increasing) w.r.t to each component of $\mathbf{v}$. Hence, assuming $h_i(\mathbf{w})$ is a smooth and Lipchitz continuous function, we can prove that $g_i(h_{i,j}(\mathbf{w}), s_i)$ is weakly convex w.r.t. to $\mathbf{w}$.

**Attention-based Pooling.** Attention-based pooling was recently introduced for deep MIL [14], which aggregates the feature representations using attention, i.e.,

$$E(\mathbf{w}; X) = \sum_{\mathbf{x} \in X} \frac{\exp(g(\mathbf{w}; \mathbf{x}))}{\sum_{\mathbf{x}' \in X} \exp(g(\mathbf{w}; \mathbf{x}'))} e(\mathbf{w}_e; \mathbf{x}) \tag{26}$$

where $g(\mathbf{w}; \mathbf{x})$ is a parametric function, e.g., $g(\mathbf{w}; \mathbf{x}) = \mathbf{w}_a^\top \tanh(Ve(\mathbf{w}_e; \mathbf{x})) + C$, where $V \in \mathbb{R}^{m \times d_o}$ and $\mathbf{w}_a \in \mathbb{R}^m$. Based on the aggregated feature representation, the bag level prediction can be computed by

$$h_{\mathbf{w}}(\mathbf{w}, X) = (\mathbf{w}_c^\top E(\mathbf{w}; X)) \tag{27}$$

$$= \left( \sum_{\mathbf{x} \in X} \frac{\exp(g(\mathbf{w}; \mathbf{x}))\delta(\mathbf{w}; \mathbf{x})}{\sum_{\mathbf{x}' \in X} \exp(g(\mathbf{w}; \mathbf{x}'))} \right),$$

where $\delta(\mathbf{w}; \mathbf{x}) = \mathbf{w}_c^\top e(\mathbf{w}_e; \mathbf{x})$.

We can see that $h_{\mathbf{w}}(X)$ itself is a compositional function. To map the problem into TCCO, we define $h_i^1(\mathbf{w}) = \frac{1}{|X_i|} \sum_{\mathbf{x} \in X_i} \exp(g(\mathbf{w}; \mathbf{x}))\delta(\mathbf{w}; \mathbf{x})$, and $h_i^2(\mathbf{w}) = \frac{1}{|X_i|} \sum_{\mathbf{x}' \in X_i} \exp(g(\mathbf{w}; \mathbf{x}'))$. Assume $|\mathbf{w}_a^\top \tanh(Ve(\mathbf{w}_e; \mathbf{x}))| \leq C_b$ then $h_i^2(\mathbf{w}) \geq \exp(C - C_b)$. Then the objective function becomes

$$\min_{\mathbf{w}, \mathbf{s}', \mathbf{s}} \frac{1}{n_+} \sum_{X_i \in \mathcal{S}_+} f_i(\psi_i(\mathbf{w}, s_i), s'),$$

where $f_i(g, s') = s' + \frac{(g - s')_+}{\alpha}$, $\quad \psi_i(\mathbf{w}, s_i) = \frac{1}{n_-} \sum_{X_j \in \mathcal{S}_-} s_i + \frac{(\ell(\frac{h_j^1(\mathbf{w})}{h_j^2(\mathbf{w})} - \frac{h_i^1(\mathbf{w})}{h_i^2(\mathbf{w})}) - s_i)_+}{\beta},$

$$\tag{28}$$

In this case we define $g_i(\ell(\mathbf{v}), s_i) = s_i + \frac{(\ell(\frac{v_3}{v_4} - \frac{v_1}{v_2}) - s_i)_+}{\beta}$ and $h_{i,j}(\mathbf{w}) = [h_i^1(\mathbf{w}), h_i^2(\mathbf{w}), h_j^1(\mathbf{w}), h_j^2(\mathbf{w})]$. We can still prove that $g_i(\ell(\mathbf{v}), s_i)$ is monotone w.r.t to each component of $\mathbf{v}$. It is not difficult to prove that $\ell(\frac{v_3}{v_4} - \frac{v_1}{v_2})$ is weakly convex w.r.t $\mathbf{v}$ because $\frac{v_3}{v_4} - \frac{v_1}{v_2}$ is a smooth mapping of $\mathbf{v}$ when $v_2, v_4$ are lower bounded and $\ell$ is a convex function [8]. As a result, since $g_i(\ell, s_i)$ is non-decreasing and convex w.r.t to $\ell$, it is easy to prove that $g_i(\ell(\mathbf{v}), s_i)$ is weakly convex w.r.t $\mathbf{v}$ and is monotone (either non-decreasing or non-increasing) w.r.t to each component of $\mathbf{v}$. Hence, assuming $h_i^1(\mathbf{w}), h_i^2(\mathbf{w})$ are smooth and Lipchitz continuous, we can prove that $g_i(h_{i,j}(\mathbf{w}), s_i)$ is weakly convex w.r.t. to $\mathbf{w}$.

### C.4 Convergence Analysis of TPAUC Maximization

#### C.4.1 Convergence analysis for Algorithm 5

We first consider TPAUC maximization in the regular learning setting. Define $F(\mathbf{w}, \mathbf{s}, s') := \frac{1}{n_+} \sum_{X_i \in \mathcal{S}_+} f_i(\psi_i(\mathbf{w}, s_i), s')$. Due to the weak-convexity of $F(\mathbf{w}, \mathbf{s}, s')$ w.r.t. $(\mathbf{w}, \mathbf{s}, s')$, we consider the following Moreau envelope and proximal map defined as

$$F_\lambda(\mathbf{w}, \mathbf{s}, s') = \min_{\tilde{\mathbf{w}}, \tilde{\mathbf{s}}, \tilde{s}'} F(\tilde{\mathbf{w}}, \tilde{\mathbf{s}}, \tilde{s}') + \frac{1}{2\lambda} \left( \|\tilde{\mathbf{w}} - \mathbf{w}\|^2 + \|\tilde{\mathbf{s}} - \mathbf{s}\|^2 + \|\tilde{s}' - s'\|^2 \right),$$

$$\text{prox}_{\lambda F}(\mathbf{w}, \mathbf{s}, s') = \arg\min_{\tilde{\mathbf{w}}, \tilde{\mathbf{s}}, \tilde{s}'} F(\tilde{\mathbf{w}}, \tilde{\mathbf{s}}, \tilde{s}') + \frac{1}{2\lambda} \left( \|\tilde{\mathbf{w}} - \mathbf{w}\|^2 + \|\tilde{\mathbf{s}} - \mathbf{s}\|^2 + \|\tilde{s}' - s'\|^2 \right).$$

Following the same proof of Lemma 4.5, we have the following error bound

**Lemma C.3.** *Consider the update for $\{u_{i,t} : X_i \in \mathcal{S}_+\}$ in Algorithm 5. Assume $\psi_i(\mathbf{w}, s_i)$ is $C_\psi$-Lipshitz continuous for all $X_i \in \mathcal{S}_+$. Assume $\mathbb{E}_t[\|G_t\|^2] \leq M^2$ and $\mathbb{E}_t[\|\frac{1}{B_1} \sum_{X_i \in \mathcal{B}_1^t} \partial_s \psi_i(\mathbf{w}_t, s_{i,t}; \mathcal{B}_2^t) \partial_u f(u_{i,t}, s_t') e_i\|^2] \leq M^2$, where $e_i$ is the $n_+$-dimensional vector with 1 at the $i$-th entry and 0 everywhere else. With $\gamma = \frac{n_+ - B_1}{B_1(1-\tau)} + (1 - \tau)$ and $\tau \leq \frac{1}{2}$, we have*

$$\mathbb{E}\left[ \frac{1}{n_+} \sum_{X_i \in \mathcal{S}_+} \|u_{i,t+1} - \psi_i(\mathbf{w}_{t+1}, s_{i,t+1})\| \right]$$

$$\leq (1 - \frac{B_1 \tau}{2n_1})^{t+1} \frac{1}{n} \sum_{X_i \in \mathcal{S}_+} \|u_{i,0} - \psi_i(\mathbf{w}_0, s_{i,0})\| + \frac{2\tau^{1/2}\sigma}{B_2^{1/2}} + \frac{8n_+ C_\psi M \eta}{B_1 \tau^{1/2}}.$$

Then we have following convergence guarantee.

**Theorem C.4.** *Under the assumptions given in Lemma C.1, with $\gamma = \frac{n_+ - B_1}{B_1(1-\tau)} + (1-\tau)$, $\tau = \mathcal{O}(B_2\epsilon^4) \leq \frac{1}{2}$, $\eta = \mathcal{O}(\frac{B_1 B_2^{1/2}\epsilon^4}{n_+})$, and $\bar{\rho} = \rho_F + \rho_\psi C_f$, Algorithm 5 converges to an $\epsilon$-stationary point of the Moreau envelope $F_{1/\bar{\rho}}$ in $T = \mathcal{O}(\frac{n_+}{B_1 B_2^{1/2}}\epsilon^{-6})$ iterations.*

*Proof of Theorem C.4.* Define $(\hat{\mathbf{w}}_t, \hat{s}_t, \hat{s}'_t) := \mathrm{prox}_{F/\bar{\rho}}(\mathbf{w}_t, \mathbf{s}_t, s'_t)$. For a given $X_i \in \mathcal{S}_+$, we have

$f_i(\psi_i(\hat{\mathbf{w}}_t, \hat{s}_{i,t}), \hat{s}'_t) - f_i(u_{i,t}, s'_t)$

$\overset{(a)}{\geq} \partial_{s'} f_i(u_{i,t}, s'_t)(\hat{s}'_t - s'_t) + \partial_u f_i(u_{i,t}, s'_t)(\psi_i(\hat{\mathbf{w}}_t, \hat{s}_{i,t}) - u_{i,t})$

$\overset{(b)}{\geq} \partial_{s'} f_i(u_{i,t}, s'_t)(\hat{s}'_t - s'_t) + \partial_u f_i(u_{i,t}, s'_t)\Big[\psi_i(\mathbf{w}_t, s_{i,t}) - u_{i,t} + \langle \partial_w \psi_i(\mathbf{w}_t, s_{i,t}), \hat{\mathbf{w}}_t - \mathbf{w}_t\rangle$

$\qquad - \frac{\rho_\psi}{2}\|\hat{\mathbf{w}}_t - \mathbf{w}_t\|^2 + \langle \partial_{s_i}\psi_i(\mathbf{w}_t, s_{i,t}), \hat{s}_{i,t} - s_{i,t}\rangle - \frac{\rho_\psi}{2}\|\hat{s}_{i,t} - s_{i,t}\|^2\Big]$

$\overset{(c)}{\geq} \partial_{s'} f_i(u_{i,t}, s'_t)(\hat{s}'_t - s'_t) + \partial_u f_i(u_{i,t}, s'_t)\big[\psi_i(\mathbf{w}_t, s_{i,t}) - u_{i,t}\big] + \langle \partial_u f_i(u_{i,t}, s'_t)\partial_w \psi_i(\mathbf{w}_t, s_{i,t}), \hat{\mathbf{w}}_t - \mathbf{w}_t\rangle$

$\qquad + \langle \partial_u f_i(u_{i,t}, s'_t)\partial_{s_i}\psi_i(\mathbf{w}_t, s_{i,t}), \hat{s}_{i,t} - s_{i,t}\rangle - \frac{\rho_\psi C_f}{2}\left(\|\hat{\mathbf{w}}_t - \mathbf{w}_t\|^2 + \|\hat{s}_{i,t} - s_{i,t}\|^2\right)$

where (a) follows from the convexity of $f_i$, (b) follows from the monotonicity of $f_i(\cdot, s')$ and weak convexity of $\psi_i$, (c) is due to $0 \leq \partial_u f_i(u_{i,t}, s'_t) \leq C_f$. Then it follows

$$
\frac{1}{n_+} \sum_{X_i \in S_+} \Big[\partial_{s'} f_i(u_{i,t}, s'_t)(\hat{s}'_t - s'_t) + \langle \partial_u f_i(u_{i,t}, s'_t)\partial_w \psi_i(\mathbf{w}_t, s_{i,t}), \hat{\mathbf{w}}_t - \mathbf{w}_t\rangle
$$

$$
\qquad + \langle \partial_u f_i(u_{i,t}, s'_t)\partial_{s_i}\psi_i(\mathbf{w}_t, s_{i,t}), \hat{s}_{i,t} - s_{i,t}\rangle\Big]
$$

$$
\leq \frac{1}{n_+} \sum_{X_i \in S_+} \Big[f_i(\psi_i(\hat{\mathbf{w}}_t, \hat{s}_{i,t}), \hat{s}'_t) - f_i(u_{i,t}, s'_t) - \partial_u f_i(u_{i,t}, s'_t)\big[\psi_i(\mathbf{w}_t, s_{i,t}) - u_{i,t}\big] \tag{29}
$$

$$
\qquad + \frac{\rho_\psi C_f}{2}\left(\|\hat{\mathbf{w}}_t - \mathbf{w}_t\|^2 + \|\hat{s}_{i,t} - s_{i,t}\|^2\right)\Big]
$$

Now we consider the change in the Moreau envelope:

$$
\mathbb{E}_t[F_{1/\bar{\rho}}(\mathbf{w}_{t+1}, \mathbf{s}_{t+1}, s'_{t+1})]
$$

$$
= \mathbb{E}_t\left[\min_{\tilde{\mathbf{w}}, \tilde{\mathbf{s}}, \tilde{s}'} F(\tilde{\mathbf{w}}, \tilde{\mathbf{s}}, \tilde{s}') + \frac{\bar{\rho}}{2}\left(\|\tilde{\mathbf{w}} - \mathbf{w}_{t+1}\|^2 + \|\tilde{\mathbf{s}} - \mathbf{s}_{t+1}\|^2 + \|\tilde{s}' - s'_{t+1}\|^2\right)\right]
$$

$$
\leq \mathbb{E}_t\left[F(\hat{\mathbf{w}}_t, \hat{\mathbf{s}}_t, \hat{s}'_t) + \frac{\bar{\rho}}{2}\left(\|\hat{\mathbf{w}}_t - \mathbf{w}_{t+1}\|^2 + \|\hat{\mathbf{s}}_t - \mathbf{s}_{t+1}\|^2 + \|\hat{s}'_t - s'_{t+1}\|^2\right)\right]
$$

$$
= F(\hat{\mathbf{w}}_t, \hat{\mathbf{s}}_t, \hat{s}'_t) + \mathbb{E}_t\Big[\frac{\bar{\rho}}{2}\big(\|\hat{\mathbf{w}}_t - (\mathbf{w}_t - \eta G_t)\|^2 + \|\hat{\mathbf{s}}_t - (\mathbf{s}_t - \eta G^1_t)\|^2
$$

$$
\qquad + \|\hat{s}'_t - (s'_t - \eta G^2_t)\|^2\big)\Big] \tag{30}
$$

$$
\leq F(\hat{\mathbf{w}}_t, \hat{\mathbf{s}}_t, \hat{s}'_t) + \frac{\bar{\rho}}{2}\left(\|\hat{\mathbf{w}}_t - \mathbf{w}_t\|^2 + \|\hat{\mathbf{s}}_t - \mathbf{s}_t\|^2 + \|\hat{s}'_t - s'_t\|^2\right)
$$

$$
\qquad + \bar{\rho}\mathbb{E}_t[\eta\langle \hat{\mathbf{w}}_t - \mathbf{w}_t, G_t\rangle + \eta\langle \hat{\mathbf{s}}_t - \mathbf{s}_t, G^1_t\rangle + \eta\langle \hat{s}'_t - s'_t, G^2_t\rangle] + \frac{3\eta^2\bar{\rho}M^2}{2}
$$

$$
= F_{1/\bar{\rho}}(\mathbf{w}_t, \mathbf{s}_t, s'_t) + \bar{\rho}\mathbb{E}_t[\eta\langle \hat{\mathbf{w}}_t - \mathbf{w}_t, G_t\rangle + \eta\langle \hat{\mathbf{s}}_t - \mathbf{s}_t, G^1_t\rangle + \eta\langle \hat{s}'_t - s'_t, G^2_t\rangle]
$$

$$
\qquad + \frac{3\eta^2\bar{\rho}M^2}{2}
$$

where for simplicity we denote $G^1_t = \frac{1}{B_1}\sum_{X_i \in \mathcal{B}^t_1} \partial_u f_i(u_{i,t}, s'_t)\partial_s \psi_i(\mathbf{w}_t, s_{i,t}; \mathcal{B}^t_2)$ and $G^2_t = \frac{1}{B_1}\sum_{X_i \in \mathcal{B}^t_1} \partial_{s'} f_i(u_{i,t}, s'_t)$. The second inequality in the above derivation uses the bounds of $\mathbb{E}[\|G_t\|^2]$, $\mathbb{E}[\|G^1_t\|^2]$ and $\mathbb{E}[\|G^2_t\|^2]$, which follow from the Lipschitz continuity and bounded variance

assumptions and are denoted by $M$. Moreover, we have

$$\mathbb{E}_t[\eta\langle\hat{\mathbf{w}}_t - \mathbf{w}_t, G_t\rangle + \eta\langle\hat{\mathbf{s}}_t - \mathbf{s}_t, G_t^1\rangle + \eta\langle\hat{s}_t' - s_t', G_t^2\rangle]$$
$$= \eta\langle\hat{\mathbf{w}}_t - \mathbf{w}_t, \mathbb{E}_t[G_t]\rangle + \eta\langle\hat{\mathbf{s}}_t - \mathbf{s}_t, \mathbb{E}_t[G_t^1]\rangle + \eta\langle\hat{s}_t' - s_t', \mathbb{E}_t[G_t^2]\rangle,$$

and

$$\mathbb{E}_t[G_t] = \frac{1}{n_+}\sum_{X_i\in\mathcal{S}_+}\partial_u f_i(u_{i,t}, s_t')\partial_w\psi_i(\mathbf{w}_t, s_{i,t})$$

$$\mathbb{E}_t[G_t^1] = \frac{1}{n_+}\sum_{X_i\in\mathcal{S}_+}\partial_u f_i(u_{i,t}, s_t')\partial_{\mathbf{s}}\psi_i(\mathbf{w}_t, s_{i,t})$$

$$\mathbb{E}_t[G_t^2] = \frac{1}{n_+}\sum_{X_i\in\mathcal{S}_+}\partial_{s'} f_i(u_{i,t}, s_t').$$

Combining inequality 29 and 30 yields

$$\mathbb{E}_t[F_{1/\bar{\rho}}(\mathbf{w}_{t+1}, \mathbf{s}_{t+1}, s_{t+1}')]$$
$$\leq F_{1/\bar{\rho}}(\mathbf{w}_t, \mathbf{s}_t, s_t') + \frac{3\eta^2\bar{\rho}M^2}{2} + \frac{\bar{\rho}\eta}{n_+}\sum_{X_i\in S_+}\left[ f_i(\psi_i(\hat{\mathbf{w}}_t, \hat{s}_{i,t}), \hat{s}_t') - f_i(u_{i,t}, s_t')\right.$$

$$\left. - \partial_u f_i(u_{i,t}, s_t')\left[\psi_i(\mathbf{w}_t, s_{i,t}) - u_{i,t}\right] + \frac{\rho_\psi C_f}{2}\left(\|\hat{\mathbf{w}}_t - \mathbf{w}_t\|^2 + \|\hat{s}_{i,t} - s_{i,t}\|^2\right)\right]$$

$$\leq F_{1/\bar{\rho}}(\mathbf{w}_t, \mathbf{s}_t, s_t') + \frac{3\eta^2\bar{\rho}M^2}{2} + \bar{\rho}\eta(F(\hat{\mathbf{w}}_t, \hat{\mathbf{s}}_t, \hat{s}_t') - F(\mathbf{w}_t, \mathbf{s}_t, s_t')) \quad (31)$$

$$+ \frac{\bar{\rho}\eta}{n_+}\sum_{X_i\in S_+}\left[ f_i(\psi_i(\mathbf{w}_t, s_{i,t}), s_t') - f_i(u_{i,t}, s_t') - \partial_u f_i(u_{i,t}, s_t')\left[\psi_i(\mathbf{w}_t, s_{i,t}) - u_{i,t}\right]\right.$$

$$\left. + \frac{\rho_\psi C_f}{2}\left(\|\hat{\mathbf{w}}_t - \mathbf{w}_t\|^2 + \|\hat{s}_{i,t} - s_{i,t}\|^2\right)\right]$$

Due to the $\rho_F$-weak convexity of $F(\mathbf{w}, \mathbf{s}, s')$, we have $(\bar{\rho} - \rho_F)$-strong convexity of $(\mathbf{w}, \mathbf{s}, s')\mapsto F(\mathbf{w}, \mathbf{s}, s') + \frac{\bar{\rho}}{2}\|(\mathbf{w}_t, \mathbf{s}_t, s_t') - (\mathbf{w}, \mathbf{s}, s')\|^2$. Then it follows

$$F(\hat{\mathbf{w}}_t, \hat{\mathbf{s}}_t, \hat{s}_t') - F(\mathbf{w}_t, \mathbf{s}_t, s_t') = \left[F(\hat{\mathbf{w}}_t, \hat{\mathbf{s}}_t, \hat{s}_t') + \frac{\bar{\rho}}{2}\|(\mathbf{w}_t, \mathbf{s}_t, s_t') - (\hat{\mathbf{w}}_t, \hat{\mathbf{s}}_t, \hat{s}_t')\|^2\right]$$
$$- \left[F(\mathbf{w}_t, \mathbf{s}_t, s_t') + \frac{\bar{\rho}}{2}\|(\mathbf{w}_t, \mathbf{s}_t, s_t') - (\mathbf{w}_t, \mathbf{s}_t, s_t')\|^2\right] \quad (32)$$
$$- \frac{\bar{\rho}}{2}\|(\mathbf{w}_t, \mathbf{s}_t, s_t') - (\hat{\mathbf{w}}_t, \hat{\mathbf{s}}_t, \hat{s}_t')\|^2$$
$$\leq (\frac{\rho_F}{2} - \bar{\rho})\|(\mathbf{w}_t, \mathbf{s}_t, s_t') - (\hat{\mathbf{w}}_t, \hat{\mathbf{s}}_t, \hat{s}_t')\|^2$$

Plugging inequality 32 into inequality 31 yields

$$\mathbb{E}_t[F_{1/\bar{\rho}}(\mathbf{w}_{t+1}, \mathbf{s}_{t+1}, s_{t+1}')]$$

$$\leq \mathbb{E}[F_{1/\bar{\rho}}(\mathbf{w}_t, \mathbf{s}_t, s_t')] + \frac{3\eta^2\bar{\rho}M^2}{2} + \bar{\rho}\eta(\frac{\rho_F}{2} - \bar{\rho})\|(\mathbf{w}_t, \mathbf{s}_t, s_t') - (\hat{\mathbf{w}}_t, \hat{\mathbf{s}}_t, \hat{s}_t')\|^2$$

$$+ \frac{\bar{\rho}\eta}{n_+}\sum_{X_i\in S_+}\left[ f_i(\psi_i(\mathbf{w}_t, s_{i,t}), s_t') - f_i(u_{i,t}, s_t') - \partial_u f_i(u_{i,t}, s_t')\left[\psi_i(\mathbf{w}_t, s_{i,t}) - u_{i,t}\right]\right. \quad (33)$$

$$\left. + \frac{\rho_\psi C_f}{2}\left(\|\hat{\mathbf{w}}_t - \mathbf{w}_t\|^2 + \|\hat{s}_{i,t} - s_{i,t}\|^2\right)\right]$$

Set $\bar{\rho} = \rho_F + \rho_\psi C_f$. We have

$$\mathbb{E}_t[F_{1/\bar{\rho}}(\mathbf{w}_{t+1}, \mathbf{s}_{t+1}, s'_{t+1})]$$

$$\leq F_{1/\bar{\rho}}(\mathbf{w}_t, \mathbf{s}_t, s'_t) + \frac{3\eta^2\bar{\rho}M^2}{2} - \frac{\bar{\rho}^2\eta}{2}\|(\mathbf{w}_t, \mathbf{s}_t, s'_t) - (\hat{\mathbf{w}}_t, \hat{\mathbf{s}}_t, \hat{s}'_t)\|^2$$

$$+ \frac{\bar{\rho}\eta}{n_+} \sum_{X_i \in S_+} \left[ f_i(\psi_i(\mathbf{w}_t, s_{i,t}), s'_t) - f_i(u_{i,t}, s'_t) - \partial_u f_i(u_{i,t}, s'_t)\big[\psi_i(\mathbf{w}_t, s_{i,t}) - u_{i,t}\big] \right]$$

$$\overset{(a)}{\leq} F_{1/\bar{\rho}}(\mathbf{w}_t, \mathbf{s}_t, s'_t) + \frac{3\eta^2\bar{\rho}M^2}{2} - \frac{\eta}{2}\|\nabla\varphi_{1/\bar{\rho}}(\mathbf{w}_t, \mathbf{s}_t, s'_t)\|^2$$

$$+ \frac{\bar{\rho}\eta}{n_+} \sum_{X_i \in S_+} \left[ f_i(\psi_i(\mathbf{w}_t, s_{i,t}), s'_t) - f_i(u_{i,t}, s'_t) - \partial_u f_i(u_{i,t}, s'_t)\big[\psi_i(\mathbf{w}_t, s_{i,t}) - u_{i,t}\big] \right]$$

where inequality (a) follows from Lemma 3.2.

Using the Lipschitz continuity of $f$, we have

$$\mathbb{E}_t[F_{1/\bar{\rho}}(\mathbf{w}_{t+1}, \mathbf{s}_{t+1}, s'_{t+1})]$$

$$\leq F_{1/\bar{\rho}}(\mathbf{w}_t, \mathbf{s}_t, s'_t) + \frac{3\eta^2\bar{\rho}M^2}{2} - \frac{\eta}{2}\|\nabla F_{1/\bar{\rho}}(\mathbf{w}_t, \mathbf{s}_t, s'_t)\|^2 \tag{34}$$

$$+ \frac{\bar{\rho}\eta}{n_+} \sum_{X_i \in S_+} 2C_f\|\psi_i(\mathbf{w}_t, s_{i,t}) - u_{i,t}\|$$

With the error bound from Lemma C.3, we have

$$\mathbb{E}\left[ \frac{1}{n_+} \sum_{X_i \in S_+} \|\psi_i(\mathbf{w}_t, s_{i,t}) - u_{i,t}\| \right] \leq (1-\mu)^t \frac{1}{n_+} \sum_{X_i \in S_+} \|\psi_i(\mathbf{w}_0, s_{i,0}) - u_{i,0}\| + R$$

with $\mu = \frac{B_1\tau}{2n_+}$, $R = \frac{2\tau^{1/2}\sigma}{B_2^{1/2}} + \frac{4n_+ C_\psi M\eta}{B_1\tau^{1/2}} + \frac{4n_+^{1/2}C_\psi M\eta}{B_1\tau^{1/2}}$. Then

$$\mathbb{E}[F_{1/\bar{\rho}}(\mathbf{w}_{t+1}, \mathbf{s}_{t+1}, s'_{t+1})]$$

$$\leq F_{1/\bar{\rho}}(\mathbf{w}_t, \mathbf{s}_t, s'_t) + \frac{3\eta^2\bar{\rho}M^2}{2} - \frac{\eta}{2}\mathbb{E}[\|\nabla F_{1/\bar{\rho}}(\mathbf{w}_t, \mathbf{s}_t, s'_t)\|^2] \tag{35}$$

$$+ 2C_f\bar{\rho}\eta \left( (1-\mu)^t \frac{1}{n_+} \sum_{X_i \in S_+} \|\psi_i(\mathbf{w}_0, s_{i,0}) - u_{i,0}\| + R \right)$$

Taking summation from $t = 0$ to $T - 1$ yields

$$\mathbb{E}[F_{1/\bar{\rho}}(\mathbf{w}_T, \mathbf{s}_T, s'_T)]$$

$$\leq F_{1/\bar{\rho}}(\mathbf{w}_0, \mathbf{s}_0, s'_0) + \frac{3\eta^2\bar{\rho}M^2 T}{2} - \frac{\eta}{2}\sum_{t=0}^{T-1}\mathbb{E}[\|\nabla F_{1/\bar{\rho}}(\mathbf{w}_t, \mathbf{s}_t, s'_t)\|^2]$$

$$+ 2C_f\bar{\rho}\eta \left( \sum_{t=0}^{T-1}(1-\mu)^t \frac{1}{n_+} \sum_{X_i \in S_+} \|\psi_i(\mathbf{w}_0, s_{i,0}) - u_{i,0}\| + RT \right) \tag{36}$$

$$\overset{(a)}{\leq} F_{1/\bar{\rho}}(\mathbf{w}_0, \mathbf{s}_0, s'_0) + \frac{3\eta^2\bar{\rho}M^2 T}{2} - \frac{\eta}{2}\sum_{t=0}^{T-1}\mathbb{E}[\|\nabla F_{1/\bar{\rho}}(\mathbf{w}_t, \mathbf{s}_t, s'_t)\|^2]$$

$$+ \frac{4C_f\bar{\rho}\eta}{\mu} \sum_{X_i \in S_+} \frac{1}{n_+}\|\psi_i(\mathbf{w}_0, s_{i,0}) - u_{i,0}\| + 2C_f\bar{\rho}\eta RT$$

where (a) uses $\sum_{t=0}^{T-1}(1-\mu)^t \leq \frac{1}{\mu}$.

Lower bounding the left-hand-side by $\min_{\mathbf{w},\mathbf{s},s'} F_{1/\bar{\rho}}(\mathbf{w},\mathbf{s},s')$, we obtain

$$
\frac{1}{T}\sum_{t=0}^{T-1}\mathbb{E}[\|\nabla F_{1/\bar{\rho}}(\mathbf{w}_t,\mathbf{s}_t,s_t')\|^2]
$$

$$
\leq \frac{2}{\eta T}\left[ F_{1/\bar{\rho}}(\mathbf{w}_0,\mathbf{s}_0,s_0') - \min_{\mathbf{w},\mathbf{s},s'} F_{1/\bar{\rho}}(\mathbf{w},\mathbf{s},s') + \frac{3\eta^2\bar{\rho}M^2 T}{2} \right.
$$

$$
\left. + \frac{4C_f\bar{\rho}\eta}{n_+}\sum_{X_i\in S_+}\|\psi_i(\mathbf{w}_0,s_{i,0}) - u_{i,0}\| + 2C_f\bar{\rho}\eta RT \right]
$$

$$
\leq \frac{2\Delta}{\eta T} + 3\eta\bar{\rho}M^2 + \frac{8C_f\bar{\rho}}{\mu T n_+}\sum_{X_i\in S_+}\|\psi_i(\mathbf{w}_0,s_{i,0}) - u_{i,0}\| + 4C_f\bar{\rho}R
$$

$$
\leq \frac{C}{T}(\frac{1}{\eta} + \frac{1}{\mu}) + C(\eta + R)
$$

where we assume $F_{1/\bar{\rho}}(\mathbf{w}_0,\mathbf{s}_0,s_0') - \min_{\mathbf{w},\mathbf{s},s'} F_{1/\bar{\rho}}(\mathbf{w},\mathbf{s},s') \leq \Delta$ and

$$
C = \max\{8\Delta, 12\bar{\rho}M^2, 32C_f\bar{\rho}\sum_{X_i\in S_+}\|\psi_i(\mathbf{w}_0,s_{i,0}) - u_{i,0}\|, 16C_f\bar{\rho}\}.
$$

Plugging the expression of $\mu$ and $R$ yields

$$
\frac{1}{T}\sum_{t=0}^{T-1}\mathbb{E}[\|\nabla F_{1/\bar{\rho}}(\mathbf{w}_t,\mathbf{s}_t,s_t')\|^2]
$$

$$
\leq \mathcal{O}\left( \frac{1}{T}(\frac{1}{\eta} + \frac{n_+}{B_1\tau}) + (\frac{\tau^{1/2}\sigma}{B_2^{1/2}} + \frac{n_+\eta}{B_1\tau^{1/2}}) \right)
$$

Setting $\tau = \mathcal{O}(B_2\epsilon^4)$ and $\eta = \mathcal{O}(\frac{B_1 B_2^{1/2}}{n_+}\epsilon^4)$, with $T = \mathcal{O}(\frac{n_+}{B_1 B_2^{1/2}}\epsilon^{-6})$ iterations, we have

$$
\frac{1}{T}\sum_{t=0}^{T-1}\mathbb{E}[\|\nabla F_{1/\bar{\rho}}(\mathbf{w}_t,\mathbf{s}_t,s_t')\|^2] \leq \epsilon^2.
$$

$\square$

### C.4.2 Convergence analysis for Algorithm 6

We now consider MIL TPAUC maximization with mean pooling. Define $F(\mathbf{w},\mathbf{s},s') := \frac{1}{n_+}\sum_{X_i\in S_+} f_i(g_i(h_j(\mathbf{w}) - h_i(\mathbf{w}), s_i), s')$. Due to the weak-convexity of $F(\mathbf{w},\mathbf{s},s')$ w.r.t. $(\mathbf{w},\mathbf{s},s')$, we consider the following Moreau envelope and proximal map defined as

$$
F_\lambda(\mathbf{w},\mathbf{s},s') = \min_{\tilde{\mathbf{w}},\tilde{\mathbf{s}},\tilde{s}'} F(\tilde{\mathbf{w}},\tilde{\mathbf{s}},\tilde{s}') + \frac{1}{2\lambda}\left( \|\tilde{\mathbf{w}} - \mathbf{w}\|^2 + \|\tilde{\mathbf{s}} - \mathbf{s}\|^2 + \|\tilde{s}' - s'\|^2 \right),
$$

$$
\mathrm{prox}_{\lambda F}(\mathbf{w},\mathbf{s},s') = \arg\min_{\tilde{\mathbf{w}},\tilde{\mathbf{s}},\tilde{s}'} F(\tilde{\mathbf{w}},\tilde{\mathbf{s}},\tilde{s}') + \frac{1}{2\lambda}\left( \|\tilde{\mathbf{w}} - \mathbf{w}\|^2 + \|\tilde{\mathbf{s}} - \mathbf{s}\|^2 + \|\tilde{s}' - s'\|^2 \right).
$$

Following the same proofs of Lemma A.3 and Lemma A.4, we have the following error bounds

**Lemma C.5.** *Consider the update for $\{v_{i,t} : X_i \in \mathcal{S}_+ \cup \mathcal{S}_-\}$ in Algorithm 6. Assume $h_i(\mathbf{w};\xi)$ is $C_h$-Lipshitz for all $X_i \in S_+ \cup S_-$, and $\mathbb{E}[\|G_t\|^2] \leq M^2$. With $\gamma_1 = \frac{n_+ - B_1}{B_1(1-\tau_1)} + (1 - \tau_1),$*

$\gamma_2 = \frac{n_- - B_2}{B_2(1-\tau_1)} + (1-\tau_1)$ and $\tau_1 \le \frac{1}{2}$, we have

$$\mathbb{E}\left[\frac{1}{n_+}\sum_{X_i \in \mathcal{S}_+}\|v_{i,t+1} - h_i(\mathbf{w}_{t+1})\|\right] \le (1 - \frac{B_1\tau_1}{2n_+})^{t+1}\sum_{X_i \in \mathcal{S}_+}\|v_{i,0} - h_i(\mathbf{w}_t)\| + 2\tau_1^{1/2}\sigma + \frac{4n_+ C_h M\eta}{B_1\tau_1^{1/2}}$$

$$\mathbb{E}\left[\frac{1}{n_-}\sum_{X_j \in \mathcal{S}_-}\|v_{j,t+1} - h_j(\mathbf{w}_{t+1})\|\right] \le (1 - \frac{B_1\tau_1}{2n_-})^{t+1}\frac{1}{n_-}\sum_{X_j \in \mathcal{S}_-}\|v_{j,0} - h_j(\mathbf{w}_t)\| + 2\tau_1^{1/2}\sigma + \frac{4n_- C_h M\eta}{B_1\tau_1^{1/2}}$$

$$\mathbb{E}\left[\frac{1}{n_+}\sum_{X_i \in \mathcal{S}_+}\|v_{i,t+1} - h_i(\mathbf{w}_{t+1})\|^2\right] \le (1 - \frac{B_1\tau_1}{2n_+})^{2(t+1)}\frac{1}{n_+}\sum_{X_i \in \mathcal{S}_+}\|v_{i,0} - h_i(\mathbf{w}_t)\|^2 + 4\tau_1\sigma^2 + \frac{16n_+^2 C_h^2 M^2\eta^2}{B_1^2\tau_1}$$

$$\mathbb{E}\left[\frac{1}{n_-}\sum_{X_j \in \mathcal{S}_-}\|v_{j,t+1} - h_j(\mathbf{w}_{t+1})\|^2\right] \le (1 - \frac{B_1\tau_1}{2n_-})^{2(t+1)}\frac{1}{n_-}\sum_{X_j \in \mathcal{S}_-}\|v_{j,0} - h_j(\mathbf{w}_t)\|^2 + 4\tau_1\sigma^2 + \frac{16n_-^2 C_h^2 M^2\eta^2}{B_1^2\tau_1}$$

**Lemma C.6.** *Consider update for $\{u_{i,t} : X_i \in \mathcal{S}_+\}$ in Algorithm 6. Assume $g_i(v_{ij}, s_i)$ is $C_g$-Lipshitz w.r.t. $(v_{ij}, s_i)$ for all $X_i \in S_+$ and $X_j \in S_-$. With $\gamma_3 = \frac{n_+ - B_1}{B_1(1-\tau_2)} + (1-\tau_2)$ and $\tau_2 \le \frac{1}{2}$, we have*

$$\mathbb{E}\left[\frac{1}{n_+}\sum_{X_i \in S_+}\|u_{i,t+1} - \frac{1}{n_-}\sum_{X_j \in S_-}g_i(v_{j,t+1} - v_{i,t+1}, s_{i,t+1})\|\right]$$

$$\le (1 - \frac{B_1\tau_2}{2n_+})^{t+1}\frac{1}{n_+}\sum_{X_i \in S_+}\|u_{i,0} - \frac{1}{n_-}\sum_{X_j \in S_-}g_i(v_{j,0} - v_{i,0}, s_{i,0})\| + 2\tau_2^{1/2}\sigma$$

$$+ C_2\frac{n_+}{B_1}(\frac{B_1^{1/2}}{n_+^{1/2}} + \frac{B_2^{1/2}}{n_-^{1/2}})\frac{\tau_1}{\tau_2^{1/2}} + C_2\frac{n_+}{B_1}(\frac{n_+^{1/2}}{B_1^{1/2}} + \frac{n_-^{1/2}}{B_2^{1/2}})\frac{\eta}{\tau_2^{1/2}} + C_2\frac{n_+^{1/2}\eta}{B_1\tau_2^{1/2}}$$

*where $C_2$ is a constant defined in the proof.*

Then we have the following covnergence guarantee.

**Theorem C.7.** *Under assumptions given in Lemma C.2, with $\gamma_1 = \frac{n_1 - B_1}{B_1(1-\tau_1)} + (1-\tau_1)$, $\gamma_2 = \frac{n_2 - B_2}{B_2(1-\tau_1)} + (1-\tau_1)$, $\gamma_3 = \frac{n_1 - B_1}{B_1(1-\tau_2)} + (1-\tau_2)$, $\tau_1 = \mathcal{O}\left(\min\left\{B_3, \frac{B_1}{n_+}\min\{\frac{n_+^{1/2}}{B_1^{1/2}}, \frac{n_-^{1/2}}{B_2^{1/2}}\}B_2^{1/2}\right\}\epsilon^4\right) \le 1/2$, $\tau_2 = \mathcal{O}(B_2\epsilon^4) \le 1/2$, $\eta = \mathcal{O}\left(\min\left\{\min\{\frac{B_1}{n_+}, \frac{B_2}{n_-}\}\min\{B_3^{1/2}, \frac{B_1^{1/2}}{n_+^{1/2}}\min\{\frac{n_+^{1/4}}{B_1^{1/4}}, \frac{n_-^{1/4}}{B_2^{1/4}}\}B_2^{1/4}\}, \frac{B_1}{n_+}\min\{\frac{B_1^{1/2}}{n_+^{1/2}}, \frac{B_2^{1/2}}{n_-^{1/2}}\}B_3^{1/2}\right\}\epsilon^4\right)$, then after*

$$T \ge \mathcal{O}\left(\max\left\{\max\{\frac{n_+}{B_1}, \frac{n_-}{B_2}\}\max\{\frac{1}{B_3^{1/2}}, \frac{n_+^{1/2}}{B_1^{1/2}}\max\{\frac{B_1^{1/4}}{n_+^{1/4}}, \frac{B_2^{1/4}}{n_-^{1/4}}\}\frac{1}{B_2^{1/4}}\}, \frac{n_+}{B_1}\max\{\frac{n_+^{1/2}}{B_1^{1/2}}, \frac{n_-^{1/2}}{B_2^{1/2}}\}\frac{1}{B_2^{1/2}}\right\}\epsilon^{-6}\right)$$

*iterations, Algorithm 6 gives $\epsilon$-stationary point to the Moreau envelope, i.e.,*

$$\frac{1}{T}\sum_{t=0}^{T-1}\|\nabla F_{1/\bar\rho}(\mathbf{w}_t, \mathbf{s}_t, s_t')\|^2 \le \epsilon^2.$$

*where $\bar\rho = \rho_F + \rho_g C_f + 8\rho_g C_f C_h + C_f C_g L_h$.*

*Proof of Theorem C.7.* Consider the change in the Moreau envelope:

$$\mathbb{E}_t[F_{1/\bar{\rho}}(\mathbf{w}_{t+1}, \mathbf{s}_{t+1}, s'_{t+1})]$$

$$= \mathbb{E}_t\left[\min_{\tilde{\mathbf{w}}, \tilde{\mathbf{s}}, \tilde{s}'} F(\tilde{\mathbf{w}}, \tilde{\mathbf{s}}_t, \tilde{s}'_t) + \frac{\bar{\rho}}{2}\left(\|\tilde{\mathbf{w}} - \mathbf{w}_{t+1}\|^2 + \|\tilde{\mathbf{s}} - \mathbf{s}_{t+1}\|^2 + \|\tilde{s}' - s'_{t+1}\|^2\right)\right]$$

$$\leq \mathbb{E}_t\left[F(\hat{\mathbf{w}}_t, \hat{\mathbf{s}}_t, \hat{s}'_t) + \frac{\bar{\rho}}{2}\left(\|\hat{\mathbf{w}}_t - \mathbf{w}_{t+1}\|^2 + \|\hat{\mathbf{s}}_t - \mathbf{s}_{t+1}\|^2 + \|\hat{s}'_t - s'_{t+1}\|^2\right)\right]$$

$$= F(\hat{\mathbf{w}}_t, \hat{\mathbf{s}}_t, \hat{s}'_t) + \mathbb{E}_t\left[\frac{\bar{\rho}}{2}\left(\|\hat{\mathbf{w}}_t - (\mathbf{w}_t - \eta G_t)\|^2 + \|\hat{\mathbf{s}}_t - (\mathbf{s}_t - \eta G_t^1)\|^2\right.\right.$$

$$\left.\left. + \|\hat{s}'_t - (s'_t - \eta G_t^2)\|^2\right)\right] \tag{37}$$

$$\leq F(\hat{\mathbf{w}}_t, \hat{\mathbf{s}}_t, \hat{s}'_t) + \frac{\bar{\rho}}{2}\left(\|\hat{\mathbf{w}}_t - \mathbf{w}_t\|^2 + \|\hat{\mathbf{s}}_t - \mathbf{s}_t\|^2 + \|\hat{s}'_t - s'_t\|^2\right)$$

$$+ \bar{\rho}\mathbb{E}_t[\eta\langle\hat{\mathbf{w}}_t - \mathbf{w}_t, G_t\rangle + \eta\langle\hat{\mathbf{s}}_t - \mathbf{s}_t, G_t^1\rangle + \eta\langle\hat{s}'_t - s'_t, G_t^2\rangle] + \frac{3\eta^2\bar{\rho}M^2}{2}$$

$$= F_{1/\bar{\rho}}(\mathbf{w}_t, \mathbf{s}_t, s'_t) + \bar{\rho}\mathbb{E}_t[\eta\langle\hat{\mathbf{w}}_t - \mathbf{w}_t, G_t\rangle + \eta\langle\hat{\mathbf{s}}_t - \mathbf{s}_t, G_t^1\rangle + \eta\langle\hat{s}'_t - s'_t, G_t^2\rangle]$$

$$+ \frac{3\eta^2\bar{\rho}M^2}{2}$$

where for simplicity we denote $G_t^2 = \frac{1}{B_1}\sum_{i\in\mathcal{B}_1^t}\partial_{s'}f_i(u_{i,t}, s'_t)$, and $G_t^1$ is a $n_+$-dimensional vector whose $i$-th coordinate is defined as

$$\begin{cases}\frac{1}{B_1}\partial_u f_i(u_{i,t}, s'_t)\left[\frac{1}{B_2}\sum_{X_j\in\mathcal{B}_2^t}\partial_{s_i}g_i(v_{j,t} - v_{i,t}, s_{i,t})\right], & X_i \in \mathcal{B}_1^t \\ 0, & X_i \notin \mathcal{B}_1^t\end{cases}.$$

The second inequality in the above derivation uses the bounds of $\mathbb{E}[\|G_t\|^2], \mathbb{E}[\|G_t^1\|^2]$ and $\mathbb{E}[\|G_t^2\|^2]$, which follow from the Lipschitz continuity and bounded variance assumptions and are denoted by $M$.

Note that

$$\mathbb{E}_t[G_t]$$

$$= \frac{1}{n_+}\sum_{X_i\in S_+}\partial_u f_i(u_{i,t}, s'_t)\left[\frac{1}{n_-}\sum_{X_j\in S_-}\partial_v g_i(v_{j,t} - v_{i,t}, s_{i,t})\left(\nabla h_i(\mathbf{w}) - \nabla h_j(\mathbf{w})\right)\right]$$

$$\mathbb{E}_t[G_t^1] = \frac{1}{n_+}\sum_{X_i\in\mathcal{S}_+}\partial_u f_i(u_{i,t}, s'_t)\left[\frac{1}{n_-}\sum_{X_j\in S_-}\partial_{\mathbf{s}}g_i(v_{j,t} - v_{i,t}, s_{i,t})\right]$$

$$\mathbb{E}_t[G_t^2] = \frac{1}{n_+}\sum_{X_i\in S_+}\partial_{s'}f_i(u_{i,t}, s'_t)$$

Define $(\hat{\mathbf{w}}_t, \hat{\mathbf{s}}_t, \hat{s}'_t) := \text{prox}_{F/\bar{\rho}}(\mathbf{w}_t, \mathbf{s}_t, s'_t)$. For a given $i \in \{1, \ldots, m\}$, we have

$$f_i\Big(\frac{1}{n_-}\sum_{X_j \in S_-} g_i(h_j(\hat{\mathbf{w}}_t) - h_i(\hat{\mathbf{w}}_t), \hat{s}_{i,t}), \hat{s}'_t\Big) - f_i(u_{i,t}, s'_t)$$

$$\overset{(a)}{\geq} \partial_{s'} f_i(u_{i,t}, s'_t)(\hat{s}'_t - s'_t) + \partial_u f_i(u_{i,t}, s'_t)\Big(\frac{1}{n_-}\sum_{X_j \in S_-} g_i(h_j(\hat{\mathbf{w}}_t) - h_i(\hat{\mathbf{w}}_t), \hat{s}_{i,t}) - u_{i,t}\Big)$$

$$\overset{(b)}{\geq} \partial_{s'} f_i(u_{i,t}, s'_t)(\hat{s}'_t - s'_t) + \partial_u f_i(u_{i,t}, s'_t)\Big[\frac{1}{n_-}\sum_{X_j \in S_-} g_i(v_{j,t} - v_{i,t}, s_{i,t}) - u_{i,t}$$

$$+ \frac{1}{n_-}\sum_{X_j \in S_-} \langle \partial_v g_i(v_{j,t} - v_{i,t}, s_{i,t}), (h_j(\hat{\mathbf{w}}_t) - h_i(\hat{\mathbf{w}}_t)) - (v_{j,t} - v_{i,t}) \rangle$$

$$- \frac{1}{n_-}\sum_{X_j \in S_-} \frac{\rho_g}{2}\|(h_j(\hat{\mathbf{w}}_t) - h_i(\hat{\mathbf{w}}_t)) - (v_{j,t} - v_{i,t})\|^2$$

$$+ \langle \frac{1}{n_-}\sum_{X_j \in S_-} \partial_{s_i} g_i(v_{j,t} - v_{i,t}), s_{i,t}, \hat{s}_{i,t} - s_{i,t} \rangle - \frac{\rho_g}{2}\|\hat{s}_{i,t} - s_{i,t}\|^2\Big] \tag{38}$$

$$\overset{(c)}{\geq} \partial_{s'} f_i(u_{i,t}, s'_t)(\hat{s}'_t - s'_t) + \partial_u f_i(u_{i,t}, s'_t)\Big[\frac{1}{n_-}\sum_{X_j \in S_-} g_i(v_{j,t} - v_{i,t}, s_{i,t}) - u_{i,t}\Big]$$

$$+ \frac{1}{n_-}\sum_{X_j \in S_-} \underbrace{\langle \partial_u f_i(u_{i,t}, s'_t)\partial_v g_i(v_{j,t} - v_{i,t}, s_{i,t}), (h_j(\hat{\mathbf{w}}_t) - h_i(\hat{\mathbf{w}}_t)) - (v_{j,t} - v_{i,t}) \rangle}_{A_1}$$

$$+ \frac{1}{n_-}\sum_{X_j \in S_-} \langle \partial_u f_i(u_{i,t}, s'_t)\partial_{s_i} g_i(v_{j,t} - v_{i,t}, s_{i,t}), \hat{s}_{i,t} - s_{i,t} \rangle$$

$$- \frac{1}{n_-}\sum_{X_j \in S_-} \frac{\rho_g C_f}{2}\|(h_j(\hat{\mathbf{w}}_t) - h_i(\hat{\mathbf{w}}_t)) - (v_{j,t} - v_{i,t})\|^2 - \frac{\rho_g C_f}{2}\|\hat{s}_{i,t} - s_{i,t}\|^2$$

where (a) follows from the convexity of $f_i$, (b) follows from the monotonicity of $f_i(\cdot, s')$ and weak convexity of $g_i$, (c) is due to $0 \leq \partial_u f_i(u_{i,t}, s'_t) \leq C_f$.

The $L_h$-smoothness assumption of $h_i(\mathbf{w}) - h_j(\mathbf{w})$ for all $i, \mathbf{w}$ implies

$$h_i(\hat{\mathbf{w}}_t) - h_j(\hat{\mathbf{w}}_t)$$

$$\geq h_i(\mathbf{w}_t) - h_j(\mathbf{w}_t) + \langle (\nabla h_i(\mathbf{w}_t) - \nabla h_j(\mathbf{w}_t)), \hat{\mathbf{w}}_t - \mathbf{w}_t \rangle - \frac{L_h}{2}\|\hat{\mathbf{w}}_t - \mathbf{w}_t\|^2 \tag{39}$$

Since $\partial_u f_i(u_{i,t}, s'_t)\partial_v g_i(v_{j,t} - v_{i,t}, s_{i,t}) \geq 0$, we bound $A_1$ as following

$$A_1 = \langle \partial_u f_i(u_{i,t}, s'_t)\partial_v g_i(v_{j,t} - v_{i,t}, s_{i,t}), (h_j(\hat{\mathbf{w}}_t) - h_i(\hat{\mathbf{w}}_t)) - (v_{j,t} - v_{i,t}) \rangle$$

$$\overset{(a)}{\geq} \langle \partial_u f_i(u_{i,t}, s'_t)\partial_v g_i(v_{j,t} - v_{i,t}, s_{i,t}), (h_i(\mathbf{w}_t) - h_j(\mathbf{w}_t)) - (v_{j,t} - v_{i,t}) \rangle$$

$$- \langle \partial_u f_i(u_{i,t}, s'_t)\partial_v g_i(v_{j,t} - v_{i,t}, s_{i,t}), \frac{L_h}{2}\|\hat{\mathbf{w}}_t - \mathbf{w}_t\|^2 \rangle$$

$$+ \langle \partial_u f_i(u_{i,t}, s'_t)\partial_v g_i(v_{j,t} - v_{i,t}, s_{i,t})(\nabla h_i(\mathbf{w}_t) - \nabla h_j(\mathbf{w}_t)), \hat{\mathbf{w}}_t - \mathbf{w}_t \rangle$$

$$\overset{(b)}{\geq} -C_f C_g[\|h_i(\mathbf{w}_t) - v_{i,t}\| + \|h_j(\mathbf{w}_t) - v_{j,t}\|] - \frac{C_f C_g L_h}{2}\|\hat{\mathbf{w}}_t - \mathbf{w}_t\|^2$$

$$+ \langle \partial_u f_i(u_{i,t}, s'_t)\partial_\ell g_i(v_{j,t} - v_{i,t}, s_{i,t})(\nabla h_i(\mathbf{w}_t) - \nabla h_j(\mathbf{w}_t)), \hat{\mathbf{w}}_t - \mathbf{w}_t \rangle$$

where inequality (a) follows from inequality 39, (b) follows from the Lipschitz continuity and monotone assumptions on $f_i, g_i, h_i, h_j$. Then plugging the new formulation of $A_1$ back to inequality 38

yields

$$f_i(\frac{1}{n_-} \sum_{X_j \in S_-} g_i(h_j(\hat{\mathbf{w}}_t) - h_i(\hat{\mathbf{w}}_t), \hat{s}_{i,t}), \hat{s}_t') - f_i(u_{i,t}, s_t')$$

$$\geq \partial_{s'} f_i(u_{i,t}, s_t')(\hat{s}_t' - s_t') + \partial_u f_i(u_{i,t}, s_t') \left[ \frac{1}{n_-} \sum_{X_j \in S_-} g_i(v_{j,t} - v_{i,t}, s_{i,t}) - u_{i,t} \right]$$

$$+ \frac{1}{n_-} \sum_{X_j \in S_-} [-C_f C_g [\|h_i(\mathbf{w}_t) - v_{i,t}\| + \|h_j(\mathbf{w}_t) - v_{j,t}\|]] - \frac{C_f C_g L_h}{2} \|\hat{\mathbf{w}}_t - \mathbf{w}_t\|^2$$

$$+ \frac{1}{n_-} \sum_{X_j \in S_-} \langle \partial_u f_i(u_{i,t}, s_t') \partial_v g_i(v_{j,t} - v_{i,t}, s_{i,t})(\nabla h_i(\mathbf{w}_t) - \nabla h_j(\mathbf{w}_t)), \hat{\mathbf{w}}_t - \mathbf{w}_t \rangle$$

$$+ \frac{1}{n_-} \sum_{X_j \in S_-} \langle \partial_u f_i(u_{i,t}, s_t') \partial_{s_i} g_i(v_{j,t} - v_{i,t}, s_{i,t}), \hat{s}_{i,t} - s_{i,t} \rangle$$

$$- \frac{1}{n_-} \sum_{X_j \in S_-} \frac{\rho_g C_f}{2} \|(h_j(\hat{\mathbf{w}}_t) - h_i(\hat{\mathbf{w}}_t)) - (v_{j,t} - v_{i,t})\|^2 - \frac{\rho_g C_f}{2} \|\hat{s}_{i,t} - s_{i,t}\|^2$$

Taking average over $i \in S_+$ gives

$$\frac{1}{n_+} \sum_{X_i \in S_+} f_i(\frac{1}{n_-} \sum_{X_j \in S_-} g_i(h_j(\hat{\mathbf{w}}_t) - h_i(\hat{\mathbf{w}}_t), \hat{s}_{i,t}), \hat{s}_t') - f_i(u_{i,t}, s_t')$$

$$\geq \langle \mathbb{E}_t[G_t^2], \hat{s}_t' - s_t' \rangle + \langle \mathbb{E}_t[G_t], \hat{\mathbf{w}}_t - \mathbf{w}_t \rangle + \langle \mathbb{E}_t[G_t^1], \hat{\mathbf{s}}_t - \mathbf{s}_t \rangle$$

$$+ \frac{1}{n_+} \sum_{X_i \in S_+} \partial_u f_i(u_{i,t}, s_t') \left[ \frac{1}{n_-} \sum_{X_j \in S_-} g_i(v_{j,t} - v_{i,t}, s_{i,t}) - u_{i,t} \right]$$

$$- C_f C_g \left[ \frac{1}{n_+} \sum_{X_i \in S_+} \|h_i(\mathbf{w}_t) - v_{i,t}\| + \frac{1}{n_-} \sum_{X_j \in S_-} \|h_j(\mathbf{w}_t) - v_{j,t}\| \right] - \frac{C_f C_g L_h}{2} \|\hat{\mathbf{w}}_t - \mathbf{w}_t\|^2$$

$$- \frac{1}{n_+} \sum_{X_i \in S_+} \frac{1}{n_-} \sum_{X_j \in S_-} \frac{\rho_g C_f}{2} \|(h_j(\hat{\mathbf{w}}_t) - h_i(\hat{\mathbf{w}}_t)) - (v_{j,t} - v_{i,t})\|^2 - \frac{1}{n_+} \sum_{X_i \in S_+} \frac{\rho_g C_f}{2} \|\hat{s}_{i,t} - s_{i,t}\|^2$$

It follows

$$\langle \mathbb{E}_t[G_t^2], \hat{s}_t' - s_t' \rangle + \langle \mathbb{E}_t[G_t], \hat{\mathbf{w}}_t - \mathbf{w}_t \rangle + \langle \mathbb{E}_t[G_t^1], \hat{\mathbf{s}}_t - \mathbf{s}_t \rangle$$

$$\leq \frac{1}{n_+} \sum_{X_i \in S_+} \left[ f_i(\frac{1}{n_-} \sum_{X_j \in S_-} g_i(h_j(\hat{\mathbf{w}}_t) - h_i(\hat{\mathbf{w}}_t), \hat{s}_{i,t}), \hat{s}_t') - f_i(u_{i,t}, s_t') \right.$$

$$- \partial_u f_i(u_{i,t}, s_t') \left[ \frac{1}{n_-} \sum_{X_j \in S_-} g_i(v_{j,t} - v_{i,t}, s_{i,t}) - u_{i,t} \right]$$

$$+ \frac{1}{n_-} \sum_{X_j \in S_-} \frac{\rho_g C_f}{2} \|(h_j(\hat{\mathbf{w}}_t) - h_i(\hat{\mathbf{w}}_t)) - (v_{j,t} - v_{i,t})\|^2 + \frac{\rho_g C_f}{2} \|\hat{s}_{i,t} - s_{i,t}\|^2 \right]$$

$$+ C_f C_g \left[ \frac{1}{n_+} \sum_{X_i \in S_+} \|h_i(\mathbf{w}_t) - v_{i,t}\| + \frac{1}{n_-} \sum_{X_j \in S_-} \|h_j(\mathbf{w}_t) - v_{j,t}\| \right] + \frac{C_f C_g L_h}{2} \|\hat{\mathbf{w}}_t - \mathbf{w}_t\|^2$$

$$\tag{40}$$

Combining inequality 37 and 40 yields

$$\mathbb{E}_t[F_{1/\bar\rho}(\mathbf{w}_{t+1}, \mathbf{s}_{t+1}, s'_{t+1})]$$

$$= F_{1/\bar\rho}(\mathbf{w}_t, \mathbf{s}_t, s'_t) + \bar\rho\eta\left[\langle\hat{\mathbf{w}}_t - \mathbf{w}_t, \mathbb{E}_t[G_t]\rangle + \langle\hat{\mathbf{s}}_t - \mathbf{s}_t, \mathbb{E}_t[G_t^1]\rangle + \langle\hat{s}'_t - s'_t, \mathbb{E}_t[G_t^2]\rangle\right]$$

$$+ \frac{3\eta^2\bar\rho M^2}{2}$$

$$\overset{(a)}{\leq} F_{1/\bar\rho}(\mathbf{w}_t, \mathbf{s}_t, s'_t) + \frac{3\eta^2\bar\rho M^2}{2} + \bar\rho\eta\Bigg\{\frac{1}{n_+}\sum_{X_i \in S_+}\Bigg[F_i(\hat{s}'_t, \hat{\mathbf{w}}_t, \hat{s}_{i,t}) - F_i(s'_t, \mathbf{w}_t, s_{i,t})$$

$$+ C_f C_g \frac{1}{n_-}\sum_{X_j \in S_-}\left[\|h_i(\mathbf{w}_t) - v_{i,t}\| + \|h_j(\mathbf{w}_t)) - v_{j,t}\|\right]$$

$$+ C_f\left\|\frac{1}{n_-}\sum_{X_j \in S_-}g_i(v_{j,t} - v_{i,t}, s_{i,t}) - u_{i,t}\right\| + C_f\left\|\frac{1}{n_-}\sum_{X_j \in S_-}g_i(v_{j,t} - v_{i,t}, s_{i,t}) - u_{i,t}\right\|$$

$$+ \frac{1}{n_-}\sum_{X_j \in S_-}\rho_g C_f\left[\|(h_i(\hat{\mathbf{w}}_t) - v_{i,t}\|^2 + \|h_j(\hat{\mathbf{w}}_t)) - v_{j,t}\|^2\right] + \frac{\rho_g C_f}{2}\|\hat{s}_{i,t} - s_{i,t}\|^2\Bigg]$$

$$+ C_f C_g\left[\frac{1}{n_+}\sum_{X_i \in S_+}\|h_i(\mathbf{w}_t) - v_{i,t}\| + \frac{1}{n_-}\sum_{X_j \in S_-}\|h_j(\mathbf{w}_t) - v_{j,t}\|\right] + \frac{C_f C_g L_h}{2}\|\hat{\mathbf{w}}_t - \mathbf{w}_t\|^2\Bigg\}$$

$$= F_{1/\bar\rho}(\mathbf{w}_t, \mathbf{s}_t, s'_t) + \frac{3\eta^2\bar\rho M^2}{2} + \bar\rho\eta(F(\hat{\mathbf{w}}_t, \hat{\mathbf{s}}_t, \hat{s}'_t) - F(\mathbf{w}_t, \mathbf{s}_t, s'_t))$$

$$+ \bar\rho\eta\Bigg\{\frac{1}{n_+}\sum_{X_i \in S_+}\Bigg[\frac{2C_f C_g}{n_-}\sum_{X_j \in S_-}\left[\|h_i(\mathbf{w}_t) - v_{i,t}\| + \|h_j(\mathbf{w}_t)) - v_{j,t}\|\right]$$

$$+ \frac{2\rho_g C_f}{n_-}\sum_{X_j \in S_-}\left[\|h_i(\mathbf{w}_t) - v_{i,t}\|^2 + \|h_j(\mathbf{w}_t)) - v_{j,t}\|^2\right] + 4\rho_g C_f C_h\|\hat{\mathbf{w}}_t - \mathbf{w}_t\|^2$$

$$+ 2C_f\left\|\frac{1}{n_-}\sum_{X_j \in S_-}g_i(v_{j,t} - v_{i,t}, s_{i,t}) - u_{i,t}\right\| + \frac{\rho_g C_f}{2}\|\hat{s}_{i,t} - s_{i,t}\|^2\Bigg] + \frac{C_f C_g L_h}{2}\|\hat{\mathbf{w}}_t - \mathbf{w}_t\|^2\Bigg\}$$

$$\tag{41}$$

where (a) follows from the Lipschitz continuity of $f_i, g_i, h_i, h_j$ and inequality 40.

Due to the $\rho_F$-weak convexity of $F(\mathbf{w}, \mathbf{s}_i, s')$, we have $(\bar\rho - \rho_F)$-strong convexity of $(\mathbf{w}, \mathbf{s}_i, s') \mapsto F(\mathbf{w}, \mathbf{s}, s') + \frac{\bar\rho}{2}\|(\mathbf{w}_t, \mathbf{s}_t, s'_t) - (\mathbf{w}, \mathbf{s}, s')\|^2$. Then it follows

$$\begin{aligned}
F(\hat{\mathbf{w}}_t, \hat{\mathbf{s}}_t, \hat{s}'_t) - F_i(\mathbf{w}_t, \mathbf{s}_t, s'_t) &= \left[F_i(\hat{\mathbf{w}}_t, \hat{\mathbf{s}}_t, \hat{s}'_t) + \frac{\bar\rho}{2}\|(\mathbf{w}_t, \mathbf{s}_t, s'_t) - (\hat{\mathbf{w}}_t, \hat{\mathbf{s}}_t, \hat{s}'_t)\|^2\right] \\
&\quad - \left[F_i(\mathbf{w}_t, \mathbf{s}_t, s'_t) + \frac{\bar\rho}{2}\|(\mathbf{w}_t, \mathbf{s}_t, s'_t) - (\mathbf{w}_t, \mathbf{s}_t, s'_t)\|^2\right] \\
&\quad - \frac{\bar\rho}{2}\|(\mathbf{w}_t, \mathbf{s}_t, s'_t) - (\hat{\mathbf{w}}_t, \hat{\mathbf{s}}_t, \hat{s}'_t)\|^2 \\
&\leq (\frac{\rho_F}{2} - \bar\rho)\|(\mathbf{w}_t, \mathbf{s}_t, s'_t) - (\hat{\mathbf{w}}_t, \hat{\mathbf{s}}_t, \hat{s}'_t)\|^2
\end{aligned} \tag{42}$$

Plugging inequality 42 back into 41, we obtain

$$\mathbb{E}_t[F_{1/\bar{\rho}}(\mathbf{w}_{t+1}, \mathbf{s}_{t+1}, s'_{t+1})]$$

$$\leq F_{1/\bar{\rho}}(\mathbf{w}_t, \mathbf{s}_t, s'_t) + \frac{3\eta^2 \bar{\rho} M^2}{2} + \bar{\rho}\eta \left\{ \frac{1}{n_+} \sum_{X_i \in S_+} \left[ (\frac{\rho_F}{2} - \bar{\rho}) \|(\mathbf{w}_t, \mathbf{s}_t, s'_t) - (\hat{\mathbf{w}}_t, \hat{\mathbf{s}}_t, \hat{s}'_t)\|^2 \right. \right.$$

$$+ \frac{2C_f C_g}{n_-} \sum_{X_j \in S_-} \left[ \|h_i(\mathbf{w}_t) - v_{i,t}\| + \|h_j(\mathbf{w}_t) - v_{j,t}\| \right]$$

$$+ \frac{2\rho_g C_f}{n_-} \sum_{X_j \in S_-} \left[ \|h_i(\mathbf{w}_t) - v_{i,t}\|^2 + \|h_j(\mathbf{w}_t) - v_{j,t}\|^2 \right]$$

$$+ 2C_f \left\| \frac{1}{n_-} \sum_{X_j \in S_-} g_i(v_{j,t} - v_{i,t}, s_{i,t}) - u_{i,t} \right\| + \frac{\rho_g C_f}{2} \|\hat{s}_{i,t} - s_{i,t}\|^2 \right] + (4\rho_g C_f C_h + \frac{C_f C_g L_h}{2}) \|\hat{\mathbf{w}}_t - \mathbf{w}_t\|^2 \bigg\}$$

$$\leq F_{1/\bar{\rho}}(\mathbf{w}_t, \mathbf{s}_t, s'_t) + \frac{3\eta^2 \bar{\rho} M^2}{2} + \bar{\rho}\eta \left\{ \frac{1}{n_+} \sum_{X_i \in S_+} \left[ -\frac{\bar{\rho}}{2} \|(\mathbf{w}_t, \mathbf{s}_t, s'_t) - (\hat{\mathbf{w}}_t, \hat{\mathbf{s}}_t, \hat{s}'_t)\|^2 \right. \right.$$

$$+ \frac{C_1}{n_-} \sum_{X_j \in S_-} \left[ \|h_i(\mathbf{w}_t) - v_{i,t}\| + \|h_j(\mathbf{w}_t) - v_{j,t}\| + \|h_i(\mathbf{w}_t) - v_{i,t}\|^2 + \|h_j(\mathbf{w}_t) - v_{j,t}\|^2 \right]$$

$$+ C_1 \left\| \frac{1}{n_-} \sum_{X_j \in S_-} g_i(v_{j,t} - v_{i,t}, s_{i,t}) - u_{i,t} \right\| \bigg] \bigg\}$$

$$\overset{(b)}{\leq} F_{1/\bar{\rho}}(\mathbf{w}_t, \mathbf{s}_t, s'_t) + \frac{3\eta^2 \bar{\rho} M^2}{2} - \frac{\eta}{2} \|\nabla F_{1/\bar{\rho}}(\mathbf{w}_t, \mathbf{s}_t, s'_t)\|^2$$

$$+ \frac{\bar{\rho}\eta C_1}{n_+ n_-} \sum_{X_i \in S_+} \sum_{X_j \in S_-} \left[ \|h_i(\mathbf{w}_t) - v_{i,t}\| + \|h_j(\mathbf{w}_t) - v_{j,t}\| + \|h_i(\mathbf{w}_t) - v_{i,t}\|^2 + \|h_j(\mathbf{w}_t) - v_{j,t}\|^2 \right]$$

$$+ \frac{\bar{\rho}\eta C_1}{n_+} \sum_{X_i \in S_+} \left\| \frac{1}{n_-} \sum_{X_j \in S_-} g_i(v_{j,t} - v_{i,t}, s_{i,t}) - u_{i,t} \right\|$$

where in inequality (a) we use $\bar{\rho} = \rho_F + \rho_g C_f + 8\rho_g C_f C_h + C_f C_g L_h$ and $C_1 = \max\{2C_f C_g, 2\rho_g C_f, 2C_f\}$, and inequality (b) uses Lemma 3.2.

With general error bounds

$$\frac{1}{n_+} \sum_{X_i \in S_+} \mathbb{E}[\|h_i(\mathbf{w}_t) - v_{i,t}\|] \leq (1 - \mu_1)^t \frac{1}{n_+} \sum_{X_i \in S_+} \|h_i(\mathbf{w}_0) - v_{i,0}\| + R_1,$$

$$\frac{1}{n_-} \sum_{X_j \in S_-} \mathbb{E}[\|h_j(\mathbf{w}_t) - v_{j,t}\|] \leq (1 - \mu_2)^t \frac{1}{n_-} \sum_{X_j \in S_-} \|h_j(\mathbf{w}_0) - v_{j,0}\| + R_2,$$

$$\frac{1}{n_+} \sum_{X_i \in S_+} \mathbb{E}[\|h_i(\mathbf{w}_t) - v_{i,t}\|^2] \leq (1 - \mu_1)^t \frac{1}{n_+} \sum_{X_i \in S_+} \|h_i(\mathbf{w}_0) - v_{i,0}\|^2 + R_3,$$

$$\frac{1}{n_-} \sum_{X_j \in S_-} \mathbb{E}[\|h_j(\mathbf{w}_t) - v_{j,t}\|^2] \leq (1 - \mu_2)^t \frac{1}{n_-} \sum_{X_j \in S_-} \|h_j(\mathbf{w}_0) - v_{j,0}\|^2 + R_4,$$

$$\frac{1}{n_+} \sum_{X_i \in S_+} \mathbb{E}\left[ \left\| \frac{1}{n_-} \sum_{X_j \in S_-} g_i(v_{j,t} - v_{i,t}, s_{i,t}) - u_{i,t} \right\| \right]$$

$$\leq (1 - \mu_3)^t \frac{1}{n_+} \sum_{X_i \in S_+} \left\| \frac{1}{n_-} \sum_{X_j \in S_-} g_i(v_{i,0} - v_{j,0}, s_{i,0}) - u_{i,0} \right\| + R_5,$$

we have
$$\mathbb{E}[F_{1/\bar{\rho}}(\mathbf{w}_{t+1}, \mathbf{s}_{t+1}, s'_{t+1})]$$

$$\leq \mathbb{E}[F_{1/\bar{\rho}}(\mathbf{w}_t, \mathbf{s}_t, s'_t)] + \frac{3\eta^2 \bar{\rho} M^2}{2} - \frac{\eta}{2}\mathbb{E}[\|\nabla F_{1/\bar{\rho}}(\mathbf{w}_t, \mathbf{s}_t, s'_t)\|^2]$$

$$+ \bar{\rho}\eta C_1\Bigg[(1-\mu_1)^t\frac{1}{n_+}\sum_{X_i \in S_+}\|h_i(\mathbf{w}_0) - v_{i,0}\| + (1-\mu_2)^t\frac{1}{n_-}\sum_{X_j \in S_-}\|h_j(\mathbf{w}_0) - v_{j,0}\|$$

$$+ (1-\mu_1)^t\frac{1}{n_+}\sum_{X_i \in S_+}\|h_i(\mathbf{w}_0) - v_{i,0}\|^2 + (1-\mu_2)^t\frac{1}{n_-}\sum_{X_j \in S_-}\|h_j(\mathbf{w}_0) - v_{j,0}\|^2$$

$$+ (1-\mu_3)^t\frac{1}{n_+}\sum_{X_i \in S_+}\left\|\frac{1}{n_-}\sum_{X_j \in S_-}g_i(v_{i,0} - v_{j,0}, s_{i,0}) - u_{i,0}\right\| + R_1 + R_2 + R_3 + R_4 + R_5\Bigg]$$

$$\leq \mathbb{E}[F_{1/\bar{\rho}}(\mathbf{w}_t, \mathbf{s}_t, s'_t)] + \frac{3\eta^2 \bar{\rho} M^2}{2} - \frac{\eta}{2}\mathbb{E}[\|\nabla F_{1/\bar{\rho}}(\mathbf{w}_t, \mathbf{s}_t, s'_t)\|^2]$$

$$+ \bar{\rho}\eta C_1\Bigg[(1-\mu_{min})^t\Big(\frac{1}{n_+}\sum_{X_i \in S_+}\|h_i(\mathbf{w}_0) - v_{i,0}\| + \frac{1}{n_-}\sum_{X_j \in S_-}\|h_j(\mathbf{w}_0) - v_{j,0}\|$$

$$+ \frac{1}{n_+}\sum_{X_i \in S_+}\|h_i(\mathbf{w}_0) - v_{i,0}\|^2 + \frac{1}{n_-}\sum_{X_j \in S_-}\|h_j(\mathbf{w}_0) - v_{j,0}\|^2$$

$$+ \frac{1}{n_+}\sum_{X_i \in S_+}\left\|\frac{1}{n_-}\sum_{X_j \in S_-}g_i(v_{i,0} - v_{j,0}, s_{i,0}) - u_{i,0}\right\|\Big) + R_1 + R_2 + R_3 + R_4 + R_5\Bigg]$$

where $\mu_{min} = \min\{\mu_1, \mu_2, \mu_3\}$.

Taking summation from $t = 0$ to $T - 1$ yields
$$\mathbb{E}[F_{1/\bar{\rho}}(\mathbf{w}_T, \mathbf{s}_T, s'_T)]$$

$$\leq F_{1/\bar{\rho}}(\mathbf{w}_0, \mathbf{s}_0, s'_0) + \frac{3\eta^2 \bar{\rho} M^2 T}{2} - \frac{\eta}{2}\sum_{t=0}^{T-1}\mathbb{E}[\|\nabla F_{1/\bar{\rho}}(\mathbf{w}_t, \mathbf{s}_t, s'_t)\|^2]$$

$$+ \bar{\rho}\eta C_1\Bigg[\sum_{t=0}^{T-1}(1-\mu_{min})^t\Big(\frac{1}{n_+}\sum_{X_i \in S_+}\|h_i(\mathbf{w}_0) - v_{i,0}\| + \frac{1}{n_-}\sum_{X_j \in S_-}\|h_j(\mathbf{w}_0) - v_{j,0}\|$$

$$+ \frac{1}{n_+}\sum_{X_i \in S_+}\|h_i(\mathbf{w}_0) - v_{i,0}\|^2 + \frac{1}{n_-}\sum_{X_j \in S_-}\|h_j(\mathbf{w}_0) - v_{j,0}\|^2$$

$$+ \frac{1}{n_+}\sum_{X_i \in S_+}\left\|\frac{1}{n_-}\sum_{X_j \in S_-}g_i(v_{i,0} - v_{j,0}, s_{i,0}) - u_{i,0}\right\|\Big) + T(R_1 + R_2 + R_3 + R_4 + R_5)\Bigg]$$

$$\leq F_{1/\bar{\rho}}(\mathbf{w}_0, \mathbf{s}_0, s'_0) + \frac{3\eta^2 \bar{\rho} M^2 T}{2} - \frac{\eta}{2}\sum_{t=0}^{T-1}\|\nabla F_{1/\bar{\rho}}(\mathbf{w}_t, \mathbf{s}_t, s'_t)\|^2$$

$$+ \bar{\rho}\eta C_1\left[\frac{\Delta_0}{\mu_{min}} + T(R_1 + R_2 + R_3 + R_4 + R_5)\right]$$

where we use $\sum_{t=0}^{T-1}(1-\mu_{min})^t \leq \frac{1}{\mu_{min}}$ and define constant $\Delta_0$ such that

$$\Big(\frac{1}{n_+}\sum_{X_i \in S_+}\|h_i(\mathbf{w}_0) - v_{i,0}\| + \frac{1}{n_-}\sum_{X_j \in S_-}\|h_j(\mathbf{w}_0) - v_{j,0}\|$$

$$+ \frac{1}{n_+}\sum_{X_i \in S_+}\|h_i(\mathbf{w}_0) - v_{i,0}\|^2 + \frac{1}{n_-}\sum_{X_j \in S_-}\|h_j(\mathbf{w}_0) - v_{j,0}\|^2$$

$$+ \frac{1}{n_+}\sum_{X_i \in S_+}\left\|\frac{1}{n_-}\sum_{X_j \in S_-}g_i(v_{i,0} - v_{j,0}, s_{i,0}) - u_{i,0}\right\|\Big) \leq \Delta_0.$$

Then it follows

$$\frac{1}{T}\sum_{t=0}^{T-1}\|\nabla F_{1/\bar{\rho}}(\mathbf{w}_t,\mathbf{s}_t,s_t')\|^2$$

$$\leq \frac{2}{\eta T}\left[F_{1/\bar{\rho}}(\mathbf{w}_0,\mathbf{s}_0,s_0') - \mathbb{E}[F_{1/\bar{\rho}}(\mathbf{w}_T,\mathbf{s}_T,s_T')] + \frac{3\eta^2\bar{\rho}M^2 T}{2}\right.$$

$$\left. + \bar{\rho}\eta C_1\left[\frac{\Delta_0}{\mu_{min}} + T(R_1+R_2+R_3+R_4+R_5)\right]\right]$$

$$\leq \frac{2\Delta}{\eta T} + (2+\frac{n_+}{B_1})\eta\bar{\rho}M^2 + \frac{2\bar{\rho}C_1\Delta_0}{\mu_{min}T} + 2\bar{\rho}C_1(R_1+R_2+R_3+R_4+R_5)$$

$$= \mathcal{O}\left(\frac{1}{T}(\frac{1}{\eta}+\frac{1}{\mu_{min}})+\eta+R_1+R_2+R_3+R_4+R_5\right)$$

where we define constant $\Delta$ such that $F_{1/\bar{\rho}}(\mathbf{w}_0,\mathbf{s}_0,s_0') - \mathbb{E}[F_{1/\bar{\rho}}(\mathbf{w}_T,\mathbf{s}_T,s_T')] \leq \Delta$.

With MSVR updates for $v_{i,t}$ and $u_{i,t}$, following from Lemma C.5 and Lemma C.6, we have

$$\mu_1 = \frac{B_1\tau_1}{2n_+}, \quad \mu_2 = \frac{B_1\tau_1}{2n_-}, \quad \mu_3 = \frac{B_1\tau_2}{2n_+}$$

$$R_1 = \frac{2\tau_1^{1/2}\sigma}{B_3^{1/2}} + \frac{4n_+ C_h M\eta}{B_1\tau_1^{1/2}}, \quad R_2 = \frac{2\tau_1^{1/2}\sigma}{B_3^{1/2}} + \frac{4n_- C_h M\eta}{B_2\tau_1^{1/2}}$$

$$R_3 = \frac{4\tau_1\sigma^2}{B_3} + \frac{16n_+^2 C_h^2 M^2\eta^2}{B_1^2\tau_1}, \quad R_4 = \frac{4\tau_1\sigma^2}{B_3} + \frac{16n_-^2 C_h^2 M^2\eta^2}{B_2^2\tau_1}$$

$$R_5 = \frac{2\tau_2^{1/2}\sigma}{B_2^{1/2}} + C_2\frac{n_+}{B_1}(\frac{B_1^{1/2}}{n_+^{1/2}}+\frac{B_2^{1/2}}{n_-^{1/2}})\frac{\tau_1}{\tau_2^{1/2}} + C_2\frac{n_+}{B_1}(\frac{n_+^{1/2}}{B_1^{1/2}}+\frac{n_-^{1/2}}{B_2^{1/2}})\frac{\eta}{\tau_2^{1/2}} + C_2\frac{n_+^{1/2}\eta}{B_1\tau_2^{1/2}}$$

Then we have

$$\frac{1}{T}\sum_{t=0}^{T-1}\|\nabla F_{1/\bar{\rho}}(\mathbf{w}_t,\mathbf{s}_t,s_t')\|^2$$

$$\leq \mathcal{O}\left(\frac{1}{T}(\frac{1}{\eta}+\frac{1}{\mu_{min}})+(\frac{\tau_1^{1/2}}{B_3^{1/2}}+\frac{\tau_2^{1/2}}{B_2^{1/2}})\sigma\right.$$

$$\left. + \frac{n_+\eta}{B_1\tau_1^{1/2}}+\frac{n_-\eta}{B_2\tau_1^{1/2}}+\frac{n_+}{B_1}\max\{\frac{B_1^{1/2}}{n_+^{1/2}},\frac{B_2^{1/2}}{n_-^{1/2}}\}\frac{\tau_1}{\tau_2^{1/2}}+\frac{n_+}{B_1}\max\{\frac{n_+^{1/2}}{B_1^{1/2}},\frac{n_-^{1/2}}{B_2^{1/2}}\}\frac{\eta}{\tau_2^{1/2}}\right)$$

Setting

$$\tau_1 = \mathcal{O}\left(\min\left\{B_3,\frac{B_1}{n_+}\min\{\frac{n_+^{1/2}}{B_1^{1/2}},\frac{n_-^{1/2}}{B_2^{1/2}}\}B_2^{1/2}\right\}\epsilon^4\right), \quad \tau_2 = \mathcal{O}(B_2\epsilon^4),$$

$$\eta = \mathcal{O}\left(\min\left\{\min\{\frac{B_1}{n_+},\frac{B_2}{n_-}\}\min\{B_3^{1/2},\frac{B_1^{1/2}}{n_+^{1/2}}\min\{\frac{n_+^{1/4}}{B_1^{1/4}},\frac{n_-^{1/4}}{B_2^{1/4}}\}B_2^{1/4}\},\frac{B_1}{n_+}\min\{\frac{B_1^{1/2}}{n_+^{1/2}},\frac{B_2^{1/2}}{n_-^{1/2}}\}B_3^{1/2}\right\}\epsilon^4\right),$$

Then with

$$T \geq \mathcal{O}\left(\max\left\{\max\{\frac{n_+}{B_1},\frac{n_-}{B_2}\}\max\{\frac{1}{B_3^{1/2}},\frac{n_+^{1/2}}{B_1^{1/2}}\max\{\frac{B_1^{1/4}}{n_+^{1/4}},\frac{B_2^{1/4}}{n_-^{1/4}}\}\frac{1}{B_2^{1/4}}\},\frac{n_+}{B_1}\max\{\frac{n_+^{1/2}}{B_1^{1/2}},\frac{n_-^{1/2}}{B_2^{1/2}}\}\frac{1}{B_2^{1/2}}\right\}\epsilon^{-6}\right)$$

iterations, we have

$$\frac{1}{T}\sum_{t=0}^{T-1}\|\nabla F_{1/\bar{\rho}}(\mathbf{w}_t,\mathbf{s}_t,s_t')\|^2 \leq \epsilon^2.$$

$\square$

# D Proofs of Lemmas and Propositions

## D.1 Additional Proposition

**Proposition D.1.** *Consider a Lipschitz continuous function $f : O \to \mathbb{R}$ where $O \subset \mathbb{R}^d$ is an open set. Assume $f$ to be non-increasing (resp. non-decreasing) with respect to each element in the input, then all subgradients of $f$ are element-wise non-positive (resp. non-negative).*

*Proof of Proposition D.1.* Let $D$ be the subset of $O$ where $f$ is differentiable. By Theorem 9.60 in [24], a Lipschitz continuous function $f : O \to \mathbb{R}$, where $O \subset \mathbb{R}^d$ is an open set, is differentiable almost everywhere, i.e., $D$ is dense in $O$. Then by Theorem 9.61 in [24], the subdifferential of $f$ at $x$ is defined as

$$\partial f(x) = \text{con}\{v | \exists x_k \to x \text{ with } x_k \in D, \nabla f(x_k) \to v\},$$

where con denotes the convex hull. If we assume that $f$ is non-increasing with respect to each element in the input, then $\nabla f(x) \leq 0$ (element-wise) for all differentiable points $x \in D$. It implies that the all vectors in $\{v | \exists x_k \to x \text{ with } x_k \in D, \nabla f(x_k) \to v\}$ are element-wise non-positive. Therefore, all subgradients of $f$ are element-wise non-positive. On the other hand, if we assume that $f$ is non-decreasing, one may follow the same argument and conclude that all subgradients of $f$ are element-wise non-negative. $\qquad\square$

For functions $f : O \to \mathbb{R}^m$ where $O \subset \mathbb{R}^d$ is an open set, one may write $f = (f_1, \ldots, f_m)$ and apply the above proposition for each $f_k, k = 1, \ldots, m$.

## D.2 Proofs of Proposition 4.2 and Proposition 4.4

To prove Proposition 4.2 and Proposition 4.4, we first present the following proposition on the weak-convexity of composition functions.

**Proposition D.2.** *Assume $f : \mathbb{R}^d \to \mathbb{R}$ is $\rho_1$-weakly-convex and $C_1$-Lipschitz continuous, $g : \mathbb{R}^{\bar{d}} \to \mathbb{R}^d$ is $C_2$-Lipschitz continuous, and either of the followings holds:*

1. *$f(\cdot)$ is monotone and $g(\cdot)$ is $L_2$-smooth;*

2. *$f(\cdot)$ is non-decreasing and $g(\cdot)$ is $L_2$-weakly-convex,*

*then $f \circ g$ is $\tilde{\rho}$-weakly-convex with $\tilde{\rho} = \sqrt{d}L_2C_1 + \rho_1 C_2^2$.*

*Proof of Proposition D.2.* The weak convexity of $f$ implies

$$f(g(y)) \geq f(g(x)) + v^\top (g(y) - g(x)) - \frac{\rho_1}{2}\|g(y) - g(x)\|^2$$

$$\geq f(g(x)) + v^\top (g(y) - g(x)) - \frac{\rho_1 C_2^2}{2}\|x - y\|^2$$

where $v \in \partial f(g(x))$. Moreover, due to the smoothness of $g(\cdot)$ (or weakly-convexity of $g(\cdot)$, then only the second inequality holds), we have

$$g(y) - g(x) \leq \nabla g(x)^\top (y - x) + \mathbf{v}\left(\frac{L_2}{2}\|x - y\|^2\right),$$
$$g(y) - g(x) \geq \nabla g(x)^\top (y - x) - \mathbf{v}\left(\frac{L_2}{2}\|x - y\|^2\right). \tag{43}$$

where $\mathbf{v}(e)$ denotes a $d$-dimensional vector with value $e$ on each dimensions. We first assume that $f$ is non-increasing, then we may use the first inequality in (43) and the Lipschitz continuity of $g$ to get

$$f(g(y)) \geq f(g(x)) + v^\top \left[\nabla g(x)^\top (y - x) + \mathbf{v}\left(\frac{L_2}{2}\|x - y\|^2\right)\right] - \frac{\rho_1 C_2^2}{2}\|x - y\|^2$$

$$\geq f(g(x)) + v^\top \nabla g(x)^\top (y - x) + v^\top \mathbf{v}\left(\frac{L_2}{2}\|x - y\|^2\right) - \frac{\rho_1 C_2^2}{2}\|x - y\|^2$$

$$\geq f(g(x)) + \langle v^\top \nabla g(x)^\top (y - x) - \frac{\sqrt{d}L_2C_1 + \rho_1 C_2^2}{2}\|x - y\|^2.$$

On the other hand, if we assume $f$ is non-decreasing, the same result follows from the second inequality in (43). Thus $f \circ g$ is $\tilde{\rho}$-weakly-convex with $\tilde{\rho}_g = \sqrt{d}L_2C_1 + \rho_1C_2^2$. $\qquad \square$

*Proof of Proposition 4.2.* Under Assumption 4.1, Proposition D.2 directly implies the $\rho_F$-weak-convexity of $F(\mathbf{w})$ with $\rho_F = \sqrt{d_1}\rho_gC_f + \rho_fC_g^2$. $\qquad \square$

*Proof of Proposition 4.4.* Under Assumption 4.3, we first apply Proposition D.2 to the composite function $g_i(h_{i,j}(\cdot))$ and obtain its $\rho_{\tilde{g}} = \sqrt{d_2}L_hC_g + \rho_gC_h^2$-weak-convexity. To show it Lipschitz continuity, we use the Lipschitz continuity of $g_i$ and $h_{i,j}$ to obtain

$$\|g_i(h_{i,j}(\mathbf{w})) - g_i(h_{i,j}(\tilde{\mathbf{w}}))\|^2 \leq C_g^2C_h^2\|\mathbf{w} - \tilde{\mathbf{w}}\|^2.$$

Thus $g_i(h_{i,j}(\mathbf{w}))$ is $C_{\tilde{g}} = C_gC_h$-Lipschitz-continuous

Since we assume $f_i(\cdot)$ is non-decreasing, $\rho_f$-weakly-convex and $C_f$-Lipschitz continuous, and $g_i(h_{i,j}(\cdot))$ is $\rho_{\tilde{g}}$-weakly-convex and $C_{\tilde{g}}$-Lipschitz-continuous, we apply Proposition D.2 again to conclude that $F(\cdot)$ is $\rho_F = \sqrt{d_1}\rho_{\tilde{g}}C_f + \rho_fC_{\tilde{g}}^2$-weakly-convex. $\qquad \square$

## D.3 Proof of Lemma 4.5

*Proof of Lemma 4.5.* With $\gamma = \frac{n_1 - B_1}{B_1(1-\tau)} + (1-\tau)$, $\tau \leq \frac{1}{2}$, MSVR update gives recursive error bound [15]

$$\mathbb{E}[\|u_{i,t+1} - g_i(\mathbf{w}_{t+1})\|^2]$$

$$\leq (1 - \frac{B_1\tau}{n_1})\mathbb{E}[\|u_{i,t} - g_i(\mathbf{w}_t)\|^2] + \frac{2\tau^2 B_1\sigma^2}{n_1 B_2} + \frac{8n_1 C_g^2}{B_1}\mathbb{E}[\|\mathbf{w}_t - \mathbf{w}_{t+1}\|^2]$$

$$\leq (1 - \frac{B_1\tau}{n_1})\mathbb{E}[\|u_{i,t} - g_i(\mathbf{w}_t)\|^2] + \frac{2\tau^2 B_1\sigma^2}{n_1 B_2} + \frac{8n_1 C_g^2}{B_1}\eta^2\mathbb{E}[\|G_t\|^2]$$

$$\leq (1 - \frac{B_1\tau}{2n_1})^2\mathbb{E}[\|u_{i,t} - g_i(\mathbf{w}_t)\|^2] + \frac{2\tau^2 B_1\sigma^2}{n_1 B_2} + \frac{8n_1 C_g^2 M^2\eta^2}{B_1}$$

Applying this inequality recursively, we obtain

$$\mathbb{E}[\|u_{i,t+1} - g_i(\mathbf{w}_{t+1})\|^2]$$

$$\leq (1 - \frac{B_1\tau}{2n_1})^{2(t+1)}\|u_{i,0} - g_i(\mathbf{w}_0)\|^2 + \sum_{j=0}^{t}(1 - \frac{B_1\tau}{2n_1})^{2(t-j)}\left(\frac{2\tau^2 B_1\sigma^2}{n_1 B_2} + \frac{8n_+ C_g^2 M^2\eta^2}{B_1}\right)$$

$$\leq (1 - \frac{B_1\tau}{2n_1})^{2(t+1)}\|u_{i,0} - g_i(\mathbf{w}_0)\|^2 + \frac{4\tau\sigma^2}{B_2} + \frac{16n_1^2 C_g^2 M^2\eta^2}{B_1^2\tau}$$

where we use $\sum_{j=0}^{t}(1 - \frac{B_1\tau}{2n_1})^{2(t-j)} \leq \frac{2n_1}{B_1\tau}$.

It follows

$$\mathbb{E}\left[\|u_{i,t+1} - g_i(\mathbf{w}_{t+1})\|\right]^2$$

$$\leq \mathbb{E}[\|u_{i,t+1} - g_i(\mathbf{w}_{t+1})\|^2]$$

$$\leq (1 - \frac{B_1\tau}{2n_1})^{2(t+1)}\|u_{i,0} - g_i(\mathbf{w}_0)\|^2 + \frac{4\tau\sigma^2}{B_2} + \frac{16n_1^2 C_g^2 M^2\eta^2}{B_1^2\tau}$$

$$\leq \left[(1 - \frac{B_1\tau}{2n_1})^{t+1}\|u_{i,0} - g_i(\mathbf{w}_0)\| + \frac{2\tau^{1/2}\sigma}{B_2^{1/2}} + \frac{4n_1 C_g M\eta}{B_1\tau^{1/2}}\right]^2$$

Thus

$$\mathbb{E}\left[\|u_{i,t+1} - g_i(\mathbf{w}_{t+1})\|\right]$$

$$\leq (1 - \frac{B_1\tau}{2n_1})^{t+1}\|u_{i,0} - g_i(\mathbf{w}_0)\| + \frac{2\tau^{1/2}\sigma}{B_2^{1/2}} + \frac{4n_1 C_g M\eta}{B_1\tau^{1/2}}$$

Taking summation over $i \in \mathcal{S}$, we obtain the desired result

$$\mathbb{E}\left[\frac{1}{n}\sum_{i\in\mathcal{S}}\|u_{i,t+1} - g_i(\mathbf{w}_{t+1})\|\right]$$

$$\leq (1 - \frac{B_1\tau}{2n_1})^{t+1}\frac{1}{n}\sum_{i\in\mathcal{S}}\|u_{i,0} - g_i(\mathbf{w}_0)\| + \frac{2\tau^{1/2}\sigma}{B_2^{1/2}} + \frac{4nC_gM\eta}{B_1\tau^{1/2}}$$

$\square$

## D.4 Proof of Lemma C.6

*Proof of Lemma C.6.* With $\gamma_3 = \frac{n_+ - B_1}{B_1(1-\tau_2)} + (1-\tau_2)$ and $\tau_2 \leq \frac{1}{2}$, MSVR update gives the following recursive error bound [15]

$$\mathbb{E}[\|u_{i,t+1} - \frac{1}{n_-}\sum_{j\in S_-} g_i(v_{j,t+1} - v_{i,t+1}, s_{i,t+1})\|^2]$$

$$\leq (1 - \frac{B_1\tau_2}{n_+})\mathbb{E}[\|u_{i,t} - \frac{1}{n_-}\sum_{j\in S_-} g_i(v_{j,t} - v_{i,t}, s_{i,t})\|^2] + \frac{2\tau_2^2 B_1\sigma^2}{n_+ B_2}$$

$$+ \frac{8n_+ C_g^2}{B_1}\mathbb{E}[\|(v_{j,t} - v_{i,t}, s_{i,t}) - (v_{j,t+1} - v_{i,t+1}, s_{i,t+1})\|^2] \qquad (44)$$

$$\leq (1 - \frac{B_1\tau_2}{n_+})\mathbb{E}[\|u_{i,t} - \frac{1}{n_-}\sum_{j\in S_-} g_i(v_{j,t} - v_{i,t}, s_{i,t})\|^2] + \frac{2\tau_2^2 B_1\sigma^2}{n_+ B_2}$$

$$+ \frac{16n_+ C_g^2}{B_1}\mathbb{E}[\|v_{i,t} - v_{i,t+1}\|^2 + \|v_{j,t} - v_{j,t+1}\|^2] + \frac{8C_g^2 M^2\eta^2}{B_1}$$

It remains to bound $\mathbb{E}[\|v_{i,t} - v_{i,t+1}\|^2]$ and $\mathbb{E}[\|v_{j,t} - v_{j,t+1}\|^2]$. We bound the former, and the latter's bound naturally follows. Consider the update of $v_{i,t+1}$ and we have

$$\mathbb{E}[\|v_{i,t} - v_{i,t+1}\|^2]$$

$$\leq \mathbb{E}\left[\frac{B_1}{n_+}\|\tau_1 v_{i,t} - \tau_1 h^{(i)}(\mathbf{w}_t; \mathcal{B}_{3,i}^t) - \gamma_1(h^{(i)}(\mathbf{w}_t; \mathcal{B}_{3,i}^t) - h^{(i)}(\mathbf{w}_{t-1}; \mathcal{B}_{3,i}^t))\|^2\right]$$

$$\leq \mathbb{E}\left[\frac{2B_1\tau_1^2}{n_+}\|v_{i,t} - h^{(i)}(\mathbf{w}_t; \mathcal{B}_{3,i}^t)\|^2 + \frac{2B_1\gamma_1^2}{n_+}\|h^{(i)}(\mathbf{w}_t; \mathcal{B}_{3,i}^t) - h^{(i)}(\mathbf{w}_{t-1}; \mathcal{B}_{3,i}^t)\|^2\right]$$

$$\leq \mathbb{E}\left[\frac{2B_1\tau_1^2}{n_+}\|v_{i,t} - h^{(i)}(\mathbf{w}_t; \mathcal{B}_{3,i}^t)\|^2 + \frac{2B_1\gamma_1^2 C_h}{n_+}\|\mathbf{w}_t - \mathbf{w}_{t-1}\|^2\right]$$

$$\overset{(a)}{\leq} \frac{8B_1\tau_1^2 M^2}{n_+} + \frac{8n_+ C_h^2\eta^2 M^2}{B_1}$$

where inequality (a) uses $\tau_1 \leq 1/2$ and $\gamma_1 = \frac{n_+ - B_1}{B_1(1-\tau_1)} + (1-\tau_1) \leq \frac{2n_+}{B_1}$. Plugging the above inequality back into inequality 44 gives

$$\mathbb{E}[\|u_{i,t+1} - \frac{1}{n_-}\sum_{j\in S_-} g_i(v_{j,t+1} - v_{i,t+1}, s_{i,t+1})\|^2]$$

$$\leq (1 - \frac{B_1\tau_2}{n_+})\mathbb{E}[\|u_{i,t} - \frac{1}{n_-}\sum_{j\in S_-} g_i(v_{j,t} - v_{i,t}, s_{i,t})\|^2] + \frac{2\tau_2^2 B_1\sigma^2}{n_+ B_2}$$

$$+ \frac{16n_+ C_g^2}{B_1}\left(8\tau_1^2 M^2(\frac{B_1}{n_+} + \frac{B_2}{n_-}) + 8C_h^2\eta^2 M^2(\frac{n_+}{B_1} + \frac{n_-}{B_2})\right) + \frac{8C_g^2 M^2\eta^2}{B_1}$$

$$\leq (1 - \frac{B_1\tau_2}{n_+})\mathbb{E}[\|u_{i,t} - \frac{1}{n_-}\sum_{j\in S_-} g_i(v_{j,t} - v_{i,t}, s_{i,t})\|^2] + \frac{2\tau_2^2 B_1\sigma^2}{n_+ B_2}$$

$$+ 128C_g^2 M^2\frac{n_+}{B_1}(\frac{B_1}{n_+} + \frac{B_2}{n_-})\tau_1^2 + 128C_g^2 C_h^2 M^2\frac{n_+}{B_1}(\frac{n_+}{B_1} + \frac{n_-}{B_2})\eta^2 + \frac{8C_g^2 M^2\eta^2}{B_1}$$

Applying this inequality recursively, we obtain

$$\mathbb{E}[\|u_{i,t+1} - \frac{1}{n_-}\sum_{j\in S_-} g_i(v_{j,t+1} - v_{i,t+1}, s_{i,t+1})\|^2]$$

$$\leq (1 - \frac{B_1\tau_2}{2n_+})^{2(t+1)}\|u_{i,0} - \frac{1}{n_-}\sum_{j\in S_-} g_i(v_{i,0} - v_{j,0}, s_{i,0})\|^2 + \sum_{j=0}^{t}(1 - \frac{B_1\tau_2}{2n_+})^{2(t-j)}\left(\frac{2\tau_2^2 B_1\sigma^2}{n_+ B_2}\right.$$

$$+ 128C_g^2 M^2 \frac{n_+}{B_1}(\frac{B_1}{n_+} + \frac{B_2}{n_-})\tau_1^2 + 128C_g^2 C_h^2 M^2 \frac{n_+}{B_1}(\frac{n_+}{B_1} + \frac{n_-}{B_2})\eta^2 + \left.\frac{8C_g^2 M^2\eta^2}{B_1}\right)$$

$$\leq (1 - \frac{B_1\tau_2}{2n_+})^{2(t+1)}\|u_{i,0} - \frac{1}{n_-}\sum_{j\in S_-} g_i(v_{i,0} - v_{j,0}, s_{i,0})\|^2 + \frac{4\tau_2\sigma^2}{B_2}$$

$$+ 256C_g^2 M^2 \frac{n_+^2}{B_1^2}(\frac{B_1}{n_+} + \frac{B_2}{n_-})\frac{\tau_1^2}{\tau_2} + 256C_g^2 C_h^2 M^2 \frac{n_+^2}{B_1^2}(\frac{n_+}{B_1} + \frac{n_-}{B_2})\frac{\eta^2}{\tau_2} + \frac{16n_+ C_g^2 M^2\eta^2}{B_1^2\tau_2}$$

$$\leq (1 - \frac{B_1\tau_2}{2n_+})^{2(t+1)}\|u_{i,0} - \frac{1}{n_-}\sum_{j\in S_-} g_i(v_{i,0} - v_{j,0}, s_{i,0})\|^2 + \frac{4\tau_2\sigma^2}{B_2} + C_2^2\frac{n_+^2}{B_1^2}(\frac{B_1}{n_+} + \frac{B_2}{n_-})\frac{\tau_1^2}{\tau_2}$$

$$+ C_2^2\frac{n_+^2}{B_1^2}(\frac{n_+}{B_1} + \frac{n_-}{B_2})\frac{\eta^2}{\tau_2} + C_2^2\frac{n_+\eta^2}{B_1^2\tau_2}$$

where we use $\sum_{j=0}^{t}(1 - \frac{B_1\tau_2}{2n_+})^{2(t-j)} \leq \frac{2n_+}{B_1\tau_1}$ and denotes $C_2^2 = 2\max\{256C_g^2 M^2, 256C_g^2 C_h^2 M^2, 16C_g^2 M^2\}$. Taking average over $i \in S_+$ gives the squared-norm error bound.

To derive the norm error bound, we derive

$$\mathbb{E}[\|u_{i,t+1} - \frac{1}{n_-}\sum_{j\in S_-} g_i(v_{j,t+1} - v_{i,t+1}, s_{i,t+1})\|]^2$$

$$\leq \mathbb{E}[\|u_{i,t+1} - \frac{1}{n_-}\sum_{j\in S_-} g_i(v_{j,t+1} - v_{i,t+1}, s_{i,t+1})\|^2]$$

$$\leq (1 - \frac{B_1\tau_2}{2n_+})^{2(t+1)}\|u_{i,0} - \frac{1}{n_-}\sum_{j\in S_-} g_i(v_{i,0} - v_{j,0}, s_{i,0})\|^2 + \frac{4\tau_2\sigma^2}{B_2} + C_2^2\frac{n_+^2}{B_1^2}(\frac{B_1}{n_+} + \frac{B_2}{n_-})\frac{\tau_1^2}{\tau_2}$$

$$+ C_2^2\frac{n_+^2}{B_1^2}(\frac{n_+}{B_1} + \frac{n_-}{B_2})\frac{\eta^2}{\tau_2} + C_2^2\frac{n_+\eta^2}{B_1^2\tau_2}$$

$$\leq \left[(1 - \frac{B_1\tau_2}{2n_+})^{t+1}\|u_{i,0} - \frac{1}{n_-}\sum_{j\in S_-} g_i(v_{i,0} - v_{j,0}, s_{i,0})\| + \frac{2\tau_2^{1/2}\sigma}{B_2^{1/2}} + C_2\frac{n_+}{B_1}(\frac{B_1^{1/2}}{n_+^{1/2}} + \frac{B_2^{1/2}}{n_-^{1/2}})\frac{\tau_1}{\tau_2^{1/2}}\right.$$

$$+ C_2\frac{n_+}{B_1}(\frac{n_+^{1/2}}{B_1^{1/2}} + \frac{n_-^{1/2}}{B_2^{1/2}})\frac{\eta}{\tau_2^{1/2}} + \left.C_2\frac{n_+^{1/2}\eta}{B_1\tau_2^{1/2}}\right]^2$$

Thus

$$\mathbb{E}[\|u_{i,t+1} - \frac{1}{n_-}\sum_{j\in S_-} g_i(v_{j,t+1} - v_{i,t+1}, s_{i,t+1})\|]$$

$$\leq (1 - \frac{B_1\tau_2}{2n_+})^{t+1}\|u_{i,0} - \frac{1}{n_-}\sum_{j\in S_-} g_i(v_{i,0} - v_{j,0}, s_{i,0})\| + \frac{2\tau_2^{1/2}\sigma}{B_2^{1/2}} + C_2\frac{n_+}{B_1}(\frac{B_1^{1/2}}{n_+^{1/2}} + \frac{B_2^{1/2}}{n_-^{1/2}})\frac{\tau_1}{\tau_2^{1/2}}$$

$$+ C_2\frac{n_+}{B_1}(\frac{n_+^{1/2}}{B_1^{1/2}} + \frac{n_-^{1/2}}{B_2^{1/2}})\frac{\eta}{\tau_2^{1/2}} + C_2\frac{n_+^{1/2}\eta}{B_1\tau_2^{1/2}}$$

Taking average over $i \in S_+$, we obtain the norm error bound

$$\mathbb{E}\left[\frac{1}{n_+}\sum_{i\in S_+}\|u_{i,t+1} - \frac{1}{n_-}\sum_{j\in S_-}g_i(v_{j,t+1} - v_{i,t+1}, s_{i,t+1})\|\right]$$

$$\leq (1 - \frac{B_1\tau_2}{2n_+})^{t+1}\frac{1}{n_+}\sum_{i\in S_+}\|u_{i,0} - \frac{1}{n_-}\sum_{j\in S_-}g_i(v_{i,0} - v_{j,0}, s_{i,0})\| + \frac{2\tau_2^{1/2}\sigma}{B_2^{1/2}}$$

$$+ C_2\frac{n_+}{B_1}(\frac{B_1^{1/2}}{n_+^{1/2}} + \frac{B_2^{1/2}}{n_-^{1/2}})\frac{\tau_1}{\tau_2^{1/2}} + C_2\frac{n_+}{B_1}(\frac{n_+^{1/2}}{B_1^{1/2}} + \frac{n_-^{1/2}}{B_2^{1/2}})\frac{\eta}{\tau_2^{1/2}} + C_2\frac{n_+^{1/2}\eta}{B_1\tau_2^{1/2}}$$

$\square$

## D.5 Proof of Lemma A.3

*Proof of Lemma A.3.* With $\gamma_1 = \frac{n_1 n_2 - B_1 B_2}{B_1 B_2 (1-\tau_1)} + (1-\tau_1)$ and $\tau_1 \leq \frac{1}{2}$, MSVR update has the following recursive error bound [15][15]

$$\mathbb{E}[\|v_{i,j,t+1} - h_{i,j}(\mathbf{w}_{t+1})\|^2]$$

$$\leq (1 - \frac{B_1 B_2 \tau_1}{n_1 n_2})\mathbb{E}[\|v_{i,j,t} - h_{i,j}(\mathbf{w}_t)\|^2] + \frac{2\tau_1^2 B_1 B_2 \sigma^2}{n_1 n_2 B_3} + \frac{8 n_1 n_2 C_h^2}{B_1 B_2}\mathbb{E}[\|\mathbf{w}_t - \mathbf{w}_{t+1}\|^2]$$

$$\leq (1 - \frac{B_1 B_2 \tau_1}{2 n_1 n_2})^2 \mathbb{E}[\|v_{i,j,t} - h_{i,j}(\mathbf{w}_t)\|^2] + \frac{2\tau_1^2 B_1 B_2 \sigma^2}{n_1 n_2 B_3} + \frac{8 n_1 n_2 C_h^2 M^2 \eta^2}{B_1 B_2}$$

Applying this inequality recursively, we obtain

$$\mathbb{E}[\|v_{i,j,t+1} - h_{i,j}(\mathbf{w}_{t+1})\|^2]$$

$$\leq (1 - \frac{B_1 B_2 \tau_1}{2 n_1 n_2})^{2(t+1)} \|v_{i,j,0} - h_{i,j}(\mathbf{w}_0)\|^2 + \sum_{j=0}^{t}(1 - \frac{B_1 B_2 \tau_1}{2 n_1 n_2})^{2(t-j)}(\frac{2\tau_1^2 B_1 B_2 \sigma^2}{n_1 n_2 B_3} + \frac{8 n_1 n_2 C_h^2 M^2 \eta^2}{B_1 B_2})$$

$$\leq (1 - \frac{B_1 B_2 \tau_1}{2 n_1 n_2})^{2(t+1)} \|v_{i,j,0} - h_{i,j}(\mathbf{w}_0)\|^2 + \frac{4\tau_1 \sigma^2}{B_3} + \frac{16 n_1^2 n_2^2 C_h^2 M^2 \eta^2}{B_1^2 B_2^2 \tau_1}$$

where we use $\sum_{j=0}^{t}(1 - \frac{B_1 B_2 \tau_1}{2 n_1 n_2})^{2(t-j)} \leq \frac{2 n_1 n_2}{B_1 B_2 \tau_1}$. Taking average over $(i,j) \in S_1 \times S_2$ gives the squared-norm error bound.

To derive the norm error bound, we derive

$$\mathbb{E}[\|v_{i,j,t+1} - h_{i,j}(\mathbf{w}_{t+1})\|]^2$$

$$\leq \mathbb{E}[\|v_{i,j,t+1} - h_{i,j}(\mathbf{w}_{t+1})\|^2]$$

$$\leq (1 - \frac{B_1 B_2 \tau_1}{2 n_1 n_2})^{2(t+1)} \|v_{i,j,0} - h_{i,j}(\mathbf{w}_0)\|^2 + \frac{4\tau_1 \sigma^2}{B_3} + \frac{16 n_1^2 n_2^2 C_h^2 M^2 \eta^2}{B_1^2 B_2^2 \tau_1}$$

$$\leq \left[(1 - \frac{B_1 B_2 \tau_1}{2 n_1 n_2})^{t+1} \|v_{i,j,0} - h_{i,j}(\mathbf{w}_0)\| + \frac{2\tau_1^{1/2} \sigma}{B_3^{1/2}} + \frac{4 n_1 n_2 C_h M \eta}{B_1 B_2 \tau_1^{1/2}}\right]^2$$

Thus

$$\mathbb{E}[\|v_{i,j,t+1} - h_{i,j}(\mathbf{w}_{t+1})\|]$$

$$\leq (1 - \frac{B_1 B_2 \tau_1}{2 n_1 n_2})^{t+1} \|v_{i,j,0} - h_{i,j}(\mathbf{w}_0)\| + \frac{2\tau_1^{1/2} \sigma}{B_3^{1/2}} + \frac{4 n_1 n_2 C_h M \eta}{B_1 B_2 \tau_1^{1/2}}$$

Taking average over $(i,j) \in \mathcal{S}_1 \times \mathcal{S}_2$, we obtain the norm error bound

$$\mathbb{E}\left[\frac{1}{n_1}\sum_{i \in \mathcal{S}_1}\frac{1}{n_2}\sum_{j \in \mathcal{S}_2}\|v_{i,j,t+1} - h_{i,j}(\mathbf{w}_{t+1})\|\right]$$

$$\leq (1 - \frac{B_1 B_2 \tau_1}{2 n_1 n_2})^{t+1}\frac{1}{n_1}\sum_{i \in \mathcal{S}_1}\frac{1}{n_2}\sum_{j \in \mathcal{S}_2}\|v_{i,j,0} - h_{i,j}(\mathbf{w}_0)\| + \frac{2\tau_1^{1/2} \sigma}{B_3^{1/2}} + \frac{4 n_1 n_2 C_h M \eta}{B_1 B_2 \tau_1^{1/2}}.$$

$\square$

## D.6 Proof of Lemma A.4

*Proof of Lemma A.4.* With $\gamma_2 = \frac{n_1 - B_1}{B_1 (1-\tau_2)} + (1-\tau_2)$ and $\tau_2 \leq \frac{1}{2}$, MSVR update has the following recursive error bound [15]

$$\mathbb{E}[\|u_{i,t+1} - \frac{1}{n_2}\sum_{j \in \mathcal{S}_2} g_i(v_{i,j,t+1})\|^2]$$

$$\leq (1 - \frac{B_1 \tau_2}{n_1})\mathbb{E}[\|u_{i,t} - \frac{1}{n_2}\sum_{j \in \mathcal{S}_2} g_i(v_{i,j,t})\|^2] + \frac{2\tau_2^2 B_1 \sigma^2}{n_1 B_2} + \frac{8 n_1 C_g^2}{B_1}\mathbb{E}[\|v_{i,j,t+1} - v_{i,j,t}\|^2]$$

(45)

It remains to bound $\mathbb{E}[\|v_{i,j,t+1} - v_{i,j,t}\|^2]$, which is done as following

$$\mathbb{E}[\|v_{i,j,t+1} - v_{i,j,t}\|^2]$$

$$\leq \mathbb{E}\left[\frac{B_1 B_2}{n_1 n_2}\|\tau_1 v_{i,j,t} - \tau_1 h_{i,j}(\mathbf{w}_t; \mathcal{B}_{3,i,j}^t) - \gamma_1(h_{i,j}(\mathbf{w}_t; \mathcal{B}_{3,i,j}^t) - h_{i,j}(\mathbf{w}_{t-1}; \mathcal{B}_{3,i,j}^t))\|^2\right]$$

$$\leq \mathbb{E}\left[\frac{2B_1 B_2 \tau_1^2}{n_1 n_2}\|v_{i,j,t} - h_{i,j}(\mathbf{w}_t; \mathcal{B}_{3,i,j}^t)\|^2 + \frac{2B_1 B_2 \gamma_1^2}{n_1 n_2}\|h_{i,j}(\mathbf{w}_t; \mathcal{B}_{3,i,j}^t) - h_{i,j}(\mathbf{w}_{t-1}; \mathcal{B}_{3,i,j}^t)\|^2\right]$$

$$\leq \mathbb{E}\left[\frac{2B_1 B_2 \tau_1^2}{n_1 n_2}\|v_{i,j,t} - h_{i,j}(\mathbf{w}_t; \mathcal{B}_{3,i,j}^t)\|^2 + \frac{2B_1 B_2 \gamma_1^2 C_h}{n_1 n_2}\|\mathbf{w}_t - \mathbf{w}_{t-1}\|^2\right]$$

$$\overset{(a)}{\leq} \frac{8B_1 B_2 \tau_1^2 M^2}{n_1 n_2} + \frac{8n_1 n_2 C_h^2 \eta^2 M^2}{B_1 B_2}$$

where inequality (a) uses $\tau_1 \leq 1/2$ and $\gamma_1 = \frac{n_1 n_2 - B_1 B_2}{B_1 B_2 (1-\tau_1)} + (1-\tau_1) \leq \frac{2n_1 n_2}{B_1 B_2}$. Plugging the above inequality back into inequality 45 gives

$$\mathbb{E}[\|u_{i,t+1} - \frac{1}{n_2}\sum_{j \in \mathcal{S}_2} g_i(v_{i,j,t+1})\|^2]$$

$$\leq (1 - \frac{B_1 \tau_2}{n_1})\mathbb{E}[\|u_{i,t} - \frac{1}{n_2}\sum_{j \in \mathcal{S}_2} g_i(v_{i,j,t})\|^2] + \frac{2\tau_2^2 B_1 \sigma^2}{n_1 B_2} + \frac{8n_1 C_g^2}{B_1}\left(\frac{8B_1 B_2 \tau_1^2 M^2}{n_1 n_2} + \frac{8n_1 n_2 C_h^2 \eta^2 M^2}{B_1 B_2}\right)$$

$$\leq (1 - \frac{B_1 \tau_2}{n_1})\mathbb{E}[\|u_{i,t} - \frac{1}{n_2}\sum_{j \in \mathcal{S}_2} g_i(v_{i,j,t})\|^2] + \frac{2\tau_2^2 B_1 \sigma^2}{n_1 B_2} + \frac{64B_2 \tau_1^2 M^2 C_g^2}{n_2} + \frac{64n_1^2 n_2 C_h^2 \eta^2 M^2 C_g^2}{B_1^2 B_2}$$

Applying this inequality recursively, we obtain

$$\mathbb{E}[\|u_{i,t+1} - \frac{1}{n_2}\sum_{j \in \mathcal{S}_2} g_i(v_{i,j,t+1})\|^2]$$

$$\leq (1 - \frac{B_1 \tau_2}{2n_1})^{2(t+1)}\|u_{i,0} - \frac{1}{n_2}\sum_{j \in \mathcal{S}_2} g_i(v_{i,j,0})\|^2 + \sum_{j=0}^{t}(1 - \frac{B_1 \tau_2}{2n_1})^{t-j}\left(\frac{2\tau_2^2 B_1 \sigma^2}{n_1 B_2} + \frac{64B_2 \tau_1^2 M^2 C_g^2}{n_2}\right.$$

$$\left. + \frac{64n_1^2 n_2 C_h^2 \eta^2 M^2 C_g^2}{B_1^2 B_2}\right)$$

$$\leq (1 - \frac{B_1 \tau_2}{2n_1})^{2(t+1)}\|u_{i,0} - \frac{1}{n_2}\sum_{j \in \mathcal{S}_2} g_i(v_{i,j,0})\|^2 + \frac{4\tau_2 \sigma^2}{B_2} + \frac{128n_1 B_2 \tau_1^2 M^2 C_g^2}{B_1 n_2 \tau_2} + \frac{128n_1^3 n_2 C_h^2 \eta^2 M^2 C_g^2}{B_1^3 B_2 \tau_2}$$

where we use $\sum_{j=0}^{t}(1 - \frac{B_1 \tau_2}{2n_1})^{2(t-j)} \leq \frac{2n_1}{B_1 \tau_1}$. Taking average over $i \in \mathcal{S}_1$ gives the squared-norm error bound.

To derive the norm error bound, we derive

$$\mathbb{E}[\|u_{i,t+1} - \frac{1}{n_2}\sum_{j \in \mathcal{S}_2} g_i(v_{i,j,t+1})\|]^2$$

$$\leq \mathbb{E}[\|u_{i,t+1} - \frac{1}{n_2}\sum_{j \in \mathcal{S}_2} g_i(v_{i,j,t+1})\|^2]$$

$$\leq (1 - \frac{B_1 \tau_2}{2n_1})^{2(t+1)}\|u_{i,0} - \frac{1}{n_2}\sum_{j \in \mathcal{S}_2} g_i(v_{i,j,0})\|^2 + \frac{4\tau_2 \sigma^2}{B_2} + \frac{128n_1 B_2 \tau_1^2 M^2 C_g^2}{B_1 n_2 \tau_2} + \frac{128n_1^3 n_2 C_h^2 \eta^2 M^2 C_g^2}{B_1^3 B_2 \tau_2}$$

$$\leq \left[(1 - \frac{B_1 \tau_2}{2n_1})^{t+1}\|u_{i,0} - \frac{1}{n_2}\sum_{j \in \mathcal{S}_2} g_i(v_{i,j,0})\| + \frac{2\tau_2^{1/2}\sigma}{B_2^{1/2}} + \frac{8\sqrt{2}n_1^{1/2} B_2^{1/2}\tau_1 M C_g}{B_1^{1/2} n_2^{1/2}\tau_2^{1/2}} + \frac{8\sqrt{2}n_1^{3/2} n_2^{1/2} C_h \eta M C_g}{B_1^{3/2} B_2^{1/2}\tau_2^{1/2}}\right]^2$$

Taking squared root on both sides and taking average over $i \in S_1$, we obtain the norm error bound

$$
\mathbb{E}\left[\frac{1}{n_1}\sum_{i\in\mathcal{S}_1}\|u_{i,t+1} - \frac{1}{n_2}\sum_{j\in\mathcal{S}_2} g_i(v_{i,j,t+1})\|\right]
$$

$$
\leq (1 - \frac{B_1\tau_2}{2n_1})^{t+1}\frac{1}{n_1}\sum_{i\in\mathcal{S}_1}\|u_{i,0} - \frac{1}{n_2}\sum_{j\in\mathcal{S}_2} g_i(v_{i,j,0})\| + \frac{2\tau_2^{1/2}\sigma}{B_2^{1/2}} + \frac{C_2 n_1^{1/2} B_2^{1/2}\tau_1}{B_1^{1/2} n_2^{1/2}\tau_2^{1/2}} + \frac{C_2 n_1^{3/2} n_2^{1/2}\eta}{B_1^{3/2} B_2^{1/2}\tau_2^{1/2}}
$$

where $C_2 = \max\{8\sqrt{2}MC_g, 8\sqrt{2}C_h MC_g\}$. $\qquad\square$

## D.7 Proof of Lemma B.2

*Proof of Lemma B.2.* Define
$$
\tilde{u}_{i,t} = (1 - \tau)u_{i,t} + \tau g_i(\mathbf{w}_t; \mathcal{B}_{2,i}^t)
$$
Then we have
$$
\mathbb{E}_{\mathcal{B}_{2,i}^t}[\|\tilde{u}_{i,t} - g_i(\mathbf{w}_t)\|^2]
$$
$$
= \mathbb{E}_{\mathcal{B}_{2,i}^t}[\|(1 - \tau)(u_{i,t} - g_i(\mathbf{w}_t)) + \tau(g_i(\mathbf{w}_t; \mathcal{B}_{2,i}^t) - g_i(\mathbf{w}_t))\|^2]
$$
$$
= \mathbb{E}_{\mathcal{B}_{2,i}^t}[(1 - \tau)^2\|u_{i,t} - g_i(\mathbf{w}_t)\|^2 + \tau^2\|g_i(\mathbf{w}_t; \mathcal{B}_{2,i}^t) - g_i(\mathbf{w}_t)\|^2
$$
$$
\quad + 2(1 - \tau)\tau\langle u_{i,t} - g_i(\mathbf{w}_t), g_i(\mathbf{w}_t; \mathcal{B}_{2,i}^t) - g_i(\mathbf{w}_t)\rangle]
$$
$$
\leq (1 - \tau)^2\|u_{i,t} - g_i(\mathbf{w}_t)\|^2 + \frac{\tau^2\sigma^2}{B_2}
$$

It follows
$$
\mathbb{E}_{\mathcal{B}_{2,i}^t}\mathbb{E}_{\mathcal{B}_1^t}[\|u_{i,t+1} - g_i(\mathbf{w}_t)\|^2]
$$
$$
= \frac{B_1}{n_1}\mathbb{E}_{\mathcal{B}_{2,i}^t}[\|\tilde{u}_{i,t} - g_i(\mathbf{w}_t)\|^2] + (1 - \frac{B_1}{n_1})\|u_{i,t} - g_i(\mathbf{w}_t)\|^2
$$
$$
\leq \frac{B_1}{n_1}(1 - \tau)^2\|u_{i,t} - g_i(\mathbf{w}_t)\|^2 + \frac{B_1\tau^2\sigma^2}{n_1 B_2} + (1 - \frac{B_1}{n_1})\|u_{i,t} - g_i(\mathbf{w}_t)\|^2
$$
$$
\leq (1 - \frac{B_1\tau}{2n_1})^2\|u_{i,t} - g_i(\mathbf{w}_t)\|^2 + \frac{B_1\tau^2\sigma^2}{n_1 B_2}
$$

where we use
$$
\frac{B_1}{n_1}(1 - \tau)^2 + (1 - \frac{B_1}{n_1}) = \frac{B_1}{n_1}(1 - 2\tau + \tau^2) + 1 - \frac{B_1}{n_1}
$$
$$
= 1 - 2\tau\frac{B_1}{n_1} + \tau^2\frac{B_1}{n_1}
$$
$$
\leq 1 - \tau\frac{B_1}{n_1}
$$
$$
\leq 1 - \tau\frac{B_1}{n_1} + (\frac{\tau B_1}{2n_1})^2 = (1 - \frac{\tau B_1}{2n_1})^2
$$

Then
$$
\mathbb{E}_t[\|u_{i,t+1} - g_i(\mathbf{w}_{t+1})\|^2]
$$
$$
\leq \mathbb{E}_t\left[(1 + \frac{B_1\tau}{4n_1})\|u_{i,t+1} - g_i(\mathbf{w}_t)\|^2 + (1 + \frac{4n_1}{B_1\tau})\|g_i(\mathbf{w}_t) - g_i(\mathbf{w}_{t+1})\|^2\right]
$$
$$
\leq (1 + \frac{B_1\tau}{4n_1})(1 - \frac{B_1\tau}{2n_1})^2\|u_{i,t} - g_i(\mathbf{w}_t)\|^2 + (1 + \frac{B_1\tau}{4n_1})\frac{B_1\tau^2\sigma^2}{n_1 B_2}
$$
$$
\quad + (1 + \frac{4n_1}{B_1\tau})C_g^2\mathbb{E}_t\|\mathbf{w}_t - \mathbf{w}_{t+1}\|^2]
$$
$$
\leq (1 - \frac{B_1\tau}{4n_1})^2\|u_{i,t} - g_i(\mathbf{w}_t)\|^2 + \frac{2B_1\tau^2\sigma^2}{n_1 B_2} + \frac{8n_1}{B_1\tau}C_g^2\mathbb{E}_t\|\mathbf{w}_t - \mathbf{w}_{t+1}\|^2]
$$
$$
\leq (1 - \frac{B_1\tau}{4n_1})^2\|u_{i,t} - g_i(\mathbf{w}_t)\|^2 + \frac{2B_1\tau^2\sigma^2}{n_1 B_2} + \frac{8n_1 C_g^2 M^2\eta^2}{B_1\tau}
$$

where we use $\frac{B_1\tau}{4n_1} \leq 1$. Applying this inequality recursively, we obtain

$$\mathbb{E}[\|u_{i,t+1} - g_i(\mathbf{w}_{t+1})\|^2]$$

$$\leq (1 - \frac{B_1\tau}{4n_1})^2 \mathbb{E}[\|u_{i,t} - g_i(\mathbf{w}_t)\|^2] + \frac{2B_1\tau^2\sigma^2}{n_1 B_2} + \frac{8n_1 C_g^2 M^2 \eta^2}{B_1\tau}$$

$$\leq (1 - \frac{B_1\tau}{4n_1})^{2(t+1)} \|u_{i,0} - g_i(\mathbf{w}_0)\|^2 + \sum_{j=0}^{t} (1 - \frac{B_1\tau}{4n_1})^{2(t-j)} \left[ \frac{2B_1\tau^2\sigma^2}{n_1 B_2} + \frac{8n_1 C_g^2 M^2 \eta^2}{B_1\tau} \right]$$

$$\leq (1 - \frac{B_1\tau}{4n_1})^{2(t+1)} \|u_{i,0} - g_i(\mathbf{w}_0)\|^2 + \frac{8\tau\sigma^2}{B_2} + \frac{32n_1^2 C_g^2 M^2 \eta^2}{B_1^2\tau^2}$$

where we use $\sum_{j=0}^{t}(1 - \frac{B_1\tau}{4n_1})^{2(t-j)} \leq \frac{4n_1}{B_1\tau}$.

To obtain the absolute bound, we derive

$$\mathbb{E}[\|u_{i,t+1} - g_i(\mathbf{w}_{t+1})\|]^2 \leq \mathbb{E}[\|u_{i,t+1} - g_i(\mathbf{w}_{t+1})\|^2]$$

$$\leq (1 - \frac{B_1\tau}{4n_1})^{2(t+1)} \|u_{i,0} - g_i(\mathbf{w}_0)\|^2 + \frac{8\tau\sigma^2}{B_2} + \frac{32n_1^2 C_g^2 M^2 \eta^2}{B_1^2\tau^2}$$

$$\leq \left[ (1 - \frac{B_1\tau}{4n_1})^{t+1} \|u_{i,0} - g_i(\mathbf{w}_0)\| + \frac{2\sqrt{2}\tau^{1/2}\sigma}{B_2^{1/2}} + \frac{4\sqrt{2}n_1 C_g M\eta}{B_1\tau} \right]^2$$

The desired result follows by taking squared root on both sides. $\qquad \square$

## D.8   Proof of Lemma B.5

*Proof of Lemma B.5.* The proof of Lemma B.5 is the same as Lemma B.2. $\qquad \square$

## D.9   Proof of Lemma B.6

*Proof of Lemma B.6.* Define

$$\tilde{u}_{i,t} = (1 - \tau_2)u_{i,t} + \tau_2 \frac{1}{B_2} \sum_{j \in \mathcal{B}_{2,i}^t} g_i(v_{i,j,t})$$

Then we have

$$\mathbb{E}_{\mathcal{B}_2^t}[\|\tilde{u}_{i,t} - \frac{1}{n_2} \sum_{j \in \mathcal{S}_2} g_i(v_{i,j,t})\|^2]$$

$$= \mathbb{E}_{\mathcal{B}_2^t}[\|(1-\tau_2)(u_{i,t} - \frac{1}{n_2} \sum_{j \in \mathcal{S}_2} g_i(v_{i,j,t})) + \tau_2(\frac{1}{B_2} \sum_{j \in \mathcal{B}_2^t} g_i(v_{i,j,t}) - \frac{1}{n_2} \sum_{j \in \mathcal{S}_2} g_i(v_{i,j,t}))\|^2]$$

$$= \mathbb{E}_{\mathcal{B}_2^t}[(1-\tau_2)^2 \|u_{i,t} - \frac{1}{n_2} \sum_{j \in \mathcal{S}_2} g_i(v_{i,j,t})\|^2 + \tau_2^2 \|\frac{1}{B_2} \sum_{j \in \mathcal{B}_2^t} g_i(v_{i,j,t}) - \frac{1}{n_2} \sum_{j \in \mathcal{S}_2} g_i(v_{i,j,t})\|^2$$

$$+ 2(1-\tau_2)\tau_2 \langle u_{i,t} - \frac{1}{n_2} \sum_{j \in \mathcal{S}_2} g_i(v_{i,j,t}), \frac{1}{B_2} \sum_{j \in \mathcal{B}_2^t} g_i(v_{i,j,t}) - \frac{1}{n_2} \sum_{j \in \mathcal{S}_2} g_i(v_{i,j,t}) \rangle]$$

$$\leq (1-\tau_2)^2 \|u_{i,t} - \frac{1}{n_2} \sum_{j \in \mathcal{S}_2} g_i(v_{i,j,t})\|^2 + \frac{\tau_2^2\sigma^2}{B_2}$$

It follows

$$\mathbb{E}_{\mathcal{B}_{2,i}^t}\mathbb{E}_{\mathcal{B}_1^t}[\|u_{i,t+1} - \frac{1}{n_2}\sum_{j\in\mathcal{S}_2}g_i(v_{i,j,t})\|^2]$$

$$= \frac{B_1}{n_1}\mathbb{E}_{\mathcal{B}_2^t}[\|\tilde{u}_{i,t} - \frac{1}{n_2}\sum_{j\in\mathcal{S}_2}g_i(v_{i,j,t})\|^2] + (1-\frac{B_1}{n_1})\|u_{i,t} - \frac{1}{n_2}\sum_{j\in\mathcal{S}_2}g_i(v_{i,j,t})\|^2$$

$$\leq \frac{B_1}{n_1}(1-\tau_2)^2\|u_{i,t} - \frac{1}{n_2}\sum_{j\in\mathcal{S}_2}g_i(v_{i,j,t})\|^2 + \frac{B_1\tau_2^2\sigma^2}{n_1 B_2} + (1-\frac{B_1}{n_1})\|u_{i,t} - \frac{1}{n_2}\sum_{j\in\mathcal{S}_2}g_i(v_{i,j,t})\|^2$$

$$\leq (1-\frac{B_1\tau_2}{2n_1})^2\|u_{i,t} - \frac{1}{n_2}\sum_{j\in\mathcal{S}_2}g_i(v_{i,j,t})\|^2 + \frac{B_1\tau^2\sigma^2}{n_1 B_2}$$

where we use

$$\frac{B_1}{n_1}(1-\tau_2)^2 + (1-\frac{B_1}{n_1}) \leq (1-\frac{\tau_2 B_1}{2n_1})^2$$

Then

$$\mathbb{E}_t[\|u_{i,t+1} - \frac{1}{n_2}\sum_{j\in\mathcal{S}_2}g_i(v_{i,j,t+1})\|^2]$$

$$\leq \mathbb{E}_t\left[(1+\frac{B_1\tau_2}{4n_1})\|u_{i,t+1} - \frac{1}{n_2}\sum_{j\in\mathcal{S}_2}g_i(v_{i,j,t})\|^2 + (1+\frac{4n_1}{B_1\tau_2})\|\frac{1}{n_2}\sum_{j\in\mathcal{S}_2}g_i(v_{i,j,t}) - \frac{1}{n_2}\sum_{j\in\mathcal{S}_2}g_i(v_{i,j,t+1})\|^2\right]$$

$$\leq (1+\frac{B_1\tau_2}{4n_1})(1-\frac{B_1\tau_2}{2n_1})^2\|u_{i,t} - \frac{1}{n_2}\sum_{j\in\mathcal{S}_2}g_i(v_{i,j,t})\|^2 + (1+\frac{B_1\tau_2}{4n_1})\frac{B_1\tau_2^2\sigma^2}{n_1 B_2}$$

$$+ (1+\frac{4n_1}{B_1\tau_2})C_g^2\mathbb{E}_t[\frac{1}{n_2}\sum_{j\in\mathcal{S}_2}\|v_{i,j,t} - v_{i,j,t+1}\|^2]$$

$$\leq (1-\frac{B_1\tau_2}{4n_1})^2\|u_{i,t} - \frac{1}{n_2}\sum_{j\in\mathcal{S}_2}g_i(v_{i,j,t})\|^2 + \frac{2B_1\tau_2^2\sigma^2}{n_1 B_2} + \frac{8C_g^2 M^2 B_2\tau_1^2}{n_2\tau_2}$$

where we use $\frac{B_1\tau_2}{4n_1} \leq 1$, and

$$\mathbb{E}_t[\|v_{i,j,t} - v_{i,j,t+1}\|^2] = \frac{B_1 B_2}{n_1 n_2}\mathbb{E}_{\mathcal{B}_{3,i,j}^t}\|\tau_1 v_{i,j,t} - \tau_1 h_{i,j}(\mathbf{w}_t;\mathcal{B}_{3,i,j}^t)\|^2 \leq \frac{B_1 B_2\tau_1^2 M^2}{n_1 n_2}.$$

Applying this inequality recursively, we obtain

$$\mathbb{E}[\|u_{i,t+1} - \frac{1}{n_2}\sum_{j\in\mathcal{S}_2}g_i(v_{i,j,t+1})\|^2]$$

$$\leq (1-\frac{B_1\tau_2}{4n_1})^2\mathbb{E}[\|u_{i,t} - \frac{1}{n_2}\sum_{j\in\mathcal{S}_2}g_i(v_{i,j,t})\|^2] + \frac{2B_1\tau_2^2\sigma^2}{n_1 B_2} + \frac{8C_g^2 M^2 B_2\tau_1^2}{n_2\tau_2}$$

$$\leq (1-\frac{B_1\tau_2}{4n_1})^{2(t+1)}\|u_{i,0} - \frac{1}{n_2}\sum_{j\in\mathcal{S}_2}g_i(v_{i,j,0})\|^2 + \sum_{j=0}^t(1-\frac{B_1\tau_2}{4n_1})^{2(t-j)}\left[\frac{2B_1\tau_2^2\sigma^2}{n_1 B_2} + \frac{8C_g^2 M^2 B_2\tau_1^2}{n_2\tau_2}\right]$$

$$\leq (1-\frac{B_1\tau_2}{4n_1})^{2(t+1)}\|u_{i,0} - \frac{1}{n_2}\sum_{j\in\mathcal{S}_2}g_i(v_{i,j,0})\|^2 + \frac{8\tau_2\sigma^2}{B_2} + \frac{32C_g^2 M^2 n_1 B_2\tau_1^2}{B_1 n_2\tau_2^2}$$

where we use $\sum_{j=0}^t(1-\frac{B_1\tau_2}{4n_1})^{2(t-j)} \leq \frac{4n_1}{B_1\tau_2}$.

To obtain the absolute bound, we derive

$$\mathbb{E}[\|u_{i,t+1} - \frac{1}{n_2}\sum_{j \in \mathcal{S}_2} g_i(v_{i,j,t+1})\|]^2$$

$$\leq \mathbb{E}[\|u_{i,t+1} - \frac{1}{n_2}\sum_{j \in \mathcal{S}_2} g_i(v_{i,j,t+1})\|^2]$$

$$\leq (1 - \frac{B_1\tau_2}{4n_1})^{2(t+1)}\|u_{i,0} - \frac{1}{n_2}\sum_{j \in \mathcal{S}_2} g_i(v_{i,j,0})\|^2 + \frac{8\tau_2\sigma^2}{B_2} + \frac{32C_g^2 M^2 n_1 B_2 \tau_1^2}{B_1 n_2 \tau_2^2}$$

$$\leq \left[(1 - \frac{B_1\tau_2}{4n_1})^{t+1}\|u_{i,0} - \frac{1}{n_2}\sum_{j \in \mathcal{S}_2} g_i(v_{i,j,0})\| + \frac{2\sqrt{2}\tau_2^{1/2}\sigma}{B_2^{1/2}} + \frac{4\sqrt{2}C_g M n_1^{1/2} B_2^{1/2} \tau_1}{B_1^{1/2} n_2^{1/2} \tau_2}\right]^2$$

The desired result follows by taking squared root on both sides. $\square$

# E    Group Distributionally Robust Optimization

NSWC FCCO finds an important application in group distributionally robust optimization (group DRO), particularly valuable in addressing distributional shift [25]. Consider $N$ groups with different distributions. Each group $k$ has an averaged loss $L_k(w) = \frac{1}{n_k}\sum_{i=1}^{n_k} \ell(f_w(x_i^k), y_i^k)$, where $w$ is the the model parameter and $(x_i^k, y_i^k)$ is a data point. For robust optimization, we assign different weights to different groups and form the following robust loss minimization problem:

$$\min_w \max_{p \in \Omega} \sum_{k=1}^N p_k L_k(w),$$

where $\Omega \subset \Delta$ and $\Delta$ denotes a simplex. A common choice for $\Omega$ is $\Omega = \{\mathbf{p} \in \Delta, p_i \leq 1/K\}$ where $K$ is an integer, resulting in the so-called CVaR losses, i.e., average of top-K group losses. Consequently, the above problem can be equivalently reformulated as [23]:

$$\min_w \min_s F(w,s) = \frac{1}{K}\sum_{k=1}^N [L_k(w) - s]_+ + s.$$

This formulation can be mapped into non-smooth weakly-convex FCCO when the loss function $\ell(\cdot, \cdot)$ is weakly convex in terms of $w$. In comparison to directly solving the min-max problem, solving the above FCCO problem avoids the need of dealing with the projection onto the constraint $\Omega$ and expensive sampling as in existing works [4].

# F    More Information for Experiments

## F.1    Dataset Statistics

Table 3: Datasets Statistics. The percentage in parenthesis represents the proportion of positive samples.

| Dataset | Train | Validation | Test |
|---|---|---|---|
| moltox21(t0) | 5834 (4.25%) | 722 (4.01%) | 709 (4.51%) |
| molmuv(t1) | 11466 (0.18%) | 1559 (0.13%) | 1709 (0.35%) |
| molpcba(t0) | 120762 (9.32%) | 19865 (11.74%) | 20397 (11.61%) |

Table 4: Data statistics for the MIL datasets. $D_+/D_-$ is the positive/negative bag number.

| Data Format | Dataset | $D_+$ | $D_-$ | average bag size | #features |
|---|---|---|---|---|---|
| Tabular | MUSK2 | 39 | 63 | 64.69 | 166 |
| | Fox | 100 | 100 | 6.6 | 230 |
| Histopathological | Lung | 100 | 1000 | 256 | 32x32x3 |
| Image | Lung | 100 | 1000 | 256 | 32x32x3 |

## F.2 Illustration for Histopathology Dataset on MIL Task

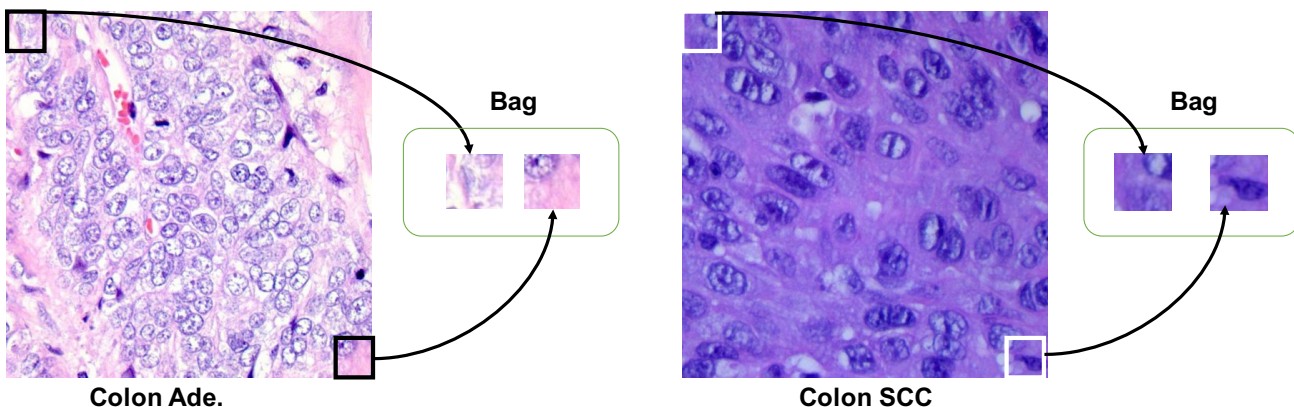

Figure 2: Illustration for Histopathology Dataset on MIL Task. Ade. is abbreviated for adenocarcinoma and SCC is short for squamous cell carcinoma. In this work, each RGB image is separated by 32×32 non-overlapped patches, which constitute the bag.

