# OpenReview forum: "Non-Smooth Weakly-Convex Finite-sum Coupled Compositional Optimization"
_NeurIPS.cc/2023/Conference — NeurIPS 2023 poster_

### Official Review · Reviewer_FiRa · 2023-07-05

**Soundness:** 3 good
**Presentation:** 3 good
**Contribution:** 3 good
**Rating:** 6
**Confidence:** 3

**Summary:**

This paper addresses the finite-sum coupled compositional optimization (FCCO) scenario, relaxing the requirement of Lipschitz gradient for the involved functions. Instead, they consider weakly convex functions with certain monotonicity conditions. The paper introduces new algorithms and provides oracle complexity guarantees for computing a point with an epsilon-gradient norm relative to the Moreau envelop. The authors also demonstrate the applicability of these methods to AUC maximization tasks.

**Strengths:**

It is always beneficial and sometimes nontrivial to extend the algorithm implementation and analytic technique for gradient Lipschitz functions to that of more general weakly convex functions. The authors also demonstrate practical applicability of the proposed new algorithms.

**Weaknesses:**

Since I'm not acquainted with the FCCO setting, I will concentrate on the broader technical aspects instead.

* L137: I don't get the idea of the "monotonic property" of a multivariate mapping f here. What is the "input" referring here? Let the function be f: R^n -> R^m. Is the input here referring to a vector in R^n? If so, how can we define the "monotonicity" with respect to such a multivariate function?

* The whole development in this paper is built on the weak convexity of F proved in Proposition 4.2 and 4.4. However, the arguments there are not transparent to me. Let me take the proof of Proposition 4.2 for example.

(a) L809 first inequality: Why do we have v'(g(y) - g(x)) \geq v'( ... )? It seems that the authors assumed v <= 0 here as f is assumed non-increasing in L808 (though the term "non-increasing" is not clearly defined in my opinion). However, it is not clear to me why such a "monotonicity" of f will promise the nonnegativity of subgradients v. For a smooth function f, the claim (v <= 0) holds trivially. But for a general non-smooth weakly convex function, even with the "monotonic assumption", this is nontrivial and a formal proof is required.

(b) L809 third inequality: should L_2C_1 be L_2C_1*sqrt(n), where n is the input dimension of f? Please check. If that is true, then the weak convexity parameter would be depend on the dimension n of the input to f.

Minor:

* L805: should v'(g(x) - g(y)) be v'(g(y) - g(x))?
* L145: \epsilon > 0 should be 1 > \epsilon >= 0.

**Questions:**

See weaknesses.

**Limitations:**

Yes.

---

> ### Author Rebuttal · Authors · 2023-08-09
>
> We thank reviewer FiRa for the detailed and insightful review. We will fix the minor issues in the future revision. Here we would like to address the remaining concerns.
>
> **Q1**. L137: Regarding the "monotonic property" of a multivariate mapping $f$.
>
> **Response**: We apologize for this confusion. Here we present a more detailed definition of monotonicity:
>
> Consider a function $f:\mathbb{R}^d\to \mathbb{R}^m$. For simplicity, we write $f=(f_1,\dots,f_m)$. We say that $f$ is non-decreasing if for each $k=1,\dots,m$, $f_k:\mathbb{R}^d\to \mathbb{R}$ is non-decreasing with respect to each element in the input. We will add this into the revision.
>
> **Q2**. Regarding the proof of Proposition 4.2, L809 first inequality: why such a "monotonicity" of f will promise the nonnegativity of subgradients v?
>
> **Response**: Thank you for raising this question. Here we present an additional proposition to address this issue. Before stating the proposition, we would like to mention the definition of subgradient we used in this work. For the class of weakly-convex Lipschitz continuous functions, the definition of Clarke subgradients [Theorem 9.61 Rockafellar and Wets, 2009] coincides with the definition of subgradient [definition 8.3 in Rockafellar and Wets, 2009] [41]. Since we assume both weak convexity and Lipschitz continuity for functions in our objectives, the definitions mentioned above are considered to be equivalent.
>
> Proposition. Consider a Lipschitz continuous function $f:O\to\mathbb{R}$ where $O\subset \mathbb{R}^d$ is an open set. Assume $f$ to be non-increasing (resp. non-decreasing) with respect to each element in the input, then all subgradients of $f$ are non-positive (resp. non-negative) element-wise.
>
> Proof. Let $D$ be the subset of $O$ where $f$ is differentiable. By Theorem 9.60 in [Rockafellar and Wets, 2009], a Lipschitz continuous function $f:O\to\mathbb{R}$, where $O\subset \mathbb{R}^d$ is an open set, is differentiable almost everywhere, i.e., $D$ is dense in $O$. Then by Theorem 9.61 in [Rockafellar and Wets, 2009], the subdifferential of $f$ at $x$ is defined as
> $$
>     \partial f(x) = \text{con} (v | \exists x_k\to x \text{ with } x_k\in D, \nabla f(x_k)\to v),
> $$
> where $\text{con}(\cdot)$ denotes the convex hull. If we assume that $f$ is non-increasing with respect to each element in the input, then $\nabla f(x)\leq 0$ (element-wise) for all differentiable points $x\in D$. It implies that the all vectors in $(v | \exists x_k\to x \text{ with } x_k\in D, \nabla f(x_k)\to v)$ are non-positive element-wise. Therefore, all subgradients of $f$ are non-positive element-wise. On the other hand, if we assume that $f$ is non-decreasing, one may follow the same argument and conclude that all subgradients of $f$ are non-negative element-wise.
>
> For functions $f:O\to\mathbb{R}^m$ where $O\subset \mathbb{R}^d$ is an open set, one may write $f=(f_1,\dots,f_m)$ and apply the above proposition for each $f_k:O\to\mathbb{R}$, $ k=1,\dots,m$.
>
>
>
>
> **Q3**. In the proof of Proposition 4.2, L809 third inequality: should $L_2C_1$ be $L_2C_1*\sqrt{d}$, where $d$ is the input dimension of f? Please check.
>
> **Response**: Thank you for pointing this out! You are correct.  The weak convexity parameter of $F$ should involve the input dimension to the outer  functions. The weak convexity constant in Proposition 4.2 should be $\rho_F= \sqrt{d_1}\rho_gC_f+\rho_f C_g^2$, where $d_1$ is the input dimension of $f_i$. The weak convexity constant in Proposition 4.4 should be $\rho_F = \sqrt{d_1}(\sqrt{d_2}L_hC_g+\rho_g C_h^2)C_f+\rho_f C_g^2C_h^2$, where $d_1$ and $d_2$ are the input dimensions of $f_i$ and $g_i$ respectively.

---

> ### Author Response · Authors · 2023-08-17
> **Questions?**
>
> Dear Reviewer FiRa and AC,
>
> Given that the discussion period is ending soon, we would like to follow up if there are some other questions from the reviewer that need us to clarify. Thank you for providing valuable comments on our paper!
>
> Regards
> Authors

---

> > ### Comment · Reviewer_FiRa · 2023-08-17
> >
> > Thank the authors for the clarification, which addressed my concerns and filled in the missing details in the statement and proof. I'll increase my score to 6.

---

### Official Review · Reviewer_GoBV · 2023-07-06

**Soundness:** 3 good
**Presentation:** 2 fair
**Contribution:** 3 good
**Rating:** 6
**Confidence:** 3

**Summary:**

The paper considers a class of finite-sum coupled compositional optimization (FCCO) problems and a class of tri-level finite-sum coupled compositional optimization (TCCO) problems. Under the setting of non-smooth weakly-convex FCCO, the paper establishes the complexity of a single-loop algorithm with the tool of Moreau envelop. The algorithm is then extended to solve the non-smooth weekly-convex TCCO. Numerical results on the two-way partial AUC maximization and multi-instance two-way partial AUC maximization are also reported.

**Strengths:**

Convergence analysis of an single-loop stochastic algorithm for solving non-smooth weakly-convex FCCO is a nice contribution of the paper.

**Weaknesses:**

1. There lacks motivation for considering non-smooth weakly-convex FCCO and TCCO. Some additional applications other than the two-way partial AUC maximization should be provided.

2. In the nonconvex setting, definitions need to be rigorously given/referred to. For example, $f$ at line 133 can be nonconvex, which definition of "subgradient" and "subdifferential" are the paper using? There some ways to define a "variance-reduced estimator", which definition is the paper using?

3. The experiments just partially support the theoretical results of the paper. Numerical results on the comparison of the evolution of the objective function over time when using the proposed algorithm and other known algorithms for solving one same non-smooth weakly-convex FCCO problem are expected. It would illustrate the importance of the obtained complexity.

**Questions:**

Consider the TPAUC problem in Section 5.The corresponding NSWC FCCO problem of the regular setting and the corresponding NSWC TCCO problem for the MIL setting still have convex $f_i$. Could the complexity of the proposed algorithms be improved if the weakly-convex assumption of $f_i$ is replaced by convexity?

**Limitations:**

n.n.

---

> ### Author Rebuttal · Authors · 2023-08-09
>
> We thank reviewer GoBV for the insightful review. Here we would like to address your concerns.
>
> **Q1**: Additional applications other than the two-way partial AUC maximization should be provided.
>
> **Response**:  We would like to provide another important application of NSWC FCCO for regularized group distributionally robust optimization (group DRO), which is useful for addressing distributional shift  [Sagawa et al. 2020].  Consider $N$ groups with different distributions. Each group $k$ has an averaged loss $L_k(w)=\frac{1}{n_k}\sum_{t=1}^{n_k}\ell (f_w(x_t^k),y_t^k)$, where $w$ is the the model parameter and $(x_t^k, y_t^k)$ is a data. For robust optimization, we assign different weights to different groups and form the following robust loss minimization problem:
> $$
> \min_w \max_{p\in \Omega} \sum_{k=1}^N p_k L_k(w),
> $$
> where $\Omega\subset\Delta$ and $\Delta$ denotes a simplex. A common choice for $\Omega$ is $\Omega=$\{$p\in\Delta, p_i\leq 1/K$\} where $K$ is an integer, which yields the so-called CVaR losses (i.e., average of top-K group losses). As a result, the above problem is equivalent to (Curi et al, 2019):
> $$
> \min_w \min_{s} F(w,s)=\frac{1}{K}\sum_{k=1}^N [L_k(w)-s]_+ + s.
> $$
> We can map this problem into non-smooth weakly-convex FCCO when the loss function $\ell(\cdot,\cdot)$ is weakly convex in terms of $w$. Compared with solving the min-max problem, solving the above FCCO problem does not involve dealing with the projection onto the constraint $\Omega$ and avoid expensive sampling as in existing works (Curi et al, 2019).
>
> **Q2**: In the nonconvex setting, definitions of "subgradient" and "subdifferential" need to be rigorously given/referred to.
>
> **Response**: Thank you for the comments! We will add these definitions into the revision. In particular, the definitions of subgradient and subdifferential follow from the definition 8.3 in [Rockafellar and Wets, 2009] and section 2.2 of [5]. In section 2.2 of [5], properties of subgradients for weakly-convex functions are discussed. For the class of weakly-convex Lipschitz continuous functions, the definition of Clarke subgradients [Theorem 9.61 Rockafellar and Wets, 2009] coincides with definition 8.3 in [Rockafellar and Wets, 2009] [41].
>
> **Q3**: Which definition of "variance-reduced estimator" is the paper using?
>
> **Response**:  The use of the term “variance-reduced estimator” follows from the line of research on variance reduction techniques in the literature of stochastic optimization for machine learning, including SVRG [Johnson and Zhang, 2013], SARAH [Nguyen et al., 2017], SPIDER [Fang et al., 2018], STORM [4], and MSVR [13]. In our context, “variance-reduced estimator” generally refers to the estimation techniques that ensures the average accumulated estimation error of the gradient estimators decays. We will clarify it in the revision.
>
> **Q4**: Numerical results on the comparison of the evolution of the objective function over time when using the proposed algorithm and other known algorithms for solving one same non-smooth weakly-convex FCCO problem are expected.
>
> **Response**: (1) Please note that this is **the first work** that studies and solves non-smooth weakly-convex FCCO problems. Hence, there is no known algorithms in the same style for solving the same non-smooth weakly-convex FCCO problems that we can compare with. (2) The experimental results serve the purpose of demonstrating the usefulness of our algorithms for solving ML problems. In particular, for TPAUC maximization we compared with the state-of-the-art baseline SOTAs [39] in order to illustrate the generalization performance of our algorithms. (3) We did provide experimental results to justify the impact of different algorithmic choices on the convergence. For example, the results in Figure 3 in global response demonstrate that using the MSVR estimator ($\gamma>0$) gives faster convergence than using the moving average estimator ($\gamma=0$), which is implied by our theoretical results. (4) To better address your concern, we have implemented a min-max optimization approach for optimizing the same TPAUC loss as ours, which was proposed in the supplement of [39] and named as SOTA. Please note that SOTA leverages the convexity of the outer function $f_i$ and solves the equivalent min-max problem. The training convergence results shown in Figure 3 in the global response demonstrate that our method has competitive and sometimes even faster convergence though our method does not explore the convexity of the outer function.
>
> **Q5**:  Could the complexity of the proposed algorithms be improved if the weakly-convex assumption of $f_i$ is replaced by convexity?
>
> **Response**: It is unclear to us whether the convexity of $f_i$ can help improve the convergence.  To the best of our knowledge, when $f_i$ is convex the problem is equivalent to a weakly-convex concave min-max problem, whose best complexity is also $O(\epsilon^{-6})$ [20,26,36,38]. Since there is no known lower bound for this problem, we do not know if it is possible to further improve the complexity.
>
> **References**:
>
> Yang et al. Two-way partial auc and its properties.Statistical methods in medical research,28(1):184–195,2019.
>
> Sagawa et al. Distributionally Robust Neural Networks for Group Shifts: On the Importance of Regularization for Worst-Case Generalization. ICLR, 2020.
>
> Curi et al. Adaptive sampling for stochastic risk-averse learning, 2019.
>
> Rockafellar and Wets. Variational analysis, volume 317. Springer Science & Business Media, 2009.
>
> Johnson and Zhang. Accelerating stochastic gradient descent using predictive variance reduction. In NIPS, 2013.
>
> Fang et al. Spider: Near-optimal non-convex optimization via stochastic path-integrated differential estimator. In Advances in Neural Information Processing Systems, 2018.
>
> Nguyen et al. SARAH: A novel method for machine learning problems using stochastic recursive gradient. In Proc. of the 34th ICML, 2017.

---

### Official Review · Reviewer_evM3 · 2023-07-12

**Soundness:** 3 good
**Presentation:** 3 good
**Contribution:** 3 good
**Rating:** 7
**Confidence:** 3

**Summary:**

This paper handles the problems of non-smooth weakly-convex compositional optimization. The first problem, referred to as FCCO (Finite-sum coupled compositional minimization), is given by

$$ \min_{w \in \mathbb{R}^d} F(w) \triangleq \frac{1}{n} \sum_{i=1}^n f_i(\mathbb{E}_{\xi\sim \mathcal{D}_i}[g_i(w; \xi)])$$

The second problem, referred to as TCCO (Tri-level coupled compositional minimization), is given by,

$$ \min_{w \in \mathbb{R}^d} F(w) \triangleq \frac{1}{n_1} \sum_{i=1}^{n_1} f_i\left(\frac{1}{n_2}\sum_{j=1}^{n_2} g_i( \underset{\xi \sim \mathcal{D}{i,j}}{\mathbb{E}} [h_{i,j}(w; \xi)])\right)$$


Here, the outer functions $f_i$ and $g_i$ are Lipschitz and weakly convex which has not been considered by previous works and significantly complicates the analysis.

The authors propose two algorithms, SONX and SONT for the two problems respectively. For the first problem, FCCO, $u_{i,t}$ are the estimates of $g_i(w_t)$, $\forall i \in \{1,2,\ldots, n\}$, which are required for an unbiased estimate of the the complete function $F(w)$. In each step, a batch $B$ of coordinates from $\{1,2,\ldots, n\}$ is sampled and only the estimates in this batch, $u_{i,t}, i\in B$, are updated using a Variance-reduced estimator. Using the estimates, a single gradient step is computed for $w_t$. The estimator has been adapted from [1].

As the functions are weakly convex, the authors show convergence to an $\epsilon$-stationary point of the Moreau envelope of $F$ at a rate of $\mathcal{O}(\epsilon^{-6})$.

The authors show that both problems can be mapped exactly to Two-way partial AUC (TPAUC) maximization of a parametrized network, even deep networks, with the TCCO problem mapping to a multi-instance learning version. TPAUC is the area under the ROC curve when False Positive rate $\leq \beta$ and True Positive Rate $\geq \alpha$. Over a set of medical datasets, the authors show that SONX and SONT out-perform baselines for TPAUC maximization as the baselines use poor approximations of TPAUC.



**References**
1. Jiang et al 2022. Multi-block-single-probe variance reduced estimator for coupled compositional optimization


**Strengths:**

- **Easy to Implement**: The algorithms are single-loop and use moderately sized batches which makes it easy to implement. Also, the algorithms can also be parallelized over the coordinates $i$.
- **Better Rates**: In addition to being easy to implement, the algorithms achieve the best possible rates for this class of functions. Existing works obtain similar  or worse rates under an easier problem setting or with multiple loops or using large batches.
- **Mapping to TPAUC**: The authors describe the TPAUC problem comprehensively and show the exact mapping with the two optimization problems.
- **Detailed presentation**: The convergence analysis, the mapping to TPAUC and the related works are very detailed which helps in understanding the paper.

**Weaknesses:**

- **Experimental and Theoretical Baselines do not match**: For theory, the baselines for comparisons are algorithms for simpler problem settings, for instance with smoothness or convexity. For experiments, the baselines for TPAUC maximization are used which do not match the theoretical baselines. It is unclear, if some theoretical baseline can be applied to TPAUC and perform better. This should not be the case ideally but it has not been verified.


**Questions:**

- Is there a lower bound for this problem setting, which can be used to verify if the rates are optimal and cannot be improved?
- Why is the dependence on $\frac{n_1}{B_1}$ worse for SONT as compared to SONT?

**Limitations:**

Described in Weaknesses

---

> ### Author Rebuttal · Authors · 2023-08-09
>
> We thank reviewer evM3 for the detailed insightful review. Here we would like to address your concerns.
>
> **Q1**: Regarding mismatch between theoretical baselines and experimental baselines. It is unclear, if some theoretical baseline can be applied to TPAUC and perform better.
>
> **Response**: (1) We would like to point out that the baseline SOTAs is a state-of-the-art method for optimizing TPAUC as demonstrated in [39]. (2) The complexity of the baseline SOTAs for TPAUC maximization [39] matches that of the theoretical baseline SOX in Table 1. SOTAs can be considered as an extension of SOX for solving a smoothed surrogate of TPAUC loss [39].  (3) A theoretical  baseline named SOTA for optimizing the same TPAUC formulation as ours has been pointed out in [39] (as discussed in lines 114-117), which reformulates the problem into a min-max problem by leveraging the convexity of the outer function and adopts an existing double-loop algorithm. In comparison, our algorithm is single loop and has the same iteration complexity. We have implemented SOTA and compared the training convergence on the two molecular datasets.  The training convergence results as shown in Figure 3 in the global response demonstrate that our method has competitive and sometimes even faster convergence though our method does not explore the convexity of the outer function.
>
> **Q2**: Is there a lower bound for this problem setting, which can be used to verify if the rates are optimal and cannot be improved?
>
> **Response**: To the best of our knowledge, there is no known lower bound for NSWC FCCO. However, the complexity of our proposed methods matches the best known complexity for solving weakly-convex concave min-max problems. With additional convexity assumption on $f_i$, NSWC FCCO can be rewritten as a weakly-convex concave min-max problem. Thus we consider the complexity of our proposed methods to be state of the art.
>
>
> **Q3**: Why is the dependence on n1/B1 worse for SONT as compared to SONX?
>
> **Response**: The worse dependence on $\frac{n_1}{B_1}$ is caused by the two layers of block-sampling in SONT. Since TCCO problem is three-level compositional, we apply block-sampling strategy in the estimation of both the 1st inner function $\{g_i\}$ and 2nd inner function $\{h_{i,j}\}$. This results in an increasing inaccuracy in the overall gradients and function values estimation, and eventually leads to a worse dependence on $\frac{n_1}{B_1}$.

---

> > ### Comment · Reviewer_evM3 · 2023-08-18
> > **Response**
> >
> > Thanks for the detailed rebuttal, especially the clarification about the baseline SOTAs.

---

### Official Review · Reviewer_mWSr · 2023-07-24

**Soundness:** 3 good
**Presentation:** 3 good
**Contribution:** 2 fair
**Rating:** 6
**Confidence:** 2

**Summary:**

The purpose of this paper is to introduce a new approach to solving a specific type of optimization problem called non-smooth weakly-convex finite-sum coupled compositional optimization (NSWC FCCO) problems. The authors specifically focus on a variation of FCCO problems where the outer function is weakly convex and non-decreasing, and the inner function is weakly convex.

To address these problems, the authors propose two new algorithms. The first algorithm is designed for two-level NSWC FCCO problems and operates using a single-loop. The second algorithm is an extension of the first algorithm and is intended for tri-level NSWC FCCO problems. The authors provide a comprehensive analysis. They establish the complexity of the algorithms in terms of finding an $ε$-stationary point of the Moreau envelope of the objective function.

**Strengths:**

The paper is of high quality as it provides a detailed and rigorous mathematical analysis of the proposed algorithms. The technique of using Moreau envelope is intuitive and interesting. I have not checked the proof line by line but it looks good.  The authors also conduct extensive experiments to validate their theoretical findings. They compare the performance of their algorithms with other competitive methods across multiple datasets, demonstrating the effectiveness of their approach.

**Weaknesses:**

The authors have presented new convergence analysis for FCCO and TCCO, which I acknowledge. However, it seems that other multistage algorithms have already achieved the optimal rate. Consequently, I would lower my evaluation score for the novelty aspect.

**Questions:**

I acknowledge and value the authors' dedication in exploring the intricacies of the technical aspects. Nevertheless, I fail to find the application of maximizing two-way partial AUC particularly inspiring. Could you please demonstrate any other compelling applications?

---

> ### Author Rebuttal · Authors · 2023-08-09
>
> We thank reviewer mWSr for the insightful review. Here we would like to address your concerns.
>
> **Q1**: It seems that other multistage algorithms have already achieved the optimal rate.
>
> **Response**: Thank you for acknowledging  our new convergence analysis.  However, there is some **misunderstanding** of our results. No previous algorithms and analysis of non-convex FCCO (even the broader family of non-convex compositional optimization) are applicable to our considered non-smooth weakly-convex FCCO/TCCO problems as they assume that $f_i$ and $g_i$ are smooth. Our work is the first to study and solve non-smooth weakly-convex FCCO and TCCO problems, i.e., where $f_i$ and $g_i$ are non-smooth weakly-convex. All the possible existing solutions to non-smooth FCCO [20,26,36,38,15] need to reformulate FCCO as a min-max problem assuming  the convexity assumption of $f_i$.
>
>
>
> **Q2**: Could you please demonstrate any other compelling applications?
>
> **Response**: (1) We would like to point out that two-way partial AUC maximization is particularly important in medical domains where it is important to restrict false positive rate to be small and true positive rate to be large (Yang et al. 2019). (2) We would like to provide another important application of NSWC FCCO for regularized group distributionally robust optimization (group DRO), which is useful for addressing distributional shift (Sagawa et al. 2020).  Consider $N$ groups with different distributions. Each group $k$ has an averaged loss $L_k(w)=\frac{1}{n_k}\sum_{t=1}^{n_k}\ell (f_w(x_t^k),y_t^k)$, where $w$ is the the model parameter and $(x_t^k, y_t^k)$ is a data of the $k$-th group. For robust optimization, we assign different weights to different groups and form the following robust loss minimization problem:
> $$
> \min_w \max_{p\in \Omega} \sum_{k=1}^N p_k L_k(w),
> $$
> where $\Omega\subset\Delta$ and $\Delta$ denotes a simplex. A common choice for $\Omega$ is $\Omega=$\{ $p\in\Delta, p_i\leq 1/K$\} where $K\leq N$ is an integer, which yields the so-called CVaR losses (i.e., average of top-K group losses). As a result, the above problem is equivalent to (Curi et al, 2019):
> $$
> \min_w \min_{s} F(w,s)=\frac{1}{K}\sum_{k=1}^N [L_k(w)-s]_+ + s.
> $$
> We can map this problem into non-smooth weakly-convex FCCO when the loss function $\ell(\cdot,\cdot)$ is weakly convex in terms of $w$. Compared with solving the min-max problem, solving the above FCCO problem does not involve dealing with the projection onto the constraint $\Omega$ and avoid expensive sampling as in existing works (Curi et al, 2019).
>
>
>
> References:
>
> Yang et al. Two-way partial auc and its properties.Statistical methods in medical research,28(1):184–195,2019.
>
> Sagawa et al. Distributionally Robust Neural Networks for Group Shifts: On the Importance of Regularization for Worst-Case Generalization. ICLR, 2020.
>
>
> Curi et al. Adaptive sampling for stochastic risk-averse learning, 2019.

---

> > ### Comment · Reviewer_mWSr · 2023-08-19
> >
> > Thank you for the detailed reply, the paper looks more promising now and i will increase the score.

---

### Official Review · Reviewer_SCt1 · 2023-07-27

**Soundness:** 2 fair
**Presentation:** 2 fair
**Contribution:** 2 fair
**Rating:** 5
**Confidence:** 5

**Summary:**

The manuscript studies a class of non-smooth non-convex compositional optimization problems in which the objective function is given in a form of finite-sum composition where the functions are assumed to be weakly convex. The authors present stochastic approximation algorithms to solve this class of problems and establish their sample complexity bounds.

**Strengths:**

The authors provide the sample complexity bound of $O(1/\epsilon^6)$ for a single-loop algorithm applied to non-smooth weakly convex compositional problems.

**Weaknesses:**

Given the existing works in the literature and assumptions made in the manuscript, the presented theoretical results are incremental.

**Questions:**

1. In Table 1, the authors compare their results with the existing ones for smooth problems. However, dependence on $n$ is hidden and the reader cannot compare this dependency. Moreover, they need to add more existing results (mentioned on page 3) to the table with their assumptions so that there is a clear picture for comparison.

2. The choice of bath sizes appear in the complexity bounds, however, there is a very limited discussion on its role.

**Limitations:**

.

---

> ### Author Rebuttal · Authors · 2023-08-09
>
> We thank reviewer SCt1 for the detailed and insightful review. Here we would like to address your concerns.
>
> **Q1**. Difference from  the existing works in the literature.
>
> **Response**: We politely disagree with the reviewer that our work is incremental in light of existing works. There are fundamental differences between our work and existing works. This is the **first work** studying finite-sum coupled compositional optimization (FCCO) problems in **the nonsmooth setting**, i.e., no smoothness assumption is imposed on neither $f_i$ nor $g_i$. (1)  As discussed in lines 47-51 for previous works on smooth FCCO and lines 65-69 for our work on the analysis of convergence, there is a key difference in the analysis. Previous works heavily reply on the smoothness of the outer function $f_i$ for bounding the error of stochastic gradient estimator, which is not applicable to our setting.  (2) Our assumptions about weak convexity assumptions on $f_i$ and $g_i$ are **weaker** than smoothness assumption made in existing works [13,21], as the latter implies the former. No existing method can solve non-smooth weakly-convex FCCO unless additional assumptions are added. (3) Moreover, our results bring new algorithms for solving  a family of weakly-convex concave min-max problems in the form of $\min_{x}\max_y \frac{1}{n}\sum_iy_i g(x) - f^*(y_i)$, where $f^*$ is convex conjugate of a convex function $f$. The best existing methods are double loop methods with complexity $O(\epsilon^{-6})$ [20,26,36,38]. However, our method is single loop with the same complexity.  Last but not least, the considered non-smooth weakly-convex TCCO is novel, which has important applications in ML but no efficient solution is available in existing works.
>
>
> **Q2**. Regarding the dependence on $n$ and the roles of batch sizes  in the complexity bounds.
>
> **Response**: We will summarize the detailed complexities exhibited in our theorems into Table 5 (can be found in the pdf file from the global response) revealing the dependence on $n$ and the batch sizes as below. We have also discussed the dependence on the batch sizes and $n$ in detail at lines 230-234 for SONX, and lines 239-242 for SONT. Our SONX algorithm for non-smooth FCCO has the same dependence on the batch sizes and $n$ as MSVR [13] for smooth FCCO. The dependence of SONT's complexity on these parameters is more complex than that of SONX. Nevertheless, we have provided some discussions in lines 239-242. Overall, we can see that increasing the batch sizes plays a role of accelerating the convergence.
>
>
> **Q3**. Comparison with more existing results (mentioned on page 3) to the table with their assumptions so that there is a clear picture for comparison.
>
> **Response**: Please see Table 5 (can be found in the pdf file from the global response) for the comparison of more existing results. We will provide Table 1 in the revision.

---

> > ### Comment · Reviewer_SCt1 · 2023-08-17
> >
> > Thanks for the clarification. I increased my rating to 5.

---

> > > ### Author Response · Authors · 2023-08-17
> > > **Thank you for raising your rating!**
> > >
> > > We are glad that our rebuttal help address your concerns. Thank you!

---

### Author Rebuttal · Authors · 2023-08-09

We thank all the reviewers from their insightful reviews. Please find Figure 3 and Table 5 in the attached pdf file.

---

### Decision · Program_Chairs · 2023-09-21

**Decision:**

Accept (poster)

**Comment:**

The reviewers generally find that the paper contains good contributions. In particular, it provides new results on the FCCO problem in the weakly convex (thus can be both non-convex and non-smooth) setting. When preparing the camera-ready version, please incorporate the reviewers' comments and clarify issues related to technical rigor.